# An improved near real-time precipitation retrieval for Brazil

Simon Pfreundschuh[1], Ingrid Ingemarsson[1], Patrick Eriksson[1], Daniel A. Vila[2], and Alan J. P. Calheiros[3]

[1]Department of Space, Earth and Environment, Chalmers University of Technology, Gothenburg, Sweden
[2]Regional office for the Americas, World Meteorological Organization, Asunción, Paraguay
[3]Coordination of Applied Research and Technological Development, National Institute for Space Research (INPE), São José dos Campos, Brazil

**Correspondence:** Simon Pfreundschuh (simon.pfreundschuh@chalmers.se)

**Abstract.**

Observations from geostationary satellites can provide spatially continuous coverage at continental scales with high spatial and temporal resolution. Because of this, they are commonly used to complement ground-based precipitation measurements, whose coverage is often more limited.

We present Hydronn, a neural-network-based, near real-time precipitation retrieval for Brazil based on visible and infrared (VIS/IR) observations from the Advanced Baseline Imager (ABI) on the Geostationary Operational Environmental Satellite 16. The retrieval, which employs a convolutional neural network to perform Bayesian precipitation retrievals, was developed with the aims of (1) leveraging the full potential of latest-generation geostationary observations and (2) providing probabilistic precipitation estimates with well-calibrated uncertainties. The retrieval is trained using more than three years of co-locations with combined radar and radiometer retrievals from the Global Precipitation Measurement (GPM) core observatory over South America.

The accuracy of instantaneous precipitation estimates is assessed using a separate year of GPM combined retrievals and compared to retrievals from passive microwave (PMW) sensors and HYDRO, the VIS/IR retrieval that is currently in operational use at the Brazilian Institute for Space Research. Using all available channels of the ABI, Hydronn achieves accuracy close to that of state-of-the-art PMW precipitation retrievals in both precipitation estimation and detection despite the lower information content of the VIS/IR observations.

Hourly, daily and monthly precipitation accumulations are evaluated against gauge measurements for June and December 2020 and compared to HYDRO, the Precipitation Estimation from Remotely Sensed Information using Artificial Neural Networks Cloud Classification System and the Integrated Multi-Satellite Retrievals for GPM (IMERG). Hydronn reduces the mean absolute error for hourly accumulations by 21 %(22 %) compared to HYDRO by 44 % (41 %) for the MSE and increases the correlation by 138 %(312 %) for June (December) 2020. Compared to IMERG, the improvements correspond to 16 % (14 %), 12 % (12 %) and 20 % (56 %), respectively. Furthermore, we show that the probabilistic retrieval is well calibrated against gauge measurements when differences in the distributions of the training data and the gauge measurements are accounted for.

Hydronn has the potential to significantly improve near real-time precipitation retrievals over Brazil. Furthermore, our results show that precipitation retrievals based on CNNs that leverage the full range of available observations from latest-generation geostationary satellites can provide instantaneous precipitation estimates with accuracy close to that of state-of-the-art PMW

retrievals. The high temporal resolution of the geostationary observation allows Hydronn to provide more accurate precipitation accumulations than any of the tested conventional precipitation retrievals. Hydronn thus clearly shows the potential of deep-learning-based precipitation retrievals to improve precipitation retrieval from currently available satellite imagery.

## 1  Introduction

Timely and highly-resolved measurements of precipitation constitute an important source of information for weather forecasting, disaster response and hydrological modeling. These measurements can be provided by dense radar and gauge networks but their coverage is typically limited in less populated regions. However, even where these measurements are available, they are not necessarily without issues. The ability of rain gauge measurements to truthfully represent spatial precipitation statistics at larger scales is limited by their extreme localization (Smith et al., 1996). Ground-based precipitation radars are affected by beam blocking as well as measurement errors caused by the varying altitude of the radar beam along its range (Holleman, 2007).

Since satellite observations provide continuous spatial coverage, they are well suited to complement the measurements from gauges and ground radars. Microwave observations generally provide the most direct space-borne measurements of precipitation because of their sensitivity to emission and scattering from precipitating hydrometeors. Unfortunately, due to their comparably low spatial resolution, these sensors are currently employed only on polar orbiting platforms. Since this limits the width of the satellite swath, a large constellation of sensors on different platforms is required to achieve low revisit times. This is the approach pursued by the Global Precipitation Measurement (GPM, Hou et al., 2014). Nonetheless, the mean revisit time for the passive microwave sensors (PMW) of the GPM constellation still exceeds 1 hour in the tropics.

Visible and infrared (VIS/IR) observations from the latest generation of geostationary satellites (Schmit et al., 2005) provide spatial resolutions between 0.5 and 2 km at the sub-satellite point and a temporal resolution of up to 10 minutes for full disk observations. The disadvantage of these observations for measuring precipitation is that they are mostly sensitive to the properties of the upper parts of clouds, which are only indirectly related to the precipitation near the surface. Their unrivaled spatial and temporal resolution makes them a valuable source of information for satellite-based precipitation estimates nonetheless.

The operational use of geostationary VIS/IR observations for precipitation retrievals dates back more than 40 years (Scofield and Oliver, 1977) and a large number of different algorithms have been developed over the years (Arkin and Meisner, 1987; Adler and Negri, 1988; Vicente et al., 1998; Sorooshian et al., 2000; Kuligowski, 2002; Scofield and Kuligowski, 2003; Hong et al., 2004; Kuligowski et al., 2016). Due to the aforementioned indirect relationship between observations and precipitation, nearly all of these methods are based on empirical relationships derived from satellite observations co-located with reference data derived from more direct measurement techniques such as ground-based radar. Moreover, operational retrievals often rely on corrections to improve the accuracy of their estimates. The Self-Calibrating Real-Time GOES Rainfall Algorithm for Short-Term Rainfall Estimates (SCaMPR, Kuligowski et al. 2016), for example, is dynamically calibrated using the latest available microwave precipitation estimates. Similarly, Karbalaee et al. (2017) develop a correction for the Precipitation Estimation from Remotely Sensed Information using Artifical Neural Networks (PERSIANN) Cloud Classification System (CCS, Hong et al.,

2004) based on retrievals from passive-microwave sensors. PERSIANN CCS is superseded by the PERSIANN PDIR (Nguyen et al., 2020) algorithm, which, in addition to refining the mathematical formulation of the regression scheme of PERSIANN CCS, adds a regional correction scheme.

Another example is the HYDRO precipitation retrieval that is currently in operational use at the National Institute for Space Resarch (INPE) in Brazil, which is based on the Hydroestimator algorithm (Scofield and Kuligowski, 2003). It employs

an empirical relationship between the $10.7\,\mu m$ IR channel and precipitation rates with additional corrections. To adapt it for application over South America yet another correction was derived by de Siqueira and Vila (2019), which improved the accuracy of precipitation accumulations but not that of instantaneous precipitation rates.

A common shortcoming of all retrieval algorithms discussed above is that they neglect retrieval uncertainties. The retrieval of precipitation rates from VIS/IR observations constitutes an inverse problem that is strongly underconstrained. This is true

even for microwave based retrievals and likely exacerbated by the less direct information content in the VIS/IR observations. The ill-posed character of the retrieval problem leads to significant retrieval uncertainties. Providing probabilistic estimates that quantify these uncertainties would help the characterization of precipitation estimates and thus increase their usefulness.

This study presents Hydronn, a novel real-time precipitation retrieval that uses VIS/IR observations from the GOES 16 Advanced Baseline Imager (ABI, Schmit et al., 2005) to retrieve precipitation over Brazil. It was designed with two aims: (1)

To leverage the full potential of observations from the latest generation of geostationary sensors and (2) to develop a Bayesian precipitation retrieval algorithm that can provide well-calibrated uncertainty estimates.

Pfreundschuh et al. (2018) have shown that when a retrieval is cast as a probabilistic regression problem and solved using a neural network, the obtained results are equivalent to those obtained using traditional Bayesian retrieval methods, given that the a priori distribution matches the distribution of the data used to train the neural network. Neural-network-based proba-

bilistic regression techniques thus provide a powerful and flexible way of combining recent advances in deep learning with the theoretically sound handling of retrieval uncertainties of Bayesian retrieval methods. Hydronn builds on this approach and uses a convolutional neural network (CNN) to predict a binned approximation of the probability density function (PDF) of the marginal posterior distribution of each output pixel.

The Hydronn retrieval is trained using more than three years of collocated observations from the ABI and combined radar-

radiometer retrievals from the GPM core observatory (Grecu et al., 2016) over South America. The accuracy of Hydronn's instantaneous precipitation estimates is evaluated using a separate year of GPM combined retrievals and compared to HYDRO and the Goddard Profiling Algorithm (GPROF, Kummerow et al., 2015) applied to PMW retrievals from the GPM Microwave Imager (GMI). The accuracy of precipitation accumulations is evaluated using gauge measurements from June and December 2020. They are compared HYDRO, and two other commonly used precipitation products: PERSIANN CCS, which is based

on geostationary IR observations only, and the Integrated Multi-Satellite Retrievals for GPM (IMERG, Huffman et al. 2020), which combines observations from microwave, geostationary sensors, and rain gauges.

## 2 Data

This section introduces the various datasets that are used to train and evaluate the Hydronn retrievals.

### 2.1 GPM CMB

The GPM DPR and GMI Combined Precipitation product (William Olson, 2017) combines observations from the dual-frequency precipitation radar (DPR) and GMI on board the GPM core observatory (Grecu et al., 2016). Although the officially listed shortname for the product is GPM_2BCMB (William Olson, 2017), we will refer to it as GPM CMB since we consider it more readable. Because of the high sensitivity to precipitating hydrometeors of the active and passive microwave observations, the product provides the most accurate space-borne precipitation estimates that are currently available. In this study the product is used as reference data to train the Hydronn retrievals and to assess their accuracy for instantaneous precipitation estimates.

### 2.2 Rain gauge data

The rain gauge measurements that are used in this study were compiled by the National Institute of Meteorology of Brazil and consist of hourly gauge measurements covering the time range May 2000 until May 2020. June and December of 2020 will be used from this data for the evaluation of Hydronn. Data from 2018 and 2019 is used to derive correction factors for the calibration of the hourly precipitation estimates produced by Hydronn, as will be described in Sec. 3.6.

From all available gauge stations only those with a data availability exceeding 90 % during June and December 2020 were selected. Their geographical distribution is displayed together with the mean precipitation in Fig. 1. The gauge density is fairly high on the south-eastern coast of Brazil but decreases markedly towards the northwest.

The precipitation in June 2020 is limited to the south of the country, small parts of the west coast, and the amazon basin, although the latter is only sparsely covered by the gauge observations. June is typically the beginning of the dry seasons in the central part of the country, which is clearly visible the gauge measurements.

December 2020 saw high precipitation amounts on the south-western coast of the country extending towards the northwest, which are associated the South Atlantic Convergence Zone (SACZ, Satyamurty et al., 1998). Very low precipitation rates are observed in the northeast of the country, which is influenced by large scale subsistence patterns (de Siqueira and Vila, 2019).

### 2.3 HYDRO

HYDRO is the currently operational near real-time precipitation retrieval at the Center for Weather Forecast and Climate Studies/National Institute for Space Research (CPTEC/INPE). It is based on the Hydroestimator (Scofield and Kuligowski, 2003) and thus uses a combination of empirical power-law relationships between $10.7\,\mu m$ IR brightness temperatures and surface precipitation with correction factors taking into account model-derived moisture and wind parameters as well as cloud structure. The current version of the retrieval is described in de Siqueira and Vila (2019), which also introduces regional correction factors based on a climatology of surface precipitation rates derived from radar measurements of the Tropical Rainfall Measurement Mission (TRMM, Simpson et al., 1996) and GPM. For this study we use the corrected version of

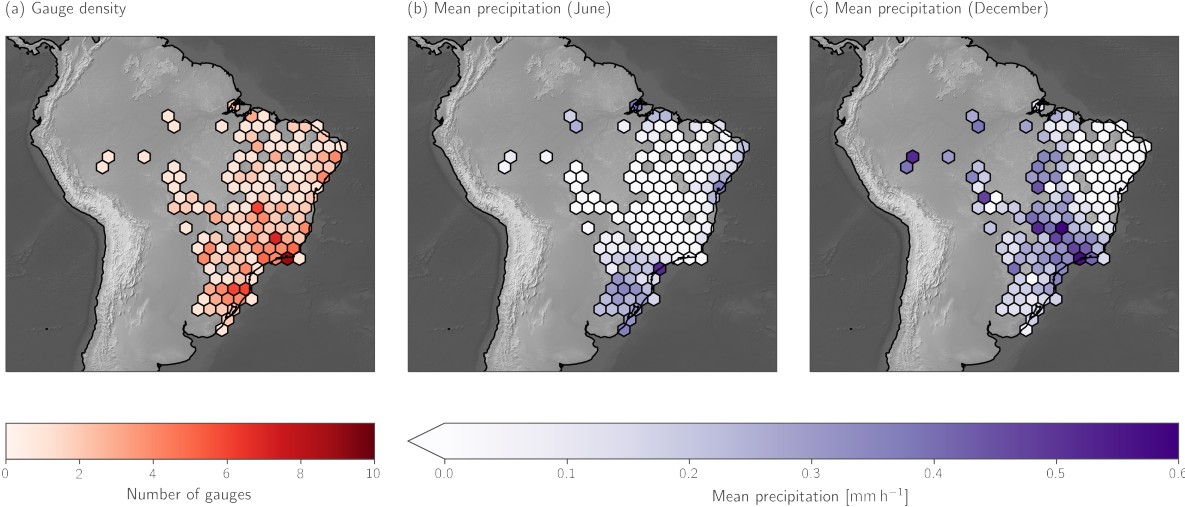

**Figure 1.** Overview of the rain gauge data from June and December 2020 used to validate the retrievals. Panel (a) displays their spatial distribution by means of the number of gauges falling into each hexagon. Hexagon-free areas are not covered by any gauges. Panels (b) and (c) show the mean precipitation measured by all gauges falling into each hexagon.

HYDRO proposed in de Siqueira and Vila (2019) with a regional correction for all of Brazil (referred to as HYDROBR in de Siqueira and Vila (2019)).

## 2.4 GPROF GMI

The Goddard Profiling Algorithm (GPROF, Kummerow et al., 2015) is used to retrieve surface precipitation from each of the PMW sensors of the GPM constellation. The algorithm is a Bayesian retrieval scheme that is based on a retrieval database principally built up of collocations of GMI observations and the GPM CMB retrievals. Because GMI is a dedicated precipitation sensor and because its retrieval is based on direct collocations with GPM CMB, the GPROF GMI retrieval is considered the most accurate of the sensors of the GPM constellation. Moreover, since GMI observations can always be collocated with the test data for the Hydronn retrieval, we use GPROF GMI as a baseline to assess the Hydronn retrievals against.

## 2.5 PERSIANN CCS

PERSIANN CCS (Hong et al., 2004) uses $10.7\,\mu\text{m}$ IR observations from geostationary satellites to retrieve precipitation. Input images are first segmented using increasing temperature thresholds in order to identify pixels that correspond to convective activity. These pixels are consecutively assumed to be precipitating and classified using a neural network based algorithm. Quantitative precipitation estimates at pixel level are derived from this classification by applying a class-specific power law relationship that relates the $10.7\,\mu\text{m}$ brightness temperatures to precipitation.

The dataset that is used for the evaluation against Hydronn are hourly precipitation rates that are distributed in near real time through the PERSIANN data portal (UCI CHRS Data Portal, 2022). Although the global CCS dataset is currently being replaced by the updated PERSIANN Dynamic Infrared-Rain rate (PDIR-Now, Nguyen et al., 2020) we were not able to use it for this study due to parts of the evaluation period missing from the online archive.

## 2.6 IMERG

IMERG (Huffman et al., 2020) combines retrievals from passive microwave and IR observations as well as rain gauge measurements to produce global, half-hourly measurements of precipitation. Due to its reliance on a wealth of measurement sources as well as the sophistication of the retrieval pipeline, the product can be considered one of the most robust satellite-based precipitation products that are currently available (Pradhan et al., 2022).

Three different configurations of IMERG products are available: IMERG-Early and IMERG-Late are based solely on satellite observations and available with latencies of 4 and 14 hours, respectively. IMERG-Final is adjusted using global gauge measurements but available only after 3.5 months. Although Hydronn has been designed to target near real-time applications and is thus more similar to IMERG-Early, we use IMERG-Final for our comparison as it constitutes the most elaborate precipitation estimates that are currently available and can thus be considered the state of the art of global quantitative precipitation estimates.

## 3 Method

This section describes the implementation of Hydronn, the proposed near real-time precipitation retrieval algorithm for Brazil. It is based on a convolutional neural network (CNN), which is used to predict the a posteriori distribution of instantaneous precipitation. Following this, it is discussed how the probabilistic precipitation estimates can be combined to hourly accumulations and an a priori adjustment is proposed to account for differences between the training data and the gauge measurements that are used to evaluate the retrieval.

### 3.1 Training data

The training data for the Hydronn retrieval is generated from co-locations of input observations from the GOES-16 ABI and retrieved surface precipitation from GPM CMB over South America. Figure 2 shows the domain over which the training data was extracted (marked as 'R1' in Fig. 2). It extends from $-85\,°\mathrm{E}$ to $-30\,°\mathrm{E}$ in longitude and $-40°\mathrm{N}$ to $10\,°\mathrm{N}$ in latitude. The plot also shows extracted training scenes and corresponding GPM CMB precipitation estimates for September 23, 2019.

The GOES 16 ABI observations were extracted at their native resolutions. The surface precipitation from GPM CMB was mapped to the $2\,\mathrm{km}$ resolution of the ABI's IR channels using nearest-neighbor interpolation. Collocations were extracted for the time range 1 January 2018 until 1 January 2020 and 1 January 2021 until 1 September 2021.

Collocations from 1 January 2020 until 1 January 2021 were extracted and set aside as test data for assessing the accuracy of the instantaneous precipitation estimates of Hydronn. In addition to this, collocations over an additional region (marked 'R2'

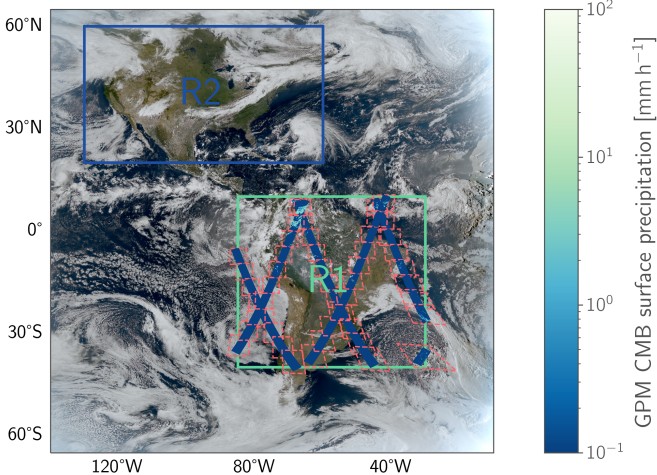

**Figure 2.** GOES-16 true-color composite from 23 September 2019 (generated using the `natural_color` composite in satpy (Raspaud et al., 2021)). The rectangle R1 marks the domain over South America, which was used for the extraction of training and testing collocations between GOES 16 ANI and GPM CMB. Dashed polygons show the boundaries of the training scenes extracted for this day together with the collocated GPM CMB results. The rectangle R2 marks the secondary domain which is used as an additional test domain to assess the impact of the spatially limited training domain.

in Fig. 2) were extracted on days 1, 6, 11, 16, 21, 26 of each month of the year 2020. This additional test data will be used to investigate the impact of the regional training database on the retrieval accuracy.

The correspondence between probabilistic neural-network retrievals and Bayesian retrieval methods shown in Pfreundschuh et al. (2018) emphasizes the importance of training data distribution for the probabilistic retrieval results. Since the retrieval uncertainties depend on the distribution of the training data, the retrieval can provide well-calibrated probabilistic predictions only for test data whose distribution is consistent with that of the training data. The distribution of precipitation rates in the training dataset is displayed in Fig. 3. The detection threshold for precipitation of the GPM radar between $0.2$ and $0.4\,\mathrm{mm\,h^{-1}}$ is clearly visible in the distributions. In addition to this, a weak seasonal cycle is apparent, which mainly impacts the likelihood of moderate precipitation. The gauge measurements exhibit stronger seasonal variability especially for strong rain. It should be noted here, however, that the precipitation estimates in the training data correspond to instantaneous precipitation estimates while the gauge measurements are integrated over the time of an hour. Differences between the seasonal cycles of the datasets may therefore be caused by changes in the temporal evolution of precipitation events. An approach to reconcile the differences between the distributions of training and validation will be proposed in Sec. 3.6 below.

### 3.2 Retrieval configurations

The Hydronn retrieval has been implemented in three different configurations in order to assess how the choice of input observations and their resolution affects its performance. The most basic retrieval configuration is the $\mathrm{Hydronn_{4,IR}}$ retrieval,

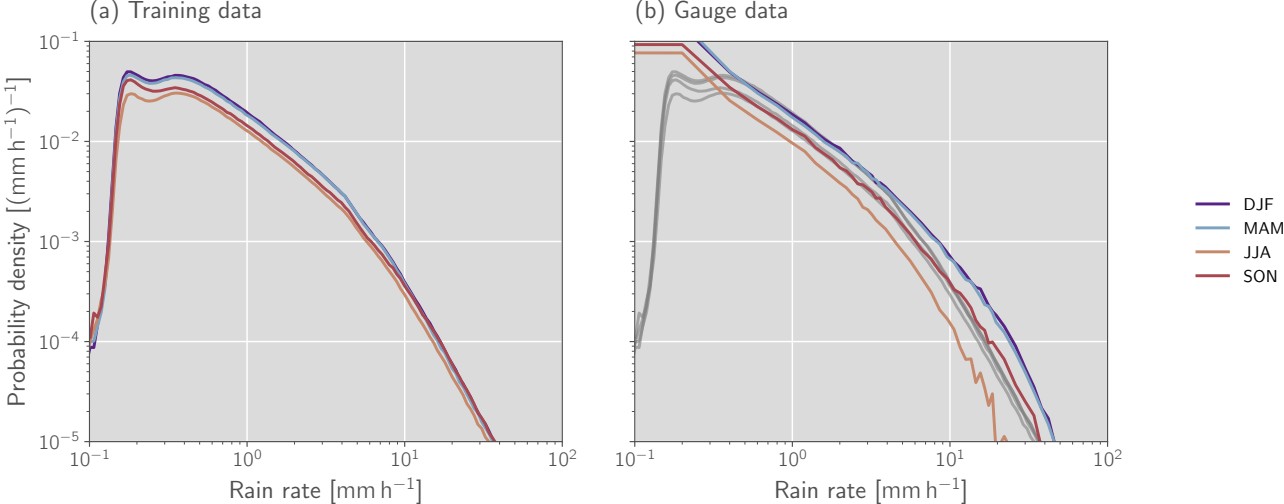

**Figure 3.** Distribution of reference precipitation rates. Panel (a) shows the seasonal PDFs of precipitation rates in the training data. Panel (b) shows the PDFs of precipitation rates measured by the gauges over the time period covered by the training data. Grey lines in the background trace the PDFs of the precipitation rates in the training data shown in Panel (a).

which only uses brightness temperatures from the $10.3\,\mu\text{m}$ channel as input. The same longwave-IR window channel is also used by HYDRO and the PERSIANN CCS retrieval. The availability of similar channels on a long time series of geostationary sensors makes them suitable for the generation of climate data records. Since this retrieval uses the same input as HYDRO and PERSIANN CCS it allows assessing the benefit afforded by the probabilistic, neural-network-based retrieval technique used by Hydronn.

The second retrieval configuration, denoted as $\text{Hydronn}_{4,\text{All}}$, uses all available GOES channels at a resolution of $4\,\text{km}$, which is the resolution at which both HYDRO and PERSIANN CCS are operating. This model configuration is included to assess the benefit of including all ABI channels in the retrieval.

The third configuration, $\text{Hydronn}_{2,\text{All}}$, uses all observations at their native resolution and retrieves precipitation at the resolution of the $2\,\text{km}$ channels. This means that GOES Ch. 2 is ingested at a resolution of $500\,\text{m}$, Ch. 1, 3 4 at a resolution

of $1\,\text{km}$ and the remaining channels at $2\,\text{km}$. This is in contrast to the other Hydronn configurations and other precipitation retrievals which typically ingest all observations at the same resolution. This configuration aims to explore the extent to which high resolution observations can improve precipitation retrievals even if the reference precipitation measurement only have a resolution of $5\,\text{km}$. Comparison of the $\text{Hydronn}_{4,\text{All}}$ and $\text{Hydronn}_{2,\text{All}}$ configuration aims to address the question whether the increased computational complexity of $\text{Hydronn}_{2,\text{All}}$ can be justified by improvements in retrieval accuracy. The characteristics

of the three configurations are summarized in Table 1.

**Table 1.** Hydronn retrieval configurations

| Name | Input bands | Input resolution | Output resolution |
|---|---|---|---|
| Hydronn$_{4,\text{IR}}$ | 13 | 4 km | 4 km |
| Hydronn$_{4,\text{All}}$ | $1, \ldots, 16$ | 4 km | 4 km |
| Hydronn$_{2,\text{All}}$ | $1, \ldots, 16$ | $0.5, 1$ and 2km | 2 km |

## 3.3 Neural network model

All Hydronn retrievals are based on a similar convolutional neural network (CNN) architecture, which is illustrated in Fig. 4. CNNs have been shown to be able to learn semantic features directly from image data (Selvaraju et al., 2017), which sets them apart from conventional regression techniques. Since satellite imagery of clouds exhibits patterns that can be related to different precipitation regimes, we expect this information to help to constrain the precipitation retrieval. In fact, a preliminary study we have conducted found that CNNs yield more accurate precipitation retrievals than a fully-connected neural network operating on independent pixels. The results have been published as parts of a Master's thesis and are available online (Ingemarsson, 2021).

CNNs principal building blocks are convolution layers. A convolution layer applies a set of convolution operations to an input image. The parameters of the layer's convolution kernels are learned during the training of the network. A convolution layers thus corresponds to a set of learnable image transformations. CNNs typically consist of a stack of convolution layers that are interleaved with normalization layers and activation functions. The activation functions are required to allow the CNN to represent non-linear transformations, while normalization has been found to be a crucial ingredient to accelerate the training of CNNs (Ioffe and Szegedy, 2015).

The neural networks used by Hydronn are built up of blocks, each of which comprises two separable convolution layers followed by batch normalization (Ioffe and Szegedy, 2015) and GELU activation functions (Hendrycks and Gimpel, 2016). These blocks were inspired by the Xception model proposed by Chollet (2017). A residual connection directly connects the input of each block to its output. These residual connection improve the flow of the gradients through the network and were found to be a crucial ingredient for the training of very deep CNNs (He et al., 2016).

The Xception blocks are organized into an encoder-decoder structure, which was popularized by the UNet model for image segmentation (Ronneberger et al., 2015). The first part of the model, the encoder, combines Xception blocks with downsampling layers. These downsampling layers reduce the size of the input image by a factor of two and thus double the receptive field of the following layers. This allows the network to efficiently combine information across different regions of the input image.

Following the encoder part of the network, the decoder consists of several stages each containing an upsampling layer and a single Xception block. The upsampling layers allow the network to combine the information extracted at coarser resolution back down to input resolution. As in the UNet model, skip connections between the corresponding stages of encoder and decoder are included to improve the flow of information through the network.

The head of the network employs 5 layers of $1 \times 1$ convolutions followed by normalization layers and GELU activation function. This network head is computationally equivalent to a fully-connected network that transforms the representation extracted by the encoder and decoder parts of the network to a probability distribution. A final $1 \times 1$ convolution maps the output for each pixel to a vector of length 128. The softmax activation function

$$\sigma(\boldsymbol{x}) = \frac{\exp(\boldsymbol{x})}{\sum_i \exp(x_i)} \tag{1}$$

is applied to each of those vectors to ensure that all elements are positive and sum to 1. This allows the results to be interpreted as probabilities of a categorical distribution.

The neural network used by the Hydronn$_{2,\text{All}}$ retrieval architecture contains two additional downsampling blocks to allow the network to ingest all ABI channels at their native resolution. These two blocks are omitted for the Hydronn$_{4,\text{IR}}$ and Hydronn$_{4,\text{All}}$ retrievals. The encoder and decoder of all Hydronn models comprise 5 stages. The number of internal features is set to $n_f = 256$ and the number of Xception blocks in each encoder stage to $N = 3$.

All available collocations from the training period are used for the training and no distinction is made between day- and nighttime observations. The Adam optimizer (Kingma and Ba, 2014) with an initial learning rate of $0.0005, \beta_1 = 0.9, \beta_2 = 0.99$ and a cosine-annealing learning rate schedule (Loshchilov and Hutter, 2016) is used for training. Warm restarts are performed every 20 epochs and repeated until the retrieval accuracy on a held-out part of the training data converges. Training of a single retrieval model takes about 3 days on an NVIDIA V40 GPU.

## 3.4 Probabilistic precipitation estimates

Hydronn builds on the findings from Pfreundschuh et al. (2018), which showed that probabilistic regression with neural network yields the same results as a traditional Bayesian retrieval using an a priori distribution that is the same as the training data of the neural network. Although Pfreundschuh et al. (2018) used quantile regression neural networks (QRNNs) to perform Bayesian remote sensing retrievals with neural networks, a different approach is taken here. Following the work by Sønderby et al. (2020), the range of possible precipitation values is discretized and the neural network output is used to predict the probabilities of the observed precipitation falling into any of the precipitation bins. By normalizing the predicted probabilities by the width of the corresponding bin, a binned approximation of the probability density function (PDF) of the Bayesian a posterior distribution can be obtained. We found this approach to be equivalent to QRNNs in retrieval accuracy. We chose this approach, which we will refer to as density regression neural network (DRNN), because we didn't find an efficient way to calculate the sum of two independent random variables from the quantiles predicted by a QRNN. For two PDFs given over discrete bins the sum can be calculated by weighing all possible sums of bin centers by the product of the corresponding probabilities, and accumulating the resulting probabilities into the bins of the result PDF.

DRNNs can be implemented by treating the retrieval as a classification problem over a discretized range of precipitation values and using the cross-entropy loss to train the network. The cross-entropy loss is defined as

$$L(\hat{\boldsymbol{y}}, y) = -\log(\hat{y}_{\text{bin}(y)}) \tag{2}$$

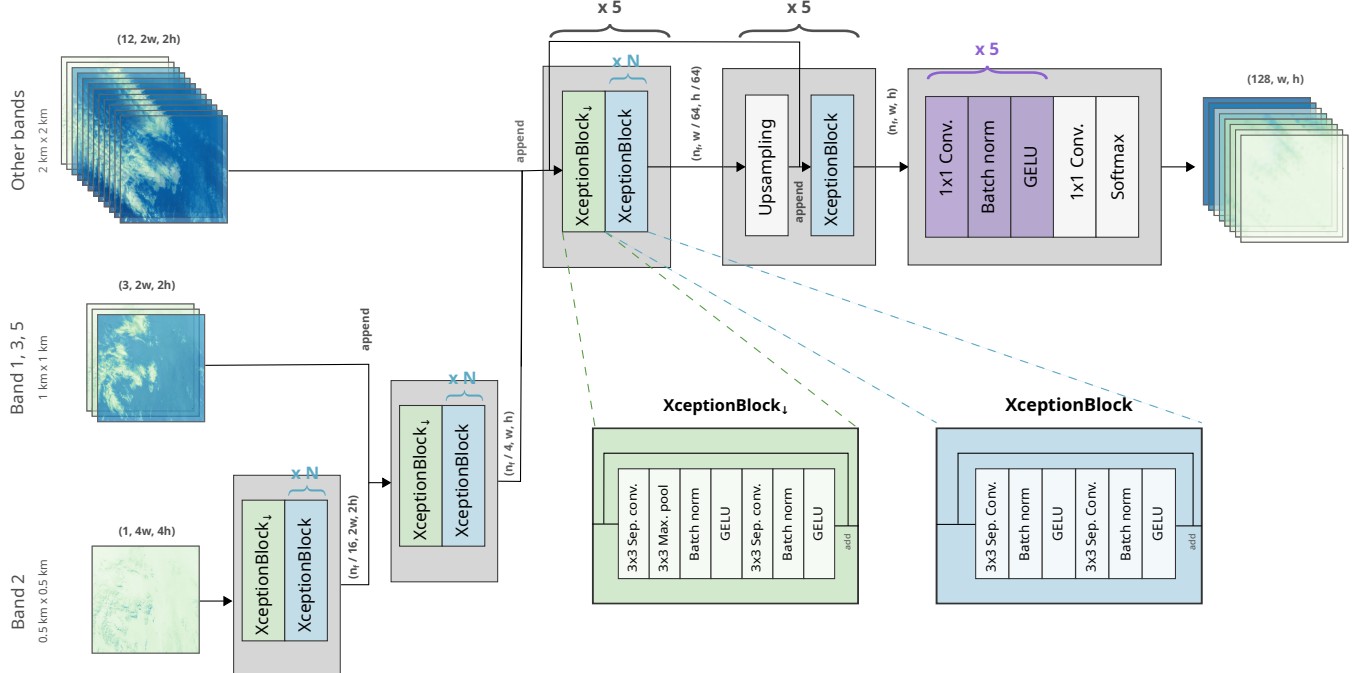

**Figure 4.** The neural network architecture used by the Hydronn$_{2,\text{All}}$ retrieval. For the Hydronn$_{4,\text{IR}}$ and Hydronn$_{4,\text{All}}$ retrievals the two additional stages for the processing of the higher-resolution inputs are omitted. Instead, the input comprises all input observations downsampled to four-kilometer resolution. Grey text in parenthesis gives the shape of the tensors at the various stages of the network using channel-first ordering and omitting the batch dimension. Grey, bold text specifies merge operations for data streams. Braces are used to mark the repetition of network components.

where $\hat{\boldsymbol{y}}$ is the vector of probabilities predicted by the network and $\text{bin}(y)$ is the index of the probability bin corresponding to the true precipitation rate $y$.

Hydronn predicts the a posteriori distribution over 128 logarithmically spaced bins covering the range from $10^{-3}$ to $10^{3}$ mm h$^{-1}$. This upper limit of the bin range may be unrealistically large but using this range has the advantage that we can compute the sum of two distributions on the same bins as the predictions are made.

The reference precipitation of pixels without rain was set to a log-uniform random value between $10^{-3}$ and $10^{-2}$ mm h$^{-1}$. Replacing zero values with small random values has the advantage of making the corresponding cumulative distribution function (CDF) continuous, which ensures that all quantiles of the distribution are always well defined. This allows us to verify the probabilistic predictions from the network using calibration curves.

Since storing the full posterior distribution for all pixels is of little use for operational processing, only a reduced number of relevant statistics are retained in the retrieval output. Those are the posterior mean as well as a sample and 14 quantiles of the posterior distribution. Note that the quantiles are always located around the region of the posterior distribution that contains most of its mass and thus provide a much compacter way of storing the probabilistic retrieval results than the full

128 probabilities. In addition to providing a direct measure of retrieval uncertainty, the quantiles can be used to reconstruct a piece-wise linear approximation of the cumulative distribution function (CDF) of the retrieved distribution. The CDF can then be used, for example, to detect the exceedence of certain precipitation thresholds. Compared to training a separate classifier to perform this task, this approach has the advantage that the precipitation threshold can be chosen dependent on the application context after the network has been trained.

### 3.5 Calculation of hourly accumulations

The precipitation estimates produced by Hydronn correspond to instantaneous precipitation rates. Since GOES 16 imagery is available every 10 minutes, a method is required to aggregate the retrieved distributions of the instantaneous precipitation rates to hourly accumulations in order to compare them to the gauge measurements. While this is not an issue when only the posterior mean is retrieved, it is unclear how retrieval uncertainties should be accumulated in time. The problem is illustrated in Fig. 5 using six, consecutive retrievals at a single output pixel. The green lines show the retrieved distributions for each input observation. Because Hydronn has no way of modeling the correlations between consecutive observations it is not clear how the instantaneous distributions can be aggregated to a posterior distribution for the hourly accumulations.

In lack of a formal way to resolve this, we have implemented two heuristics for calculating probabilistic estimates of hourly accumulations from instantaneous measurements. The first heuristic is to average the predicted posterior distributions. For the case of multiple identical observations, this preserves the retrieval uncertainties and thus corresponds to the assumption of strong dependence of the retrieval errors for consecutive observations. The second approach is to assume temporal independence of the retrieval uncertainty.

The blue and red curves in Fig. 5 show the resulting posterior distributions of the hourly accumulations for the assumptions of dependent errors and independent errors, respectively. Despite the differences in the two distributions they both have the same mean value. Under the assumption of temporal independence, the instantaneous retrieval errors have a tendency to compensate for each other, which reduces the retrieval uncertainty. Conversely, strongly dependent errors have a tendency to conserve the uncertainty of the instantaneous retrievals resulting in higher probabilities assigned to stronger precipitation.

For the evaluation of the Hydronn retrieval, we calculate PDFs of hourly accumulations using both approaches. Two types of accumulations are thus produced for each Hydronn configuration: One corresponding to the assumption of dependent retrieval errors, which will be identified with the qualifier '(dep.)', as well as one corresponding to the assumption of independent retrieval errors, which will be identified with the qualifier '(indep.)'. Since the assumptions only affect the predicted retrieval uncertainties and not the predicted mean values, such a distinction is not required when point estimates of precipitation are considered.

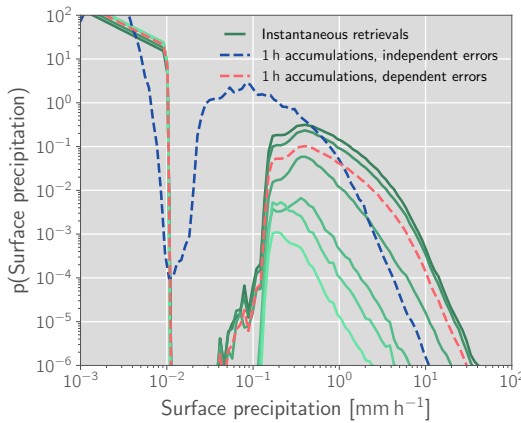

**Figure 5.** Retrieved posterior distributions of instantaneous precipitation (green, solid lines) for an hour of ABI observations. The corresponding derived distributions for the hourly accumulations are show in red and blue for the assumptions of dependent and independent errors, respectively.

## 3.6 Correcting for a priori assumptions

According to Bayes theorem, the posterior distribution of retrieved precipitation $p(x|\mathbf{y})$ for given input observations $\mathbf{y}$ is proportional to the product of the probability of observing $\mathbf{y}$ for a given precipitation rate $x$ and the a priori probability of $x$:

$$p(x|\mathbf{y}) \propto p(\mathbf{y}|x)p(x) \tag{3}$$

One difficulty with machine learning based retrievals is that the a priori distribution cannot be chosen freely but is dictated by the training data distribution. Fig. 3 indicates that there are inconsistencies between the training data and gauge measurements. For example, the retrieval will learn from the training data that the probability of precipitation values between $10^{-1}$ and $10^{-2}$ mm h$^{-1}$ is effectively zero.

This raises the question whether it is possible to correct for the effect of the a priori assumptions encoded in the training data of the retrieval. To explore this, we propose the following method to correct the probabilistic predictions. Let $p_{\text{Gauges}}(x)$ denote the PDF of precipitation as measured by the available gauges shown in Fig 3 (b). Moreover let

$$r(x) = \frac{p_{\text{Gauges}}(x)}{p(x)} \tag{4}$$

denote the ratio of the PDFs of the gauge measurements and the a priori distribution of precipitation as defined by the training data. Assuming that $p_{\text{Gauges}}(x) = 0$ wherever $p(x) = 0$ and that the conditional distribution $p(\mathbf{y}|x)$ of the observations remains unchanged, a corrected posterior distribution can be obtained by point-wise multiplying the likelihood ratio $r$ with the posterior distribution predicted by Hydronn:

$$p_{\text{Corrected}}(x|y) \propto p(y|x)r(x)p(x), \tag{5}$$

The difficulty with this approach is that we only know the a priori distribution corresponding to the instantaneous precipitation retrievals, i. e., the distribution of the training data, but not for the hourly accumulations retrieved using Hydronn. To infer them, we calculated hourly accumulations for randomly sampled hours over the full year of 2019 for each retrieval configuration. The resulting correction factors for the Hydronn$_{4,\mathrm{All}}$ retrieval are displayed in Fig. 6.

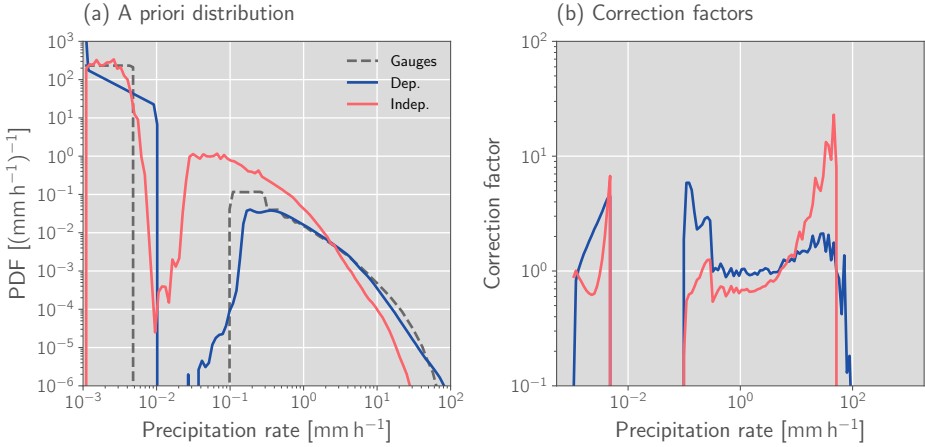

**Figure 6.** A priori distributions of hourly accumulations and derived correction factors $r$ for the Hydronn$_{4,\mathrm{All}}$ retrieval. Panel (a) displays the a priori distributions of hourly precipitation accumulations derived assuming strong temporal dependence of measurements (blue) and complete independence (red). The gray, dashed line shows the PDF of the gauge measurements. Panel (b) displays the corresponding correction factors for the two assumptions calculated as the ratio between the respective PDFs and the PDF of the gauge measurements.

## 4  Results

This section evaluates the Hydronn retrievals in three steps. First, the accuracy of the instantaneous precipitation estimates with respect to the GPM CMB reference data is assessed and compared to the GPROF GMI and HYDRO retrievals. Second, the retrieved precipitation accumulations from June and December 2020 are evaluated against gauge measurements and compared to accumulations from HYDRO, IMERG, and PERSIANN CCS. Third, a case study of a heavy, flood-producing precipitation event is presented. Definitions of the metrics used to evaluate the retrievals are provided in Appendix A1.

### 4.1  Instantaneous precipitation estimates

#### 4.1.1  Case study

As first step in the evaluation of the instantaneous precipitation estimates, we consider retrieved precipitation for an overpass of the GPM satellite over a meso-scale convective system (MCS) in the border region between Argentina, Paraguay and Brazil on 16 December 2020, 13:59:00 UTC. The retrieval results are displayed together with a natural color composite in Fig. 7. The

GPROF GMI and IMERG retrievals exhibit good agreement with the GPM CMB results. This is expected, not only because GPROF and IMERG incorporate PMW observations, but also because GPM CMB is used to derive the retrieval database used by GPROF, and GPROF is in turn used by IMERG.

The HYDRO retrieval, on the other hand, does not agree well with the GPM CMB results. The heavy precipitation retrieved by HYDRO is located in the western part of the MCS, whereas the GPM CMB shows the very heavy precipitation in the north-eastern parts of the system. The $\text{Hydronn}_{4,\text{IR}}$ retrieval captures the overall structure of the MCS better than HYDRO but fails to represent its smaller-scale structures. Both, the $\text{Hydronn}_{4,\text{All}}$ and $\text{Hydronn}_{2,\text{All}}$ retrievals improve upon this and yield results that are very similar to those of GPROF GMI and IMERG.

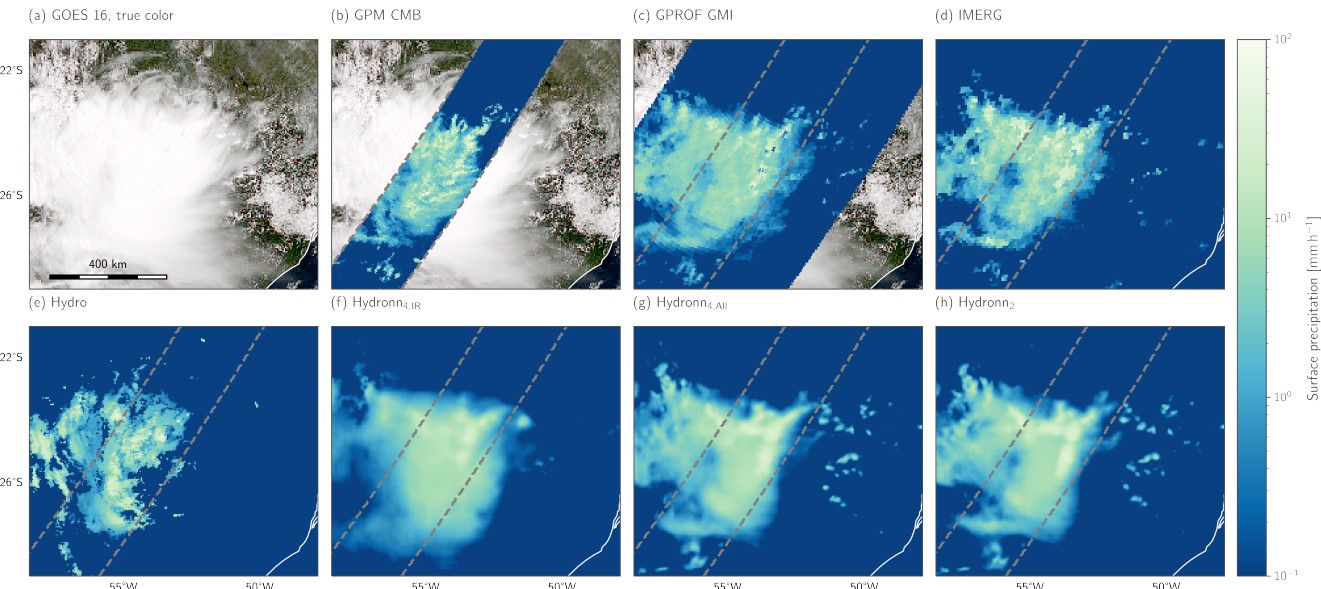

**Figure 7.** A mesoscale convective system over the border region between Argentina, Paraguay, and Brazil on 16 December 2020 observed by GPM and GOES 16. Panel (a) shows a natural color composite (generated using (Raspaud et al., 2021)). Panel (b) shows retrieved surface precipitation from CMB retrieved using combined radar and passive microwave observations. Panel (b) shows precipitation retrieved by GPROF GMI using only passive microwave observations. Panel (c) shows the surface precipitation from the IMERG Final product. Panel (d) shows surface precipitation retrieved by HYDRO from GOES ABI observation. Panels (e), (f), (g) show the corresponding results from the three Hydronn configurations.

Accuracy metrics for the the MCS overpass with respect to the CMB reference data are provided in Table 2. $\text{Hydronn}_{4,\text{IR}}$ and $\text{Hydronn}_{2,\text{All}}$ both exhibit dry biases of the same magnitude as HYDRO and $\text{Hydronn}_{4,\text{All}}$ even exceeds those. However, all Hydronn retrievals yield significantly more accurate results than HYDRO in terms of the other metrics. The $\text{Hydronn}_{2,\text{All}}$ retrieval even surpasses IMERG in terms of MSE, MAE, and CSI and achieves results close to those of GPROF GMI.

This evaluation indicates that, while the total amount of precipitation remains less accurate for Hydronn than for the PMW retrievals, the spatial structure of the retrieved precipitation is captured equally well. Moreover, it should be noted that the

**Table 2.** Retrieval accuracy metrics for the MCS overpass shown in Fig. 7. Definitions of all metrics can be found in appendix A1.

| Retrieval | Bias | MAE [mm h$^{-1}$] | MSE [(mm h$^{-1}$)$^2$] | Correlation | POD | FAR | CSI |
|---|---|---|---|---|---|---|---|
| Hydro | -0.598 | 2.495 | 46.291 | 0.228 | 0.707 | 0.169 | 0.618 |
| GPROF GMI | **−0.163** | **1.699** | **27.467** | 0.552 | **0.998** | 0.519 | 0.481 |
| IMERG | 0.28 | 2.204 | 40.664 | 0.429 | 0.973 | 0.213 | 0.77 |
| Hydronn$_{4,IR}$ | 0.612 | 2.36 | 30.768 | 0.506 | 0.901 | 0.192 | 0.74 |
| Hydronn$_{4,All}$ | 0.813 | 2.362 | 33.038 | 0.524 | 0.916 | **0.138** | 0.798 |
| Hydronn$_{2,All}$ | 0.57 | 2.105 | 29.918 | **0.564** | 0.922 | 0.14 | **0.801** |

revisit time for the GPM constellation of PMW sensors at these latitudes is around 1 h (Hou et al., 2014). Hydronn, however, can provide precipitation retrievals every 10 min. While increasing the temporal coverage of the precipitation measurements is also what IMERG aims to achieve by merging PMW retrievals with observations and retrievals from geostationary sensors, this seems to lead to a degradation of the accuracy of the precipitation estimates. To further illustrate the capabilities of the Hydronn retrievals a video of precipitation estimates for the MCS case is provided as a digital supplement to this manuscript (Pfreundschuh, 2022b).

### 4.1.2 Accuracy over target region

To assess the accuracy of the instantaneous precipitation estimates of Hydronn, we use collocations with GPM CMB from all GPM overpasses of the year 2020 over the target region, i.e., the region to which the Hydronn training data was restricted (R1 in Fig. 2). The results of Hydronn are compared to HYDRO and GPROF GMI. Since GPROF retrievals can be directly collocated with the results from CMB and because GPROF is used by IMERG, we chose GPROF GMI over IMERG for this comparison as it can be expected to provide a stronger baseline. This is corroborated by the case study presented above.

Fig. 8 displays the resulting PDFs of retrieved precipitation conditioned on the value of the reference precipitation for all assessed algorithms. While all distributions exhibit noticeable spread, GPROF shows the best agreement with the reference data. Conversely, HYDRO hardly shows any sensitivity to the strength of the reference precipitation. For the Hydronn results, slight improvements between the three configurations are discernible. While the Hydronn$_{4,IR}$ retrieval exhibits the weakest relationship between reference and retrieved precipitation, the Hydronn$_{4,All}$ configuration yields slightly more accurate results. This can be seen in the sharpening of the conditional PDFs for precipitation rates occurring between 2 and 20 mm h$^{-1}$ as well as an increase in the slope of the conditional mean retrieved precipitation for rain rates exceeding 2 mm h$^{-1}$. Clearer improvements in retrieval accuracy are observed for the Hydronn$_{2,All}$ configuration, which yields a slightly sharper distribution and an increased slope in the conditional mean for precipitation rates larger than 0.5 mm h$^{-1}$. Comparing the Hydronn$_{2,All}$ results to GPROF shows that the distributions are quite similar for precipitation rates below 10 mm h$^{-1}$. Above this threshold, GPROF shows better sensitivity to the reference rain rates whereas Hydronn$_{2,All}$ exhibits a stronger tendency to underestimation.

For a more quantitative analysis, Fig. 9 displays accuracy metrics for the quantitative precipitation estimates for the full year of test data. The results confirm that Hydronn yields more accurate retrievals than HYDRO. Moreover, the Hydronn versions

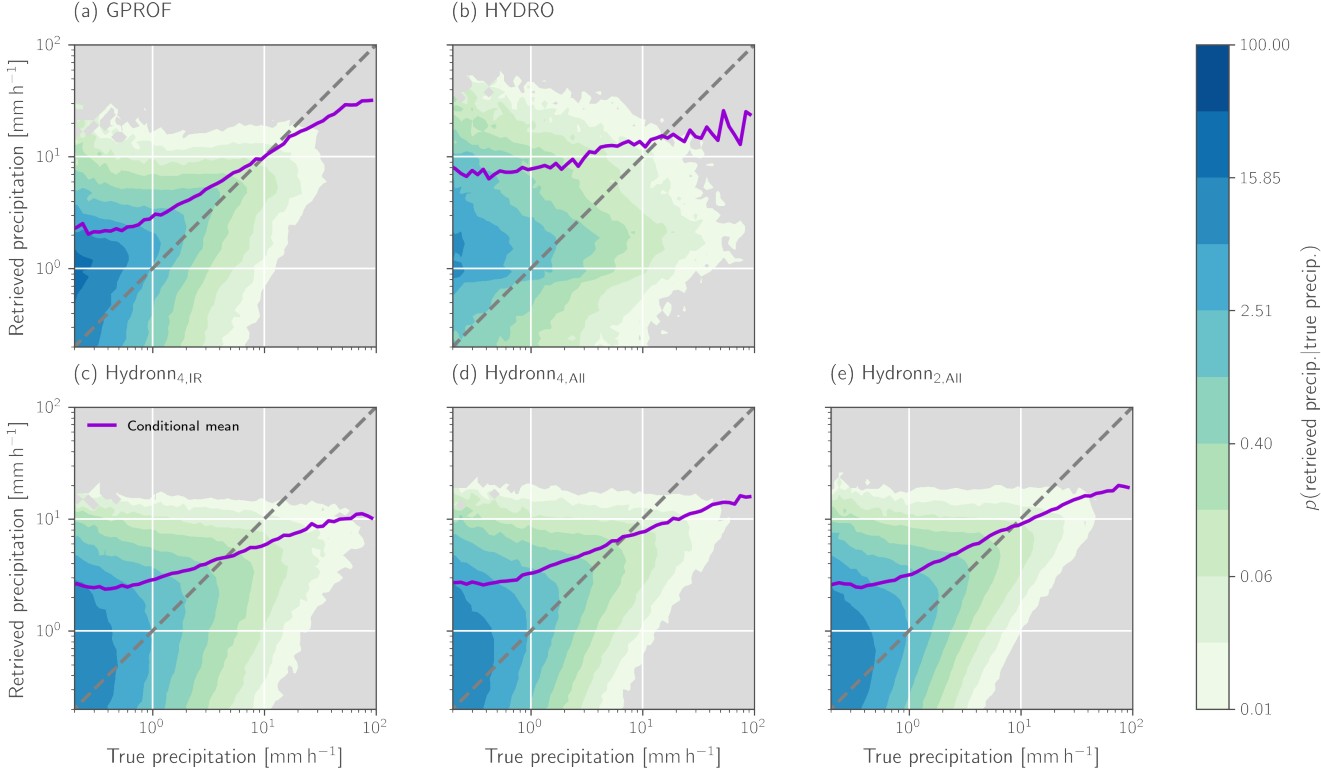

**Figure 8.** PDFs of retrieved precipitation conditioned on the reference precipitation for GPROF GMI, HYDRO and the three Hydronn configurations. The purple line in each panel shows the mean of the retrieved surface precipitation conditioned on the reference precipitation.

that use all ABI channels are close to GPROF in terms of their accuracy. All retrievals exhibit weak seasonal variability across
all metrics but this does not affect their relative performance significantly.

### 4.1.3 Accuracy over northern hemisphere

The neural network models used by Hydronn were trained using only observations over Brazil (R1 in Fig. 2). The results from the previous section indicate that Hydronn achieves significantly higher accuracy than HYDRO and even approaches the accuracy of GPROF GMI when all available ABI channels are used. This, of course, raises the question whether Hydronn still
works outside the region used for its training.

To investigates this, we have evaluated the retrievals using collocations from the 1st, 6th, 11th, 16th, 21st and 26th day of every month of 2020 over the northern hemisphere (marked as R2 in Fig. 2). The results for GPROF and the Hydronn retrievals are displayed in Fig. 10. While the accuracy of GPROF is higher than over Brazil, the accuracy of all Hydronn configurations decreases. However, the decrease remains relatively small compared to the improvements over HYDRO that were observed

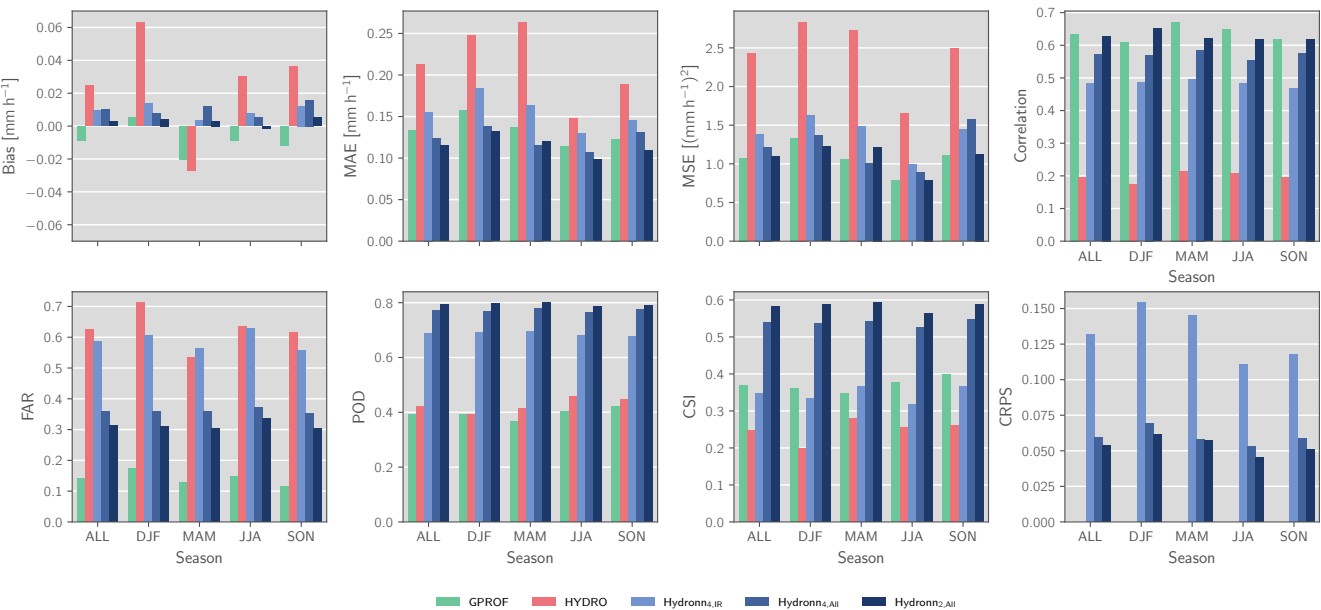

**Figure 9.** Retrieval accuracy with respect to GPM CMB for all overpasses over the training domain in 2020. Each panel shows the average of a metric over the full year as well as its seasonal variability.

over Brazil. This suggests that the neural networks learned robust relationships between the ABI observations and surface precipitation that generalize to observations from outside their training domain.

## 4.2 Validation against rain gauge data

From the estimates of instantaneous precipitation we now turn to accumulated precipitation. Since all of the reference algorithms considered here neglect the probabilistic nature of the precipitation retrieval and provide only a single value as retrieval
result, we first assess the accuracy of the deterministic quantitative precipitation against the gauge measurements. Following this, the probabilistic estimates produced by Hydronn and their potential to improve the characterization of the observed precipitation are assessed.

### 4.2.1 Quantitative precipitation estimates

Accuracy metrics for all retrievals evaluated against the gauge measurements for hourly, daily and monthly precipitation means
are provided in Table 3. In terms of correlations for hourly means, HYDRO yields the worst performance with a correlation of 0.282 for June and 0.134 for December. It is followed by PERSIANN CCS with a correlation of 0.32 and 0.26, respectively. IMERG yields a correlation of 0.53 for June and 0.4 for December. All Hydronn retrievals yield higher correlations with Hydronn$_{4,IR}$ achieving 0.59 in June and 0.4 in December, Hydronn$_{4,All}$ 0.65 and 0.5, and Hydronn$_{2,All}$ 0.67 and 0.59. Hydronn$_{4,IR}$ has higher MAE for hourly accumulations than both HYDRO and IMERG in June and higher MSE than IMERG

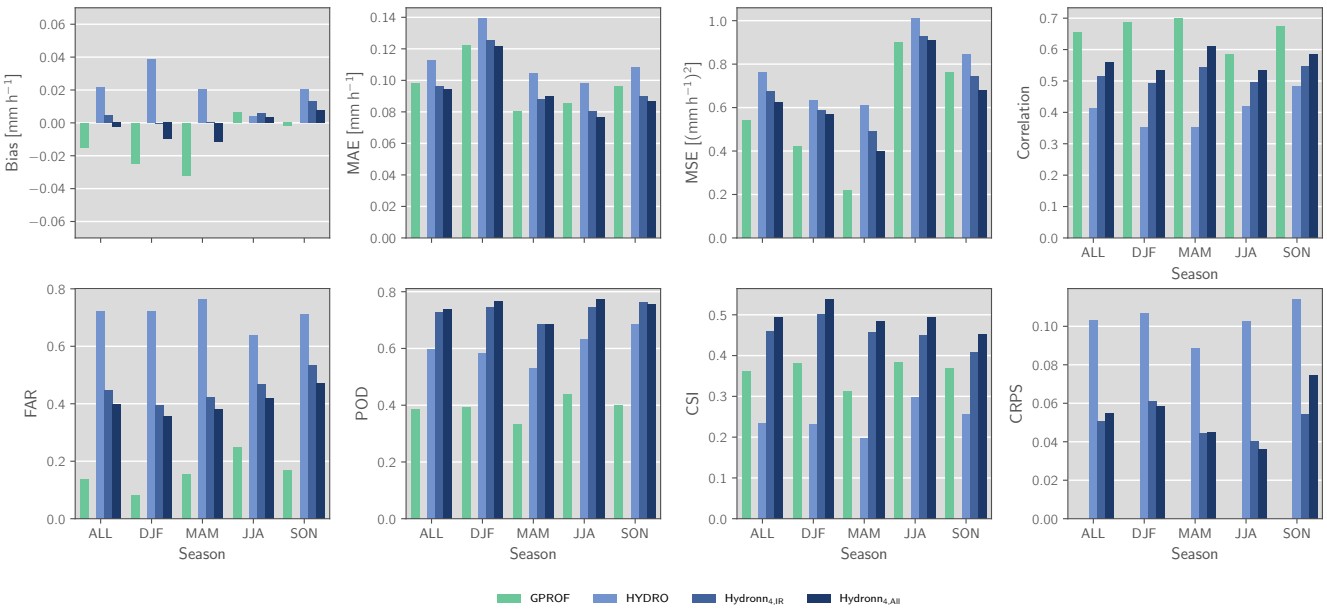

**Figure 10.** Retrieval accuracy with respect to GPM CMB for overpasses on the 1st, 6th, 11th 16th 21st and 26th day of each month of 2020 over the domain R2. Each panel shows the average of a metric over the full year as well as its seasonal variability.

in both June and December. Except for this, however, the Hydronn retrievals generally yield more accurate results in terms of MSE and MAE for hourly accumulations than the other retrievals.

The accuracy of all retrievals improves as the accumulation time increases. For daily means, the ranking of the retrieval algorithms remains mostly the same as for hourly means. This is also the case for monthly means with the exception that the accuracy of IMERG increases and rises to the level of the best Hydronn configuration in terms of MSE and outperforms it in terms of MAE in June. A likely explanation for this is the calibration that is applied to the IMERG Final product, which matches it to monthly gauge measurements.

A graphical analysis of the accuracy of the retrieved daily accumulations is provided in Fig 11. In this representation, the large uncertainties that are present in all retrievals are evident. Nonetheless, the results confirm the general findings from the analysis above. The two conventional IR retrievals, HYDRO and PERSIAN CCS, yield the least accurate results. In particular, both retrievals show a tendency to miss or strongly underestimate accumulations below $40 \, \mathrm{mm \, d^{-1}}$. This tendency is decreased in the IMERG results for accumulations $> 10 \, \mathrm{mm \, h^{-1}}$ but still evident for weaker precipitation. Overall, the Hydronn retrievals achieve higher accuracy for both weak and strong precipitations and the retrieval accuracy increases with the information content of the input. Nonetheless, systematic underestimation of strong rain rates affects all Hydronn retrievals.

The spatial distribution of the biases of the monthly mean precipitation is displayed in Fig. 12. For June, the results from all algorithms exhibit dry biases on the west coast of Brazil and in the south of the country. HYDRO and PERSIANN CCS exibit the strongest biases, while they are weakest in IMERG and $\mathrm{Hydronn_{4,IR}}$.

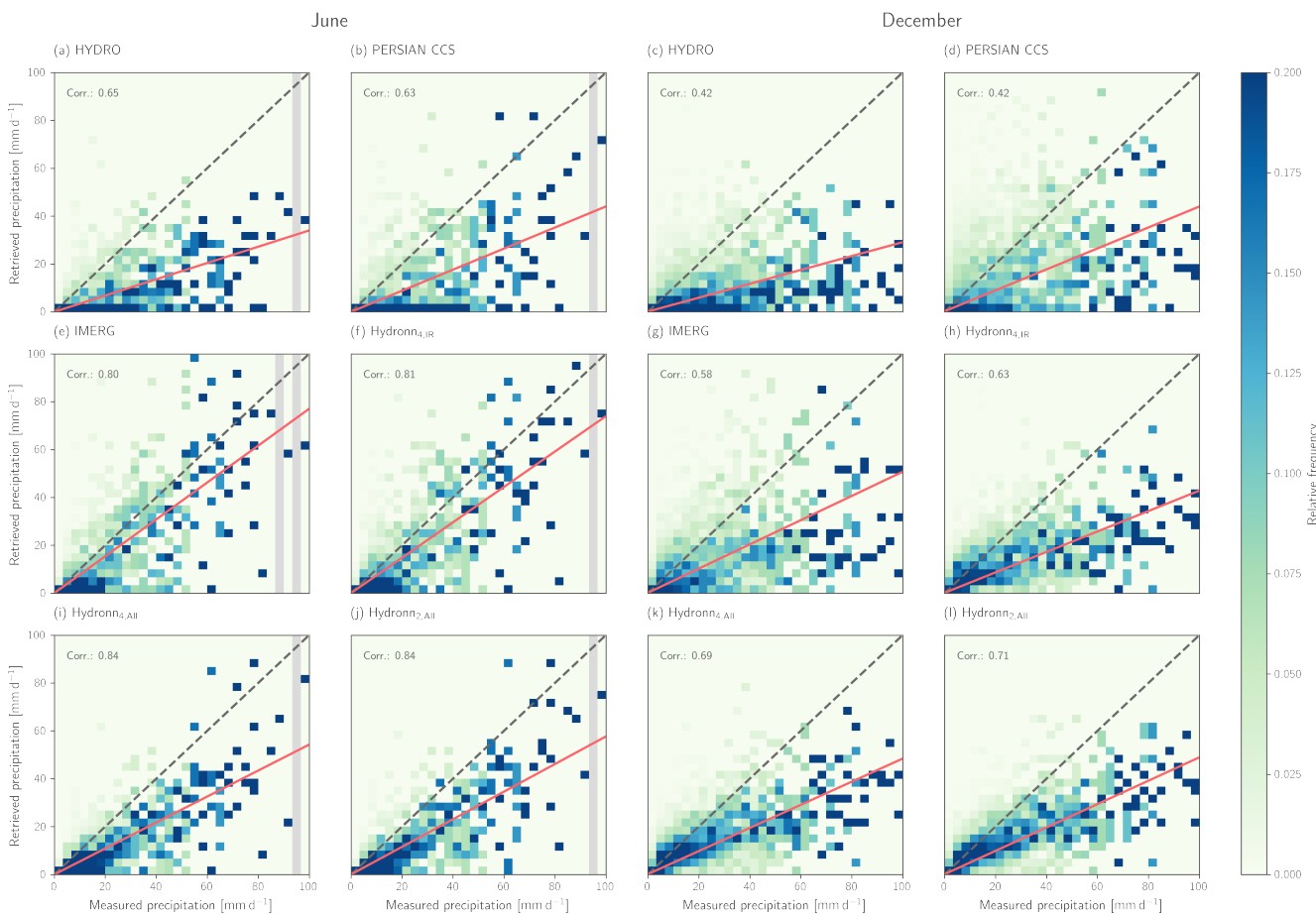

**Figure 11.** Scatter plots of gauge measurements against retrieved daily accumulations for HYDRO, PERSIANN CCS, IMERG and the three Hydronn configurations. The first two columns show the results for June 2020. Columns three and four show the results for December 2020. Frequencies have been normalized column-wise to improve the visibility of heavy precipitation events.

**Table 3.** Accuracy metrics for the retrieved mean precipitation compared to gauge measurements at different time scales. The best values in each column are marked using bold font. Definitions of all metrics are provided in appendix A1.

**June 2020**

| Retrieval | Bias [mm h$^{-1}$] | MAE [mm h$^{-1}$] | | | MSE [(mm h$^{-1}$)$^2$] | | | Correlation | | |
|---|---|---|---|---|---|---|---|---|---|---|
| | | Hourly | Daily | Monthly | Hourly | Daily | Monthly | Hourly | Daily | Monthly |
| HYDRO | -0.055 | 0.106 | 0.079 | 0.06 | 0.611 | 0.077 | 0.01 | 0.28 | 0.65 | 0.72 |
| PERSIANN CCS | -0.035 | 0.115 | 0.085 | 0.053 | 0.671 | 0.077 | 0.008 | 0.32 | 0.63 | 0.72 |
| IMERG | -0.013 | 0.1 | 0.065 | **0.034** | 0.393 | 0.048 | **0.004** | 0.56 | 0.8 | 0.87 |
| Hydronn$_{4,\text{IR}}$ | **−0.002** | 0.108 | 0.07 | 0.036 | 0.404 | 0.045 | **0.004** | 0.59 | 0.81 | 0.85 |
| Hydronn$_{4,\text{All}}$ | -0.034 | 0.084 | 0.059 | 0.043 | 0.361 | 0.043 | 0.005 | 0.65 | 0.84 | 0.88 |
| Hydronn$_{2,\text{All}}$ | -0.031 | **0.084** | **0.058** | 0.04 | **0.345** | **0.04** | **0.004** | **0.67** | **0.84** | **0.89** |

**December 2020**

| Retrieval | Bias [mm h$^{-1}$] | MAE [mm h$^{-1}$] | | | MSE [(mm h$^{-1}$)$^2$] | | | Correlation | | |
|---|---|---|---|---|---|---|---|---|---|---|
| | | Hourly | Daily | Monthly | Hourly | Daily | Monthly | Hourly | Daily | Monthly |
| HYDRO | -0.037 | 0.32 | 0.215 | 0.106 | 3.1 | 0.219 | 0.02 | 0.13 | 0.42 | 0.62 |
| PERSIANN CCS | 0.096 | 0.398 | 0.285 | 0.151 | 3.594 | 0.308 | 0.041 | 0.26 | 0.42 | 0.56 |
| IMERG | 0.014 | 0.285 | 0.196 | 0.08 | 1.9 | 0.18 | 0.014 | 0.38 | 0.573 | 0.73 |
| Hydronn$_{4,\text{IR}}$ | **−0.002** | 0.283 | 0.189 | 0.088 | 2.011 | 0.15 | 0.016 | 0.48 | 0.63 | 0.7 |
| Hydronn$_{4,\text{All}}$ | −0.006 | 0.235 | 0.159 | 0.076 | 1.797 | 0.128 | 0.013 | 0.56 | 0.69 | 0.75 |
| Hydronn$_{2,\text{All}}$ | 0.011 | **0.226** | **0.153** | **0.074** | **1.704** | **0.121** | **0.013** | **0.59** | **0.71** | **0.76** |

The results for December are less homogeneous between algorithms. The strongest biases are observed in the PERSIANN CCS results, which strongly overestimate precipitation in central and northern Brazil. HYDRO and Hydronn$_{4,\text{IR}}$, as well as to a lesser extent PERSIANN CCS, Hydronn$_{4,\text{All}}$ and Hydronn$_{2,\text{All}}$, exhibit a systematic dry bias in southern Brazil. Overall, the biases of IMERG are smallest in magnitude and exhibit the least extent of spatial correlation. However, the differences between IMERG and the best Hydronn configuration, Hydronn$_{2,\text{All}}$, are small.

Finally, we consider the retrieved daily cycles of precipitation, which are displayed in Fig. 13. From the reference retrievals, IMERG yields the best agreement with the gauge measurements in both June and December. In June, the daily cycle is mostly flat with a weak peak around 14 h. IMERG reproduces this behavior well but exhibits a weak peak that is delayed by about two hours. In addition to exhibiting larger biases, the daily cycles derived from HYDRO and PERSIANN CCS show an increase of precipitation towards the afternoon, which is in disagreement with the gauge measurements. Hydronn$_{4,\text{IR}}$ and Hydronn$_{4,\text{All}}$ exhibit biases comparable to those of HYDRO and PERSIANN CCS but remain flat throughout the day. The Hydronn$_{4,\text{IR}}$ results track the gauge measurements almost exactly although the afternoon peak is delayed by about about an hour.

Both Imerg and HYDRO yield relatively good agreement with the gauge measurements for December. IMERG is slightly closer to the gauge measurements during morning and early afternoon but overestimates precipitation in the afternoon and

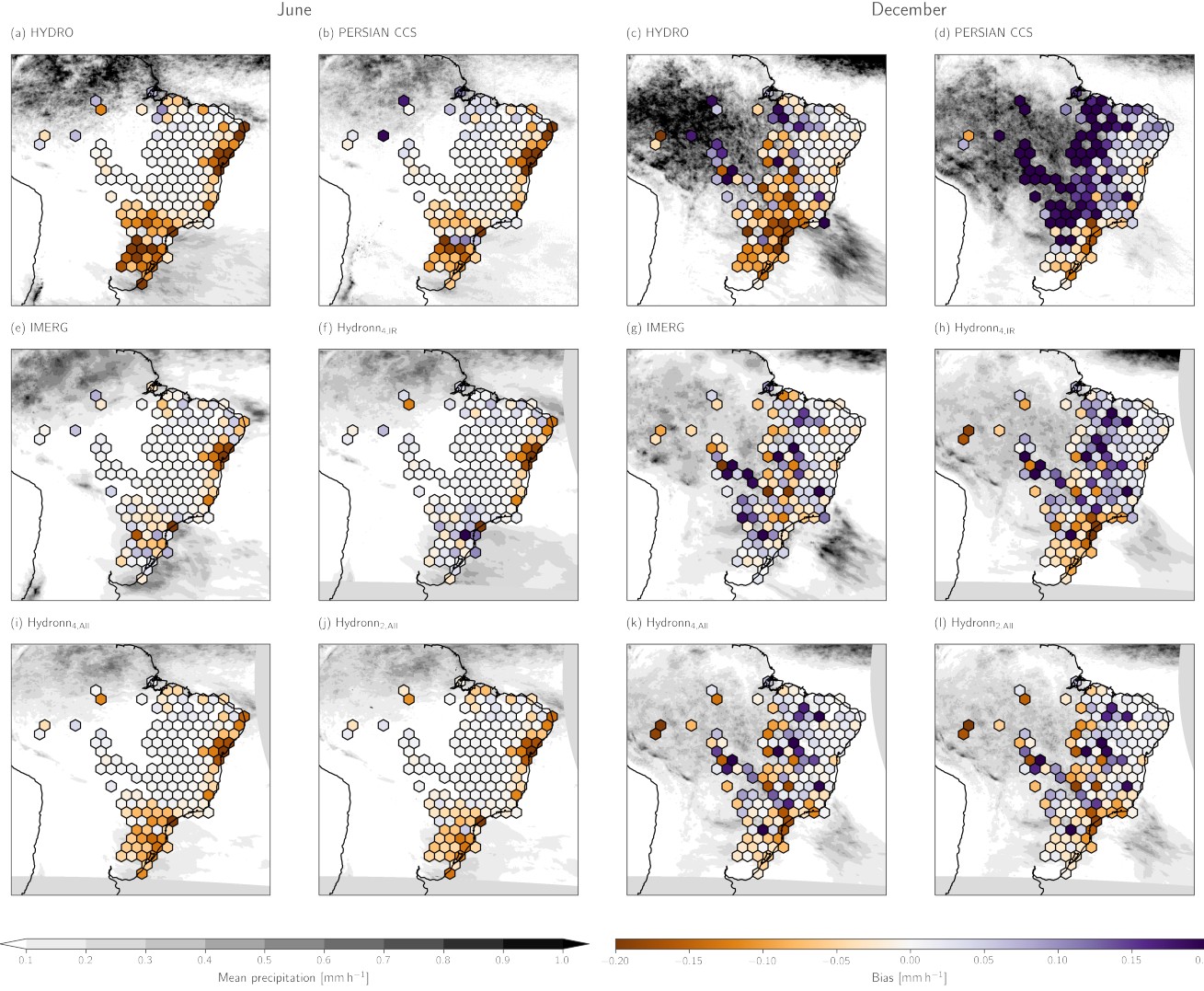

**Figure 12.** Retrieved mean precipitation during June and December 2020. The first two columns show the results for June. Columns three and four show the results for December. Shading in the background of each panel shows the spatial distribution of the mean precipitation of the corresponding retrieval. Colored hexagons show the spatial distributions of the retrieval biases with respect to the gauge measurements.

evening. HYDRO slightly underestimates precipitation during the first half of the day but its afternoon peak, despite being close in magnitude to that of the gauge measurements, is delayed by about three hours. PERSIANN CCS shows good agreement with the gauge measurements in the first half of the day but strongly overestimates the afternoon peak. All Hydronn configurations yield good agreement with the gauge measurements.

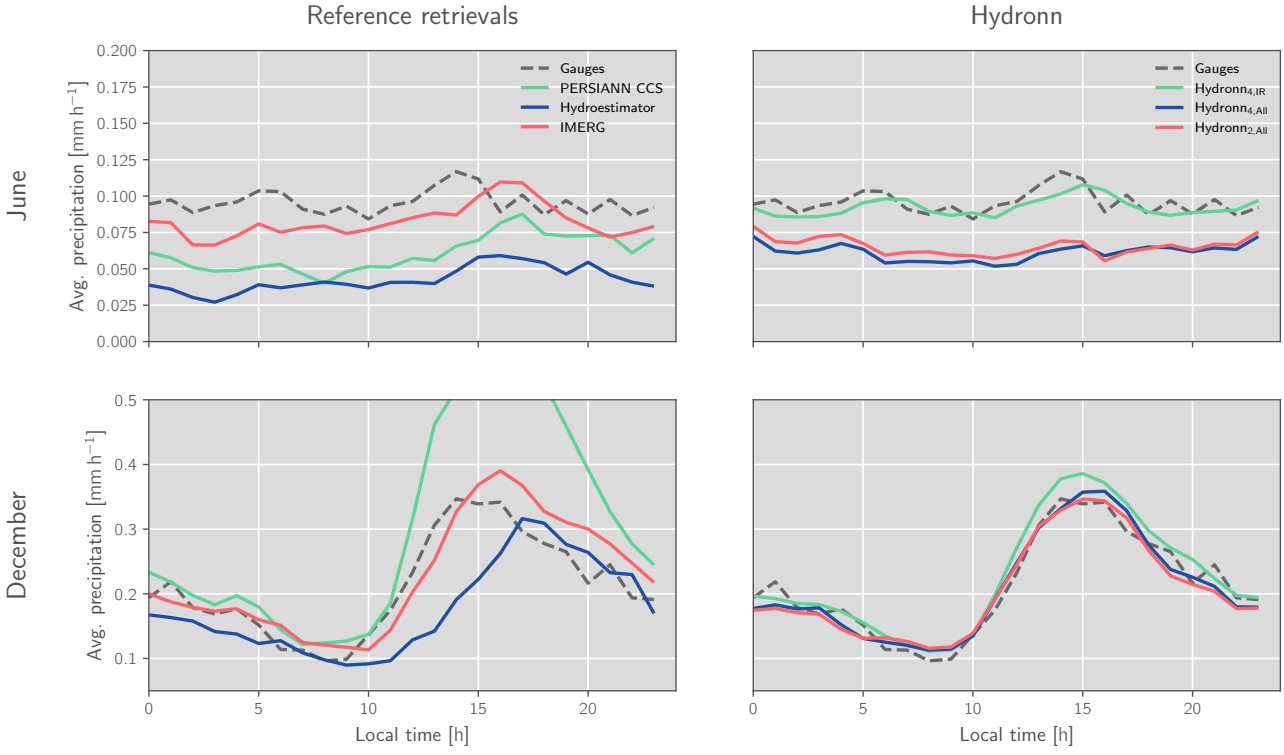

**Figure 13.** Measured and retrieved daily cycles of precipitation. The first column displays the daily cycles retrieved by the three reference retrievals (solid lines) and the gauge measurements (dashed line) for reference. The second column corresponding diurnal cycles for the three Hydronn configurations. The first row shows the results for June 2020 and the second results for December 2020.

### 435    4.2.2    Probabilistic estimates

We now proceed to evaluate the probabilistic precipitation estimates that are produced by Hydronn. As explained in Sec. 3.5, two probabilistic estimates of the hourly precipitation rates were produced. The first one, (dep.), assumes strong dependence of retrieval errors for consecutive observations, while the second one, (indep.), assumes independent errors. In addition to that, a corrected distribution has been calculated for each of these predictions using the correction factors described in Sec. 3.6. 440   This yields four probabilistic predictions for each Hydronn configuration. These predictions will be referred to with the configuration name and the qualifiers (dep.), (dep., corr.) for the predictions derived assuming dependent uncertaintes with and

without correction, respectively, and (indep.), (indep. corr.) for the corresponding predictions derived using the independence assumption.

We first consider the distribution of precipitation rates as measured by gauges and the retrieval algorithms, which is shown in Fig. 14. For June, IMERG precipitation accumulations agree very well with the gauge measurements but fail to capture the heavy precipitation exceeding $20 \, \mathrm{mm \, h^{-1}}$. HYDRO and PERSIANN CCS underestimate the occurrence of light precipitation $< 1 \, \mathrm{mm \, h^{-1}}$. While HYDRO tends to underestimate moderate and heavy precipitation, the tail of the precipitation distribution is well represented by PERSIANN CCS.

The distribution of the retrieved mean precipitation for the Hydronn retrievals is close to the gauge measurements up until precipitation of around $8 \, \mathrm{mm \, h^{-1}}$ but underestimates the frequency of heavier precipitation. When instead samples from the posterior distribution are considered the representation of heavy precipitation is improved. However, for (indep.) the frequency of heavy precipitation events is still underestimated, which is slightly improved by the correction. Conversely, the correction of the (indep.) results of $\mathrm{Hydronn_{4, IR}}$ leads to an overestimation of heavy precipitation. The correction slightly improves the representation of very light precipitation for the (dep.) results of $\mathrm{Hydronn_{4, All}}$ and $\mathrm{Hydronn_{2, All}}$.

For December, IMERG overestimates the frequency of light precipitation while they are well represented by HYDRO and PERSIANN CCS. All retrievals overestimate the frequency moderate precipitation. HYDRO and IMERG underestimate the frequency of heavy precipitation, while PERSIANN CCS captures the tail of the distribution well. For the Hydronn retrievals the distribution of the retrieved mean precipitation is again similar to the distribution of IMERG except with a stronger tendency to underestimate the frequency of heavy precipitation. The sampling again recovers the heavy precipitation events. The correction slightly improves the frequencies of low and heavy precipitation but the effect remains small.

The missing heavy precipitation events in the distribution of the mean retrieved by Hydronn retrievals should thus be understood as an effect of the uncertainties in the retrieval results. Because the information content of the VIS/IR observations is insufficient to accurately determine the strength of heavy precipitation events, the retrieved mean will always underestimate the heaviest of those events. When instead samples from the posterior distribution are considered, the extreme values of the distribution are correctly reproduced.

The assumption used to calculate the posterior distribution of hourly accumulations has a clear impact on the distribution of the retrieved precipitation rates. Assuming independent retrieval errors leads to an overestimation of the frequency of light precipitation at the expense of heavy precipitation, while assuming dependent errors has the opposite effect. It is interesting to note that for June the assumption of independent errors yields better agreement with the gauge measurements while for December it is the latter.

In addition to sampling from the posterior distribution, the retrieved quantiles can be used to derive confidence intervals for the predicted precipitation. The reliability of the confidence interface for December 2020 is assessed in Fig. 15 using calibration curves. The corresponding results for June can be found in Fig. A1 in the appendix. We discuss only the results from December here because the results from June are practically the same.

For the assumption of dependent retrieval errors, the calibration curve tends to lie above the diagonal, which signifies that the true precipitation values fall into the predicted interval more often than expected. The retrieved confidence intervals thus

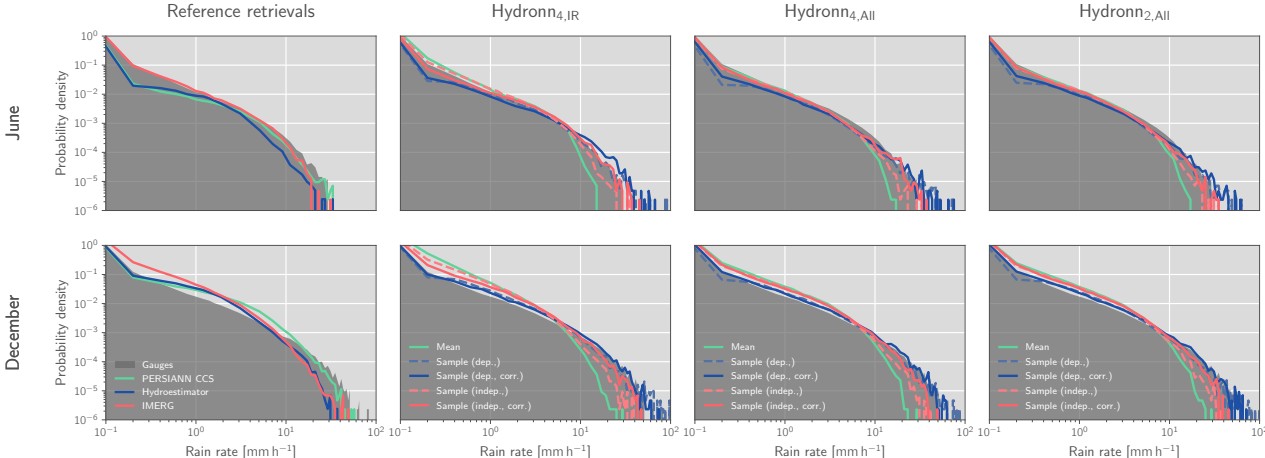

**Figure 14.** Distributions of measured and retrieved rain rates. The dark grey, filled curve in the background of each panel shows the PDF of the gauge measurements. Colored lines drawn on top show the corresponding PDFs of the retrieved precipitation. The distributions for the reference algorithms are shown in Panel (a). Panel (b), (c) and (d) show the distribution of mean values and random samples from the retrieval posterior.

overestimate the retrieval uncertainty. The opposite effect is observed for the assumption of independent errors. Applying the a priori correction improves the calibration for both assumptions.

The results presented in Fig. 15 use the modified gauge measurements described in Sec. 3.6 to which small random noise has
been added to non-zero measurements and zeros were replaced with small random values. This was required because quantiles are ill-defined when the CDF of a quantity is discontinuous. So while the results above show that the predicted uncertainties from the Hydronn retrieval are well calibrated, they would not be when compared against the raw gauge measurements. However, since the modifications to the measured precipitation are well within their uncertainty this still shows that the predicted confidence intervals are meaningful.

The retrieved quantiles can also be used to estimate the probability that an observed pixel exceeds certain precipitation thresholds. This has been used to calculate probabilities of hourly accumulations exceeding 5 and 20 $\mathrm{mm\,h^{-1}}$, which correspond roughly to the 99 and 99.9 percentiles of the distribution of gauge measurements. The ability of the retrievals to detect high-impact precipitation events in December 2020 is assessed using precision-recall (PR) curves (see A1 for definition) in Fig. 16. The corresponding results for June can be found in Fig. A2 in the appendix. For the non-probabilistic retrievals the
curves were generated using the predicted precipitation and classifying all pixels above a varying threshold as exceeding the sought-after precipitation rate. The corresponding curves for the Hydronn retrievals were obtained by varying the probability threshold above which a pixel is classified as an high-impact event.

For the detection of events exceeding 5 $\mathrm{mm\,h^{-1}}$, HYDRO exhibits the least skill, followed by PERSIANN CCS. Also here, IMERG yields better results than the two conventional IR retrievals. The detection skill of Hydronn$_{4,\mathrm{IR}}$ is noticeably better
than that of IMERG, while the two other configurations further improve the detection performance. For events exceeding

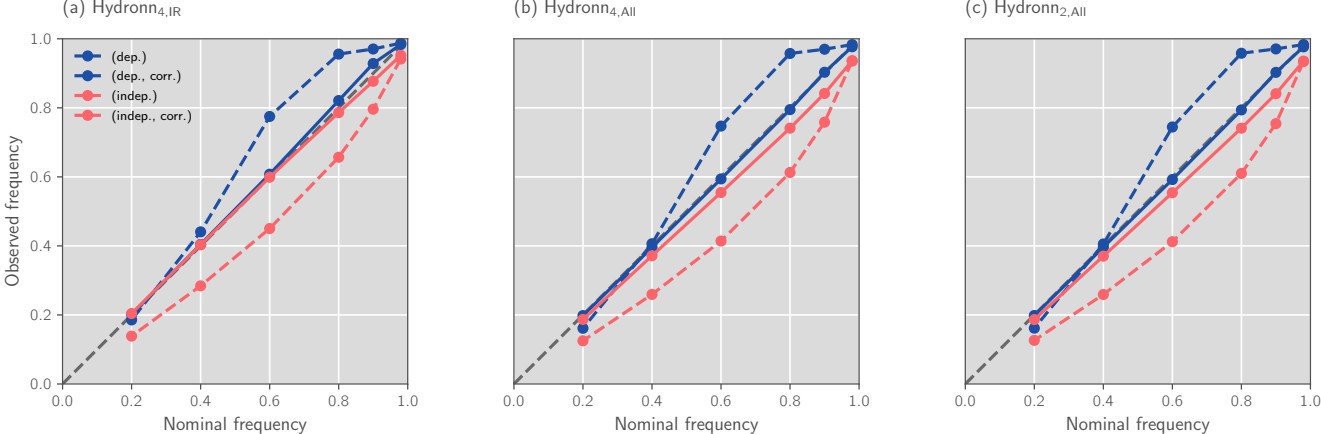

**Figure 15.** Calibration of the confidence intervals derived using the quantiles of the retrieval posterior distribution predicted by each Hydronn configuration for December 2020. Grey, dashed line in the background shows the expected results for perfectly calibrated results.

$20 \, \mathrm{mm} \, \mathrm{h}^{-1}$, all retrievals yield worse detection accuracy than at $5 \, \mathrm{mm} \, \mathrm{h}^{-1}$. Also here HYDRO exhibits the least skill, followed by PERSIANN CCS and IMERG, which yield very similar results. All Hydronn configurations outperform the reference retrievals.

Interestingly, the assumption used to accumulate the uncertainties as well as the a priori correction do not any noticeable effect on the detection skill. The reason for this is likely that the PR curves are invariant to any transformation that preserves the ranking of the strength of the precipitation events. This means that, although the two assumptions may lead to different assigned probabilities, they both contain the same information on the strength of the observed precipitation event.

Since the Hydronn retrievals can be used to derive a probability of an observation exceeding a given precipitation threshold, a relevant question is how accurate these probabilities are. The calibration of the detection probabilities of events exceeding $5 \, \mathrm{mm} \, \mathrm{h}^{-1}$ is displayed in Fig. 17 for December 2020 and Fig. A3 for June 2020 in in the appendix. For December, the predictions derived assuming temporally dependent uncertainties are fairly well calibrated but larger deviations from the diagonal are observed for the probabilities derived assuming independent uncertainties. For June, (dep.) yields the best calibrated results for Hydronn$_{4,\mathrm{IR}}$, while for Hydronn$_{4,\mathrm{All}}$ and Hydronn$_{2,\mathrm{All}}$ (indep.) yields the best calibration. However, the differences are relatively small and both assumptions yield reasonably well calibrated probabilities. The corrections only have a minor effect on the calibration and don't lead to any consistent improvements.

### 4.3 Case study

As final part of this evaluation, a case of heavy precipitation in the city of Duque de Caxias in the State of Rio de Janeiro is considered, which occurred between the 22 and 24 December 2020 and lead to flooding (Fohla De S. Paulo, 2020). About $250 \, \mathrm{mm}$ of accumulated precipitation was measured by the rain gauge in the neighborhood of Xerém over the period of two days.

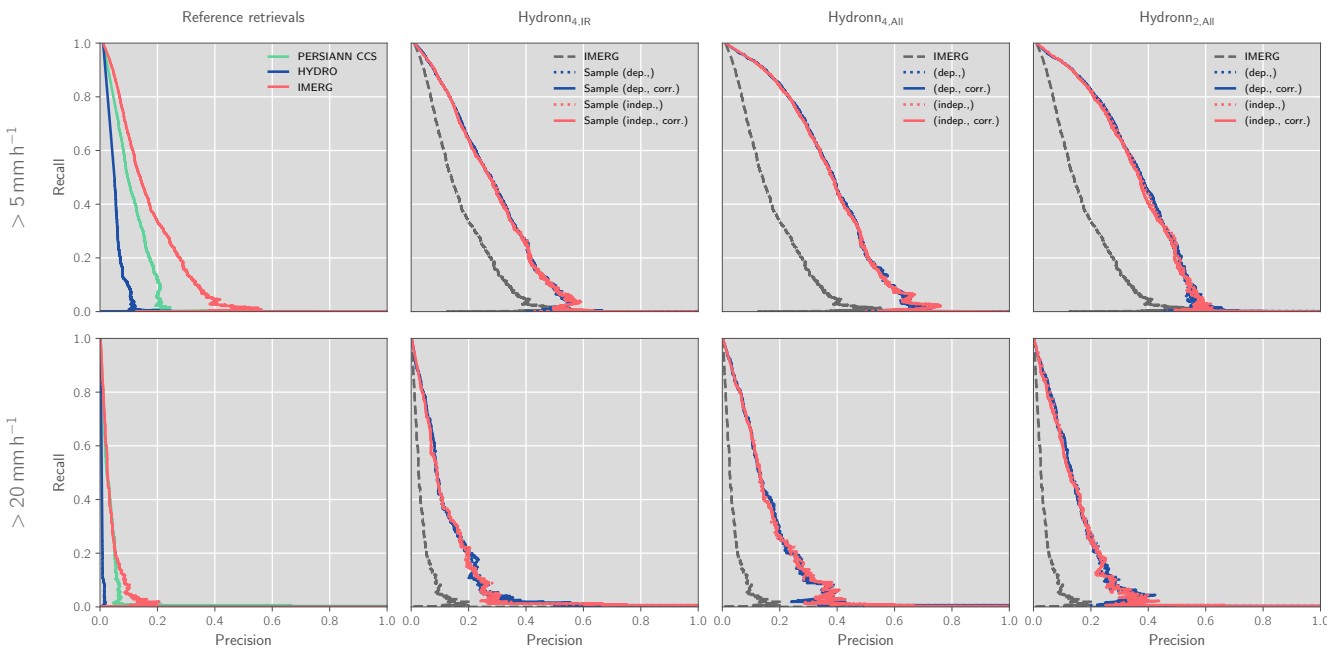

**Figure 16.** Precision-recall curves for the detection of precipitation events with precipitation rates larger than $5$ mm h$^{-1}$ (first row) and $20$ mm h$^{-1}$ (second row) in December 2020. Columns show the results for the reference retrievals as well as for each of the Hydronn configurations.

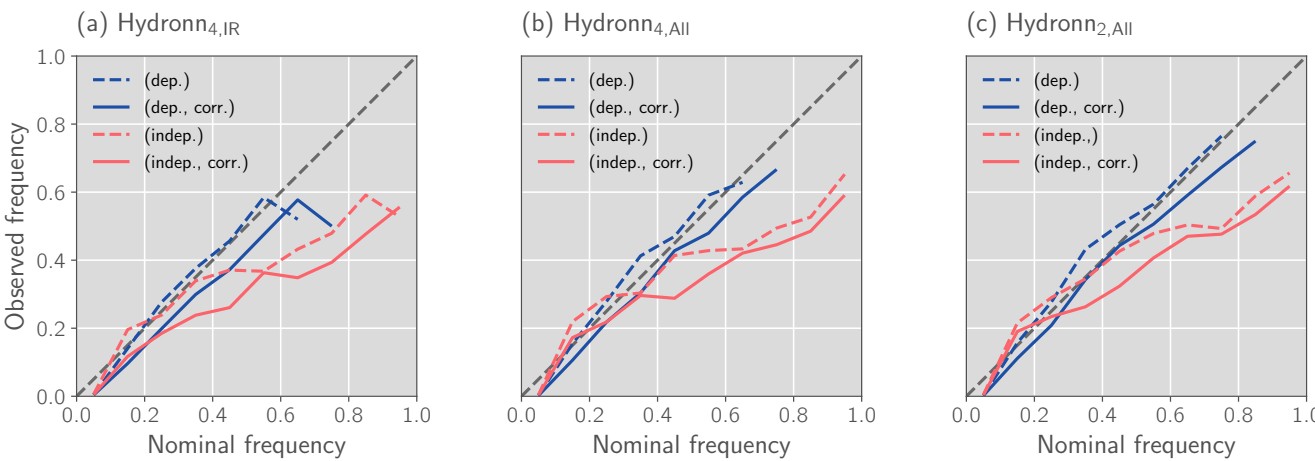

**Figure 17.** Calibration of the probabilistic precipitation event detection for precipitation exceeding $5$ mm h$^{-1}$

Gauge measurements and retrieved e precipitation accumulations in the area surrounding the gauge in Xerém during the two days are shown in Fig. 18. The gauge measurements show localized occurrence of heavy precipitation in a number of gauges to the north and east of Xerém and light precipitation along the coast towards the south and east.

The HYDRO retrievals are very high in the north-east of the region but miss the precipitation that fell around Xerém. PERSIANN CCS misses nearly all of the precipitation that fell during the two days. In contrast, IMERG and the Hydronn retrieve more precipitation in the area around Xerém. In particular, Hydronn$_{4,\text{All}}$ and Hydronn$_{2,\text{All}}$ both capture the precipitation in the vicinity of Xerém well.

It is notable that the Hydronn$_{4,\text{All}}$ and Hydronn$_{2,\text{All}}$ retrievals exhibit considerably finer structures in their results than IMERG or the other retrievals. Moreover, these structures agree well with the gauge measurements. This suggests that the high resolution of the VIS/IR observations allows the retrieval to better resolve small-scale precipitation events.

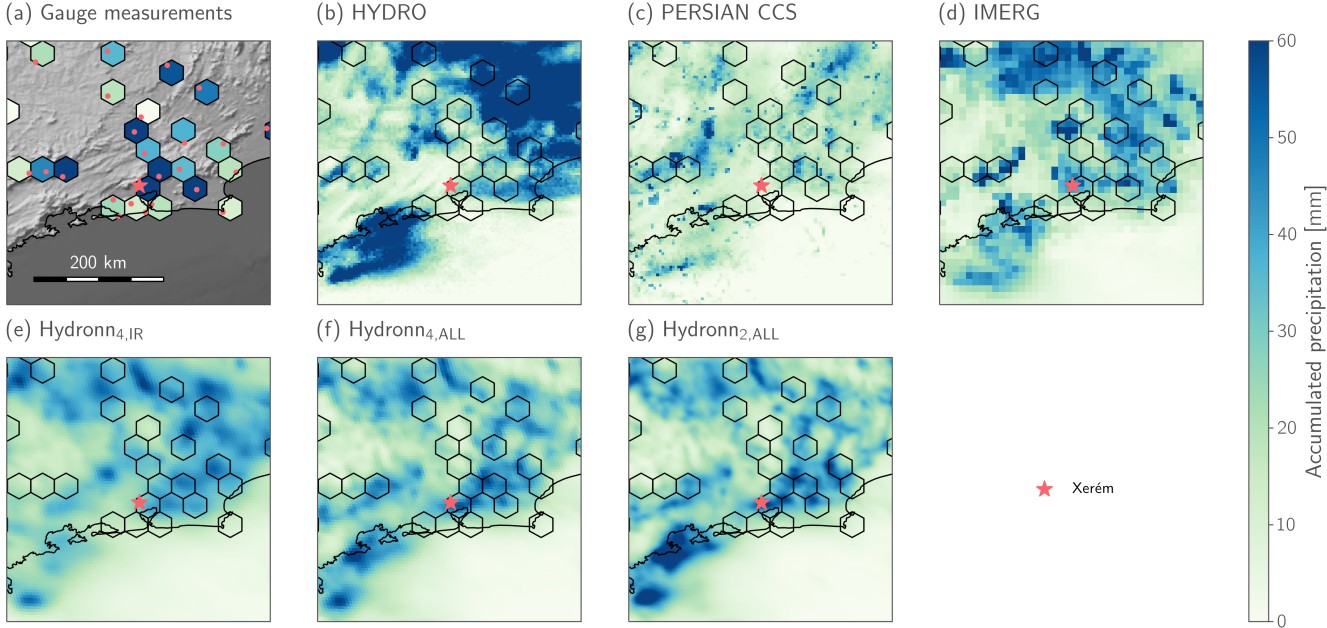

**Figure 18.** Retrieved precipitation accumulations for an extreme precipitation event in the city of Duque de Caxias in the state of Rio de Janeiro. Panel (a) shows gauge-measured precipitation accumulations using colored hexagons. Locations of the gauges are marked using red points. The red star marks the location of the Xerém neighborhood of Duque de Caxias in which the gauge station closest to the reported flooding is located. The remaining panels show the retrieved precipitation accumulations for the tested retrieval algorithms.

The rain rates at the gauge station in Xerém (location marked by red star in Fig. 18) are displayed in Fig. 19. The plots show the hourly precipitation rates retrieved by the reference retrievals as well as the mean and posterior distribution for all Hydronn retrievals. Only results obtained with the assumption of dependent retrieval uncertainties are shown. The precipitation measured at the rain gauge far exceeds the precipitation measured by any of the reference retrievals or the retrieved mean of the Hydronn retrievals. The Hydronn retrievals predict elevated uncertainties for the period during which the strongest precipitation

is observed. However, the precipitation peaks still exceed the 99th percentile. Two factors may explain that more than the expected 1 % of gauge measurements lie outside the predicted uncertainty range. Firstly, the observations considered here are not randomly sampled but correspond to an event that is known to be extreme. Secondly, as stated in the article in Fohla De S. Paulo (2020), heavy precipitation events are common in this region. This may indicate that regional factors act to intensify the precipitation, which is unlikely to be captured in the training data of the retrieval.

Nonetheless, an encouraging results is that the predicted value of the 99th percentile increases with the information content in the retrieval input. This indicates that the neural network can leverage the additional information to better represent the uncertainty in the retrieval.

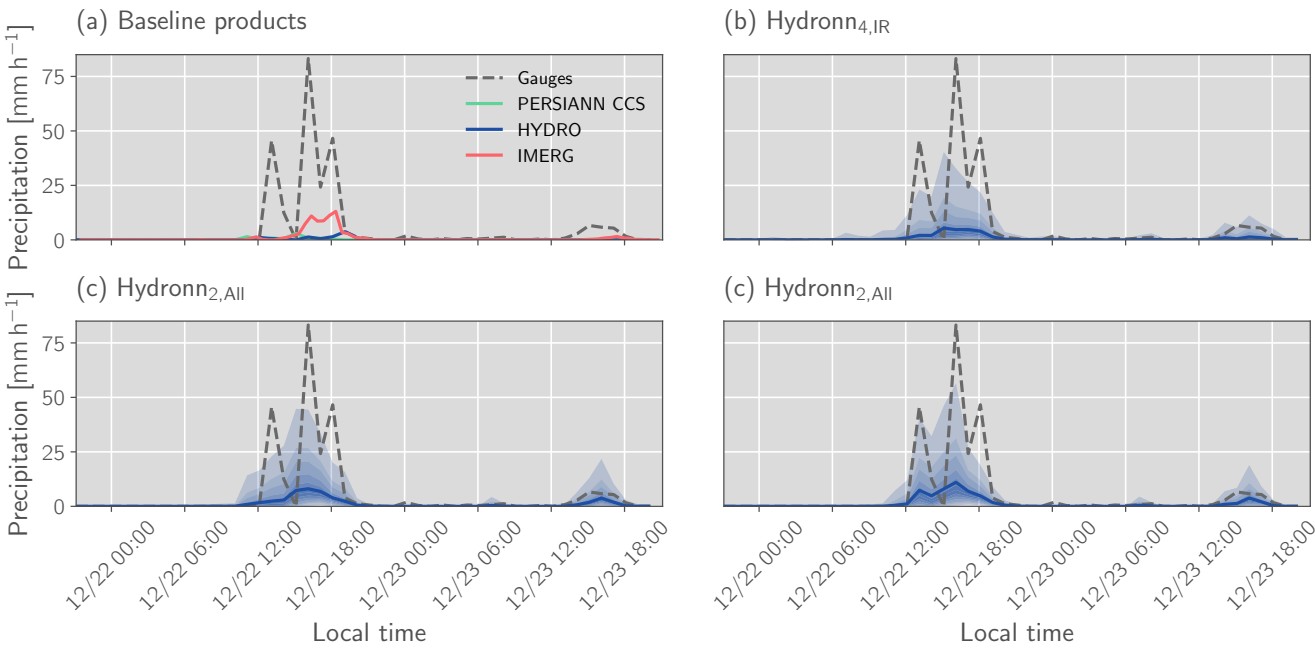

**Figure 19.** Retrieved precipitation for an extreme precipitation even that occurred between December 22, 2020 and December 24, 2020, in the city of Duque de Caxias in the state of Rio de Janeiro. Grey, dashed lines show the precipitation at the gauge station in Xerém. Solid lines show the retrieved mean precipitation for each retrieval algorithm. The shading shows filled contours of the posterior CDF at values $[0.01, 0.1, 0.2, \ldots, 0.8, 0.9, 0.99]$.

## 5 Discussion

The study presented Hydronn, a neural-network-based precipitation retrieval for Brazil, which has been trained using combined radar and radiometer measurements from the GPM core observatory. Using all ABI channels at their native resolution, the retrieval yields instantaneous precipitation estimates that are close in accuracy to those of GPROF GMI. The derived accumulations compare favorably against the currently operational precipitation retrieval, HYDRO, as well as the PERSIANN

CCS product. The configurations that use all ABI channels yield more accurate precipitation accumulations than the IMERG
Final product across most considered metrics.

## 5.1 Information content of VIS/IR observations

The three tested retrieval configurations use input observation of increasing information content. The Hydronn$_{4,IR}$ configuration uses only a single IR channel at a resolution of $4$ km while Hydronn$_{4,All}$ uses all available bands of the ABI. The best performing retrieval, Hydronn$_{2,All}$, combines the observations from all channels of the GOES ABI at their native resolutions.
Clear increases in retrieval performance are observed when all ABI bands are incorporated into the retrieval and an additional, albeit smaller, improvement is achieved when all channels are ingested at their native resolutions. This demonstrates the ability of the neural-network-based retrieval to efficiently combine observations across channels and different spatial scales. The fact that HYDRO and Hydronn$_{4,IR}$ use the same observations as retrieval input demonstrates the significant improvement that neural-network based retrievals can achieve compared to conventional methods. This is in good agreement with the findings
from Sadeghi et al. (2019), who also report improvements when comparing PERSIANN CCS to a CNN-based retrieval.

The retrieval accuracy of Hydronn in its most advanced configuration is comparable to that of GPROF GMI at the $5$ km resolution considered here. Certainly, it must be taken into account that the Hydronn retrievals are evaluated against the data they were trained to reproduce, which will tend to overestimate their accuracy with respect to independent measurements. However, this is also the case for GPROF GMI, whose retrieval database is to large extent built up of collocations with GPM
CMB. This result is notable because VIS/IR observations from geostationary satellites are typically understood to merely 'augment' (Kidd et al., 2021) the more capable PMW sensors. While the case study from Sec. 4.1.1 indicates that the VIS/IR observation are still inferior in terms of their ability to quantify the total amount of precipitation, the structure of the MCS is truthfully represented. Leveraging the high temporal resolution of the GOES ABI observations, the retrieval easily outperforms all other tested algorithms in terms of hourly and daily accumulations with a considerable margin. This suggests that space-
borne precipitation measurements may benefit significantly by making better use of available observations from geostationary satellites.

## 5.2 Probabilistic precipitation retrievals

A novel aspect of the proposed precipitation retrievals is their ability to provide probabilistic precipitation estimates. In this study we have demonstrated multiple ways in how this may improve the utility of the retrieval results:

1. The results in Fig. 14 show that samples from the retrieval posterior reproduce the gauge-measured distribution of rain rates more accurately than the retrieved mean. The deviations of the distribution of the posterior mean from the gauge measurements should thus be understood as a consequence of the statistical properties of this estimator instead of a retrieval deficiency. The random samples may be useful for applications that are sensitive to heavy precipitation rates, such as runoff modeling or climatological studies. To illustrate this, Fig. 20 shows scatter plots of the 99th percentile
of the monthly distribution of hourly accumulations at each gauge station and the 99th percentile of the corresponding

retrievals for June and December 2020. HYDRO and PERSIANN CCS yield accuracy similar to IMERG in this analysis, despite IMERG having higher accuracy for all other metrics considered in this study. Both HYDRO and PERSIANN CCS were developed with a focus on convective precipitation. The regression relations underlying both retrievals were developed from summer precipitation in the US and enforce monotonically decreasing relationship (Hong et al., 2004; Vicente et al., 1998) between brightness temperatures and precipitation rates. This may explain why they succeed in representing heavy, convective precipitation events but fail to represent more general conditions. By explicitly resolving the probabilistic nature of the precipitation retrieval, HYDRONN can provide both climatologically accurate accumulations (see Table 3) and improved representation of heavy precipitation.

It should be noted, however, that these random samples do not take into account spatial correlations. To what extent this may negatively impact applications of the retrieval results remains to be investigated.

2. The retrieved quantiles allow the derivation of confidence intervals to quantify retrieval uncertainty. By correcting for the difference in a priori distributions as well as the degeneracy of quantiles due to discontinuities in the CDF of gauge measurements, we were able to show that the retrieval uncertainties are well calibrated even against gauge measurements (Fig. 15). Due to the large uncertainties that are inherent to precipitation retrieval from VIS/IR observations (Fig. 8, 11), quantifying these uncertainties has the potential to increases the trustworthiness of the predictions.

3. We have shown that the retrieved quantiles can be used to detect heavy precipitation events (Fig. 16, Table 3). Here all Hydronn retrievals perform better than IMERG although they are based on observations with a significantly lower information content. This clearly shows the benefits of quantifying retrieval uncertainties. Moreover, we were able to show that the probabilistic detection of these events is fairly well calibrated (Fig.17, Fig.17).

Finally, we have also investigated how uncertainties from instantaneous precipitation retrievals can be propagated to the full hour. The two approaches that we have tested correspond to assuming temporally independent and temporally dependent retrieval uncertainties. Our results indicate that the assumption of dependent uncertainties overestimates the actual retrieval uncertainty, whereas assuming independent uncertainties underestimates actual uncertainties (Fig A1, Fig. 15). In lack of a better method to infer the distribution of hourly accumulations, our results indicate that assuming dependent uncertainties yields slightly more reliable results (Fig 15, Fig. A1, Fig. 17, Fig. A3). Moreover, it is encouraging that the way the accumulations are calculated does not affect the overall detection skill for heavy precipitation events (Fig. 16, Fig. A2).

### 5.3 Utility of a priori corrections

We have proposed a method to correct for variations in the distribution of precipitation in the training data relative to comparable ground validation data. The most distinct effect of the a priori correction was observed when the predicted confidence intervals were evaluated against gauge data (Fig. 15, Fig. A1). This allowed us to show that the Hydronn retrievals can provide well-calibrated uncertainty estimates for their predictions when the differences between the a priori distributions of the training data and the gauge measurements are taken into account.

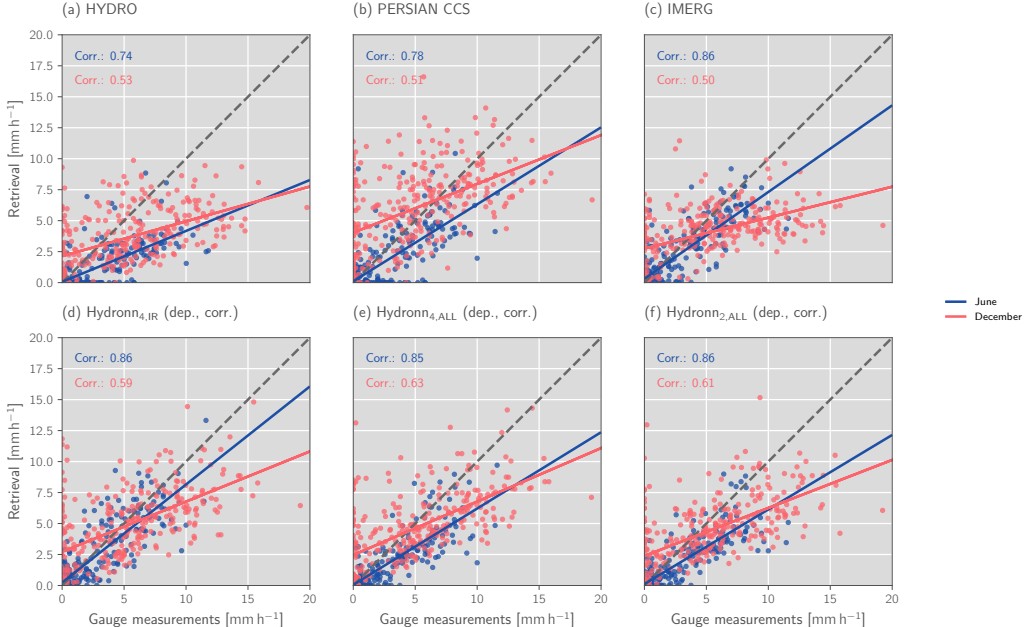

**Figure 20.** Scatter plots of the 99th percentile of the monthly distribution of hourly precipitation accumulations of each gauge station for June (blue) and December (red) plotted against the 99th percentile of the corresponding retrieved distribution of precipitation accumulations. The results of the Hydronn retrievals use samples from the posterior distribution of hourly accumulations obtained assuming dependent retrieval errors instead of the posterior mean.

However, the correction had only minor effects on the observed distribution of precipitation and did not improve the calibration of the detection of heavy precipitation events. We suspect the reason for this to be that the correction mostly affects the probabilities of light precipitation, which because of their frequency have a strong effect on the calibration of the confidence intervals. However, the statistics used to derive the correction may not be precise enough to correct for the differences of the much rarer heavy precipitation events. Whether more specialized corrections that take into account seasonal variability can help with the detection of extreme precipitation remains to be investigated.

## 6 Conclusions

Hydronn, the presented neural-network-based precipitation retrieval, improves real time precipitation estimates over Brazil. Its performance is superior to both the currently operational algorithm and the much more sophisticated, global IMERG Final product, which combines observations from VIS/IR and PMW sensors as well as global gauge measurements.

Our results demonstrate the potential of region-specific retrieval algorithms, which exploit the full potential of locally available satellite observations. This is made possible by the availability of accurate surface precipitation retrievals from the GPM

core observatory, which were used to derived the training data for the retrieval. Since this data is available globally between $-65$ and $65\,°\mathrm{N}$, the approach can potentially be applied to many other regions around the world.

We have shown that our retrievals work reasonably well even outside their training domain over Brazil. This indicates that, not only the regional training data, but also the ability of deep CNNs to leverage previously discarded, spectral and structural information from the satellite imagery contribute to the good performance of the Hydronn retrievals. Moreover, we have shown how a probabilistic regression approach can be used to perform VIS/IR precipitation retrievals using a Bayesian framework and that the probabilistic predictions improve the characterization of the observed precipitation.

Finally, the fact that our relatively simple retrieval outperforms state-of-the-art precipitation products despite being solely based on VIS/IR observations, shows the potential of deep learning for quantitative precipitation estimation. The ability of the neural network retrieval to leverage information from all channels of the ABI at their native resolutions shows the strength of the end-to-end approach to retrieval design. This suggests that there is considerable room to improve space-borne precipitation estimates by making better use of currently available satellite imagery.

*Code availability.* The code to generate the training data, train the machine learning models, run the retrievals and analyze the results is available from a public repository (Pfreundschuh, 2022a).

*Video supplement.* A video of a full day of $\mathrm{Hydronn_{2,All}}$ retrieval compared to IMERG is provided as a supplement to this manuscript (Pfreundschuh, 2022b).

## Appendix A: Accuracy metrics

### A1 Quantitative precipitation estimation

The metrics used in the study to assess quantitative estimates of surface precipitation are the bias, mean-absolute error (MAE), mean-squared error (MSE) and the Pearson correlation coefficient . Their definitions are provided in table A1.

| Name | Calculation | Range | Best value |
|------|-------------|-------|------------|
| Bias | $\overline{y_{\mathrm{retrieved}} - y_{\mathrm{true}}}$ | $-\infty$ to $\infty$ | 0 |
| MAE | $\overline{\left\lvert y_{\mathrm{retrieved}} - y_{\mathrm{true}} \right\rvert}$ | 0 to $\infty$ | 0 |
| MSE | $\overline{\left( y_{\mathrm{retrieved}} - y_{\mathrm{true}} \right)^2}$ | 0 to $\infty$ | 0 |
| Correlation | $\dfrac{\overline{\left( y_{\mathrm{retrieved}} - \overline{y_{\mathrm{retrieved}}} \right)\left( y_{\mathrm{true}} - \overline{y_{\mathrm{true}}} \right)}}{\sigma_{y_{\mathrm{retrieved}}} \sigma_{y_{\mathrm{true}}}}$ | 0 to 1 | 1 |

**Table A1.** Accuracy metrics used to assess the accuracy of quantitative precipitation estimates $y_{\mathrm{retrieved}}$ against reference measurements $y_{\mathrm{true}}$. Overbars are used here to denote the mean over all pixels in the test data set and $\sigma_y$ to denote the standard deviation of the quantity $y$.

The disadvantage of these metrics is that they neglect the probabilistic character of the Hydronn retrievals and do not provide any information on well uncertainties are represented. We use the mean of the continuous ranked probability score (MCRPS) to assess the accuracy of probabilistic precipitation estimates. For a given, retrieved cumulative probability function $F$ the continuous ranked probability score (CRPS) with respect to the reference value $y_{\text{true}}$ is defined as

$$\text{CRPS}(F, y_{\text{true}}) = \int_{-\infty}^{\infty} (F(x') - \text{I}_{x > y_{\text{true}}})^2 \, dx, \tag{A1}$$

For a perfect prediction of $y_{\text{true}}$ without uncertainty, the predicted CDF $F$ takes the form of a step function and the CRPS is zero. A CRPS larger than zero measures the deviation from this perfect prediction. The CRPS takes into account both sharpness and calibration of the probabilistic precipitation estimates (Gneiting and Raftery, 2007).

## A2 Precipitation detection

To evaluate the skill of the retrievals to detect precipitation or precipitation exceeding a certain intensity we primarily rely on precision-recall (PR) curves. Since Hydronn provides a probability that represents how likely an observed pixel contains precipitation, the classification is based on the probability exceeding a certain detection threshold. The detection threshold can be adapted to balance false positive against false negative errors.

Because of this additional degree of freedom, assessing the classification accuracy for a single classification threshold does not fully characterize the skill of the retrieval. PR curves overcome this limitation by displaying the precision, i.e., the fraction of true positives and the total number of predictions, and the recall, i.e., the fraction of all raining pixels that are correctly detected, for all possible detection thresholds.

Retrievals that only predict a single precipitation rate can be used for precipitation detection by counting all pixels with precipitation exceeding a certain threshold as raining. Here the precipitation threshold can be used like a detection threshold to balance false positives and false negatives. In this way a PR curve can be drawn also for those retrievals.

In addition to PR curves we will use the false alarm rate (FAR), probability of detection (POD) and critical success index (CSI) as scalar metrics. They are defined as follows:

$$\text{FAR} = \frac{\text{FP}}{\text{TP} + \text{FP}} \tag{A2}$$

$$\text{POD} = \frac{\text{TP}}{\text{TP} + \text{FP}} \tag{A3}$$

$$\text{CSI} = \frac{\text{TP}}{\text{TP} + \text{FP} + \text{FN}} \tag{A4}$$

$$\tag{A5}$$

where TP denotes the number of true positives, FP false positives, and FN false negatives.

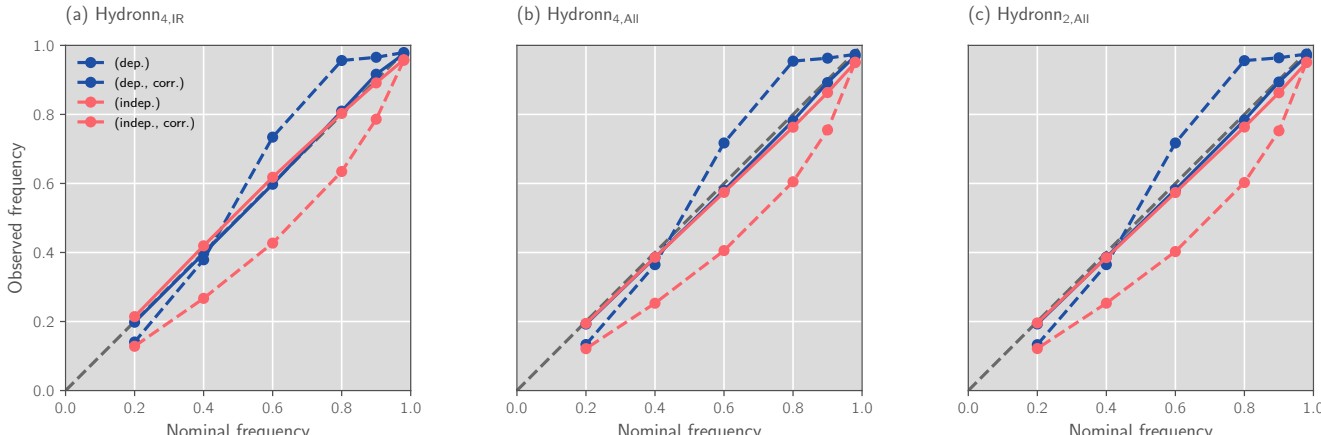

**Figure A1.** Like Fig. A1 but for June 2020.

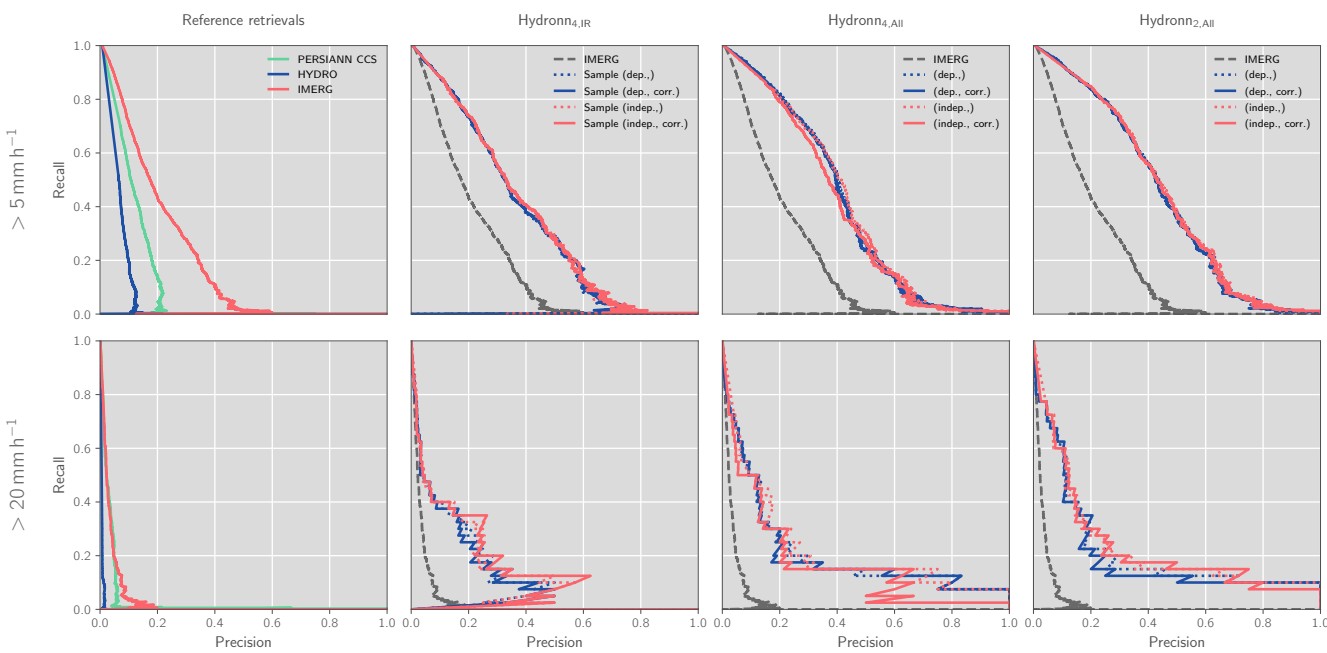

**Figure A2.** Like Fig. 16 but for June 2020.

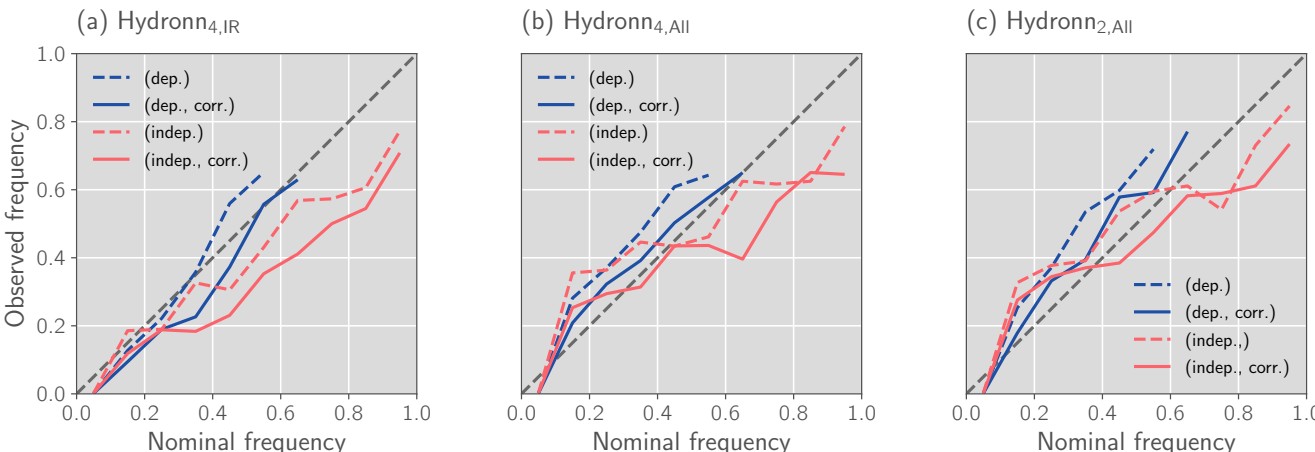

**Figure A3.** Like Fig. 17 but for June 2020.

*Author contributions.* II, PE and SP designed the study. SP and II developed the retrieval and analyzed the retrieval results. SP prepared the manuscript. AC and DV provided the gauge measurements, HYDRO retrieval results and valuable feedback.

*Competing interests.* No competing interests are present.

*Acknowledgements.* We would like to acknowledge the Brazilian National Institute of Meteorology for the provision of the gauge measurements.

Computations for this study were performed using several freely available programming languages and software packages, most prominently the Python language (Python Language Foundation, 2018), the IPython computing environment (Perez and Granger, 2007), the numpy package for numerical computing (van der Walt et al., 2011). Xarray (Hoyer and Hamman, 2017) and satpy (Raspaud et al., 2021) were used

for the processing of satellite data, PyTorch (Paszke et al., 2019) for implementing the machine learning models as well as matplotlib (Hunter, 2007) and cartopy (Met Office, 2010 - 2015) for generating figures.

The training of the machine learning models used in the study were performed on resources at Chalmers Centre for Computational Science and Engineering (C3SE) provided by the Swedish National Infrastructure for Computing (SNIC).

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
