# Peer review of "An improved near real-time precipitation retrieval for Brazil"

_EGUsphere, 2022_

## Author Comment (AC3)

**An improved near real-time precipitation retrieval for Brazil**

**Response to reviewer comments**

**1 Comments from reviewer 2**

**1.1 Specific comments**

**Reviewer comment 1:**

What is the purpose for the 4-km experiments given that the ABI has a native resolution of 2 km?

**Author response:**

The principal motivation for the 4-km experiments is that the current operational algorithm, HYDRO, operates at this resolution. Running the operational retrieval at 2-km resolution quadruples the computational and storage requirements. The 4-km experiments thus serve as a baseline to assess the benefits of running the retrieval at a higher resolution.
Furthermore, the GPM combined retrieval, which is used to generate the training data, has a comparably low resolution of $5\,\mathrm{km}$. Therefore, it is not evident that the retrieval can benefit from the increased resolution of the input data.

**Changes in manuscript:**

- We will reformulate the section that introduces the retrieval configurations and clearly motivate them.

**Reviewer comment 2:**

Line 152: The availability of sunlight does not affect the other IR and WV bands, only the availability of VIS bands. Therefore, it does not justify the use of only a single IR channel. Please clarify the reasoning here.

**Author response:**

Only the IR window channel is used for the $Hydronn_{4,IR}$ configuration because the same channel is used by the HYDRO and PERSIANN CCS retrievals. Therefore the $Hydronn_{4,IR}$ configuration can be used to assess the benefit of neural-network-based retrieval over the traditional power-law-based retrieval. Moreover, because this channel has been continuously available on a long sequence of geostationary sensors, it is suitable for the generation of precipitation records and used, for example, in GPM IMERG (Huffman et al., 2020) and the PERSIANN CDR datasets (Ashouri et al., 2015).

**Changes in manuscript:**

- We will extend the section that introduces the Hydronn$_{4,\text{IR}}$ configuration to clearly state this motivation.

**Reviewer comment 3:**

Lines 155-156: How are the values of the visible and near-IR bands treated by the CNN to differentiate daytime from nighttime scenes? Or is this something the CNN does without any intervention?

**Author response:**

The CNN is trained with input from all GOES channels regardless of the time of the day. Therefore, it learns to handle both day- and night-time observations, and no intervention is required to handle them.

**Changes in manuscript:**

- We will add a sentence to the description of the training scheme to mention this feature of the CNN retrieval.

**Reviewer comment 4:**

Sections 3.3 and 3.4: The description of the CNN needs much more detail to be understood by readers who are not experts on CNNs.

**Author response:**

We will extend the description of the CNN model upon which Hydronn is based.

**Changes in manuscript:**

- Section 3.3, which describes the neural network model used by Hydronn, will be extended.

**Reviewer comment 5:**

Lines 181-183: Please briefly define terms such as cross-entropy loss, logits, and softmax activation that would probably be unfamiliar to most readers.

**Author response:**

We will add definitions of these terms to the manuscript.

**Changes in manuscript:**

- Definitions of the terms 'cross-entropy loss', 'logits' and 'softmax activation' will be added to Sec. 3.4 of the manuscript.

**Reviewer comment 6:**

Line 165: Why is downsampling done for the 2-km retrievals rather than the 4-km retrievals? Shouldnt́ it be the other way around?

**Author response:**

$\text{Hydronn}_{2,\text{All}}$ ingests GOES observations at resolutions of $500\,\text{m}$, $1\,\text{km}$ and $2\,\text{km}$, which means that the network has to handle three input streams of different sizes. The additional downsampling layers in the $\text{Hydronn}_{2,\text{All}}$ retrieval are required to reduce the size of the $500\,\text{m}$ and $1\,\text{km}$ inputs so that they can be combined with the observations at $2\,\text{km}$ resolution. Since the network applies learnable transformations before and after the downsampling layers, it can actually learn to make use of the information at $500\,\text{m}$ and $1\,\text{km}$ resolution. This would not be the case if the observations were down-sampled prior to feeding them into the network.

The inputs for the $4\,\text{km}$ retrievals are down-scaled prior to feeding them into the network. Therefore no additional downsampling layers are required to reduce the size of the input to that of the output.

**Changes in manuscript:**

- We will extend the description of the neural-network architecture to better explain the role of the downsampling layers.

**Reviewer comment 7:**

Line 167: Please provide references to support this assertion about the number of internal features relative to other architectures.

**Author response:**

We reconsidered the sentence in question an decided to remove it from the manuscript. The principal reason for this is that by correcting a bug in our training code we were able to increase the number of features used in the NN architectures. In retrospect, we also consider the sentence imprecise and not really helpful for the reader.

**Changes in manuscript:**

- We will remove the sentence in question from the manuscript.

**Reviewer comment 8:**

Line 178: What in particular makes it easier to compute this sum on the binned PDF than on the quantiles? Please explain this more thoroughly.

**Author response:**

The principal reason for this is that we are not aware of any other way to calculate the distribution of the sum of two independent random variables.

Therefore, if a distribution is given in terms of a sequence of quantiles, it would be necessary to (1) use the quantiles to calculate the PDF of the distribution, (2) calculate the binned PDF of the sum of the variables, and finally (3) compute the desired quantiles of the resulting distribution.

If, on the other hand, the retrieval results is already a PDF in binned form, the sum can be calculated directly.

**Changes in manuscript:**

- We will extend the explanation of the calculation of the sum of the retrieval results to make the advantage of the binned PDF format clearer.

**Reviewer comment 9:**

Line 186, 442: Please explain what the degeneracy of (low) quantiles means.

**Author response:**

Non-raining pixels, which are assigned a precipitation rate (PR) of exactly $0\,\mathrm{mm\,h^{-1}}$, cause a discontinuity in the CDF of the distribution of precipitation rates. Since this makes it impossible to invert the CDF, not all quantiles are well defined. For example, it is impossible to determine the 25th percentile of the CDF of a pixel that is assigned a raining probability of $50\,\%$ because the CDF is 0 for all $PR < 0$ and larger than 0.5 for all $PR \geq 0$.

**Changes in manuscript:**

- We will add an explanation of the degeneracy of quantiles to Sec. 3.4 in the manuscript.

**Reviewer comment 10:**

Line 189: What is the rationale for creating outputs for 128 bins if only 14 quantiles are used?

**Author response:**

The retrieval results of Hydronn are represented as a binned approximation of the posterior PDF. The number of bins must be such that the full range of possible output values is covered and that the bins are sufficiently fine to ensure that they can accurately represent the posterior PDF in the region where most of its mass is located. Since it is impossible to know a priori where the mass will be located, Hydronn employs a relatively fine grid across the full range of possible output values.

A posteriori, the PDF can be represented more compactly using quantiles. To ensure that these quantiles, but also the mean, mode and samples of the posterior distribution, can be calculated with high accuracy, Hydronn such a high number of output bins.

**Changes in manuscript:**

- We will extend Sec. 3.4 to more clearly motivate the choice for number of output bins.

**Reviewer comment 11:**

Line 193: What does inference mean in this context?

**Author response:**

In the field of machine learning, 'inference' refers to the application of a statistical model to unseen data. It is used to distinguish the actual usage of a machine-learning model from its training process. In this case it means during the retrieval processing.

**Changes in manuscript:**

- We will add an explanatory phrase to the sentence in question.

**Reviewer comment 12:**

Line 196: What does posterior mean in this context?

**Author response:**

The posterior distributions here are just the results of the retrieval for each observation.

**Changes in manuscript:**

- We will rephrase the sentence in question to make it clear what is meant with 'posterior'.

**Reviewer comment 13:**

Lines 203-204: Why specifically will assuming that the retrieval uncertainty is temporally independent cause the uncertainty to decay for consecutive identical observations?

**Author response:**

Our assertion that the retrieval uncertainty of independent measurements decays is based on the following reasoning:
Given a sequence of random variables $X_1, \ldots X_n$ with finite mean and finite and bounded variance, let $\sigma_{\max} = \max\{\mathrm{Var}(X_1), \ldots \mathrm{Var}(X_n)\}$ denote the maximum variance of any of the distributions. It is then possible to derive the following upper bound for the variance of the mean of the random variables:

$$\mathrm{Var}(\frac{1}{n}\sum_i X_i) = \frac{1}{n^2}\sum_i \mathrm{Var}(X_i) \tag{1.1}$$

$$\leq \frac{1}{n^2} n \sigma_{\max} \tag{1.2}$$

$$\leq \frac{\sigma_{\max}}{n} \tag{1.3}$$

Note that (1.1) holds because of the independence of the random variables. This means that the variance of the mean of the random variables will always be lower than that of the distribution with the highest variance and decay as more observations are included in the mean.

**Reviewer comment 14:**

Lines 279-280: This is true, but it would be very scientifically interesting to see the relative degree of improvement during the day and night e.g., to quantify the value of the visible and near-IR data.

**Author response:**

We agree with the reviewer that this would be an interesting question to investigate. However, we fear that a simple comparison of retrievals during day and night time would be misleading due to the confounding effect of the pronounced daily cycle of precipitation. A fairer comparison may be to add an additional retrieval configuration to the study. However, considering the associated computational cost and that the objective of our study was maximizing the accuracy of the precipitation retrievals, we consider this to be outside the scope of our study.

**Reviewer comment 15:**

Line 342: Why does assuming dependent retrieval errors lead to the uncertainties being overestimated?

**Author response:**

That the assumption of dependent retrieval errors causes uncertainties to be estimated can be seen from the fact that the calibration curves in Fig. 10 lie above the diagonal. This means that the retrieved confidence are too wide, which causes the true precipitation value to lie withing them more often that is expected based on the interval.
The observation that the assumption of dependent retrieval errors leads to an overestimation of the uncertainties is therefore primarily an experimental result. The likely reason for this is that the true errors are not completely dependent but include an independent component that causes the real uncertainties to decay.

**Changes in manuscript:**

- We will extend the discussion of Fig. 10 to reflect the above reasoning.

**Reviewer comment 16:**

Lines 361-362 and 473: How precisely does varying the probability threshold have a calibrating effect on the retrieval results?

**Author response:**

Upon reconsidering the statements in question, we have come to the conclusion that our results don't provide any evidence of a calibrating effect. We will therefore remove the statements from the manuscript.

**Changes in manuscript:**

- We will remove the statements from the manuscript.

**Reviewer comment 17:**

Lines 378-379: What probability threshold was tuned, and why was a FAR close to IMERG the criterion for doing so?

**Author response:**

Due to its probabilistic nature, Hydronn is able to predict the probability that the precipitation at a given pixel exceeds a certain threshold. This probability can be used to detect strong precipitation by choosing a probability threshold above which a pixel is assumed to contain heavy precipitation. The probability threshold can be used to tune either POD or FAR to an arbitary value. In practice, the threshold should be chosen according to the application at hand. Since purpose of the evaluation was the comparison to IMERG, we tuned the FAR to that of IMERG.

**Reviewer comment 18:**

Table 4: Why precisely does correcting when assuming independent errors actually degrade the POD, FAR, and CSI relative to the uncorrected version?

**Author response:**

We think that the degradation of POD, FAR, and CSI for the assumption of independent retrieval uncertainties is due to an error in the calculation of the correction factors. The calculation of the a priori distribution of hourly accumulations assumed fully independent samples of the a priori distribution of instantaneous precipitation estimates. Since this neglects the dependence that is introduced by the temporal coherence of the satellite observations, it likely caused the correction factors to be incorrect.

We will adopt a different approach for calculating the correction factors in the revised version of the manuscript to see whether this improves the results.

**Changes in manuscript:**

- We will adopt a corrected approach to calculate the correction factors.

- We will rewrite Sec. 3.6 to reflect these changes.

**Reviewer comment 19:**

Figure 13: The use of grayscale for the rain rates and colors for the errors makes the plots very hard to read. Would it be possible to instead plot the satellite rain rates in color and plot the corresponding gauge values using the same color scheme? Similar values would have very little contrast whereas large errors would produce sharp contrasts.

**Author response:**

We agree with the reviewer that Fig. 13 contains too much information for a single plot. We will replace it in the revised manuscript with two plots showing accumulation maps and a scatter plot comparing the accumulations against the gauge measurements.

**Changes in manuscript:**

- We will revise Fig. 13 to only show an interpolated map of the gauge measurements and the accumulation maps obtained from the different retrievals.

- We will add a figure showing a scatter plots of the gauge-measured and retrieved precipitation accumulations.

[Figure]

Figure 1.1: 99th percentile of hourly precipitation during December 2020. Points show the locations of the gauges used for the evaluation during December 2020. The coloring shows the 99th percentile of the distribution of hourly precipitation

**Reviewer comment 20:**

Line 432 and Fig. 15: Please define more precisely what the 99th percentile of the distribution means. If each point in Fig. 15 is the 99th percentile of all of the rainfall values for a particular gauge location during the month of Dec. 2020, why are there so many values < 5 mm/h? Is it the dry season in some of these locations?

**Author response:**

Figure 15 does, in fact, show the 99th percentile of the rainfall values for each gauge location for December 2020. Although December is generally the beginning of the rain season in many parts of the country, the 99th percentile of the hourly precipitation remains below 5 mm for several stations.
As can be seen in Fig. 1.1, most of them are located in the semi-arid east of the country which makes these results plausible. In addition to this, the precipitation in December 2020 was below average in large parts of the country (Source in Portuguese: Grupo de Previsão de Tempo CPTEC/INPE, 2020).
Nonetheless, some of the stations in the western parts of the country exhibit very low values for the 99th percentile of the distribution of precipitation. This indicates that some of thos measurements may be faulty.

**Changes in manuscript:**

- We will reformulate the description of Fig. 15 to make it clear that the displayed quantity corresponds to the 99th percentile of the distribution of hourly precipitation accumulations for each gauge station.

**Reviewer comment 21:**

Lines 435-436: Are there any specific assertions in the published literature that HYDRO and PERSIANN-CCS were both developed to correctly represent heavy precipitation at the presumed expense of skill for lighter precipitation?

**Author response:**

We would like to thank the reviewer for commenting on this statement, whose formulation we consider problematic in hindsight. A more suitable statement would be that HYDRO and PERSIANN CCS were developed with a focus on convective precipitation at the expense of retrieval skill for stratiform scenarios. This is also acknowledged in the published literature.

**Regarding HYDRO:**

The HYDRO retrieval is based on the Hydroestimator, which is in-turn based on the Autoestimator. The study by Vicente et al. (1998) presents the original form of the Autoestimator. Regarding the data used to derive the regression curve that relates IR brightness temperatures and precipitation rates the manuscript states:

> The original set of observations, collected during the months of March to June 1995, was composed of 120 pairs of IR cloud-top temperatures and radar- derived rainfall estimates with 4 km by 4 km pixel resolution. Only convective rain systems were considered.

Although the algorithm includes corrections to adopt the relation to other meteorological regimes, the methodology exhibits a bias for convective precipitation, which is also acknowledged in the conclusions of the paper:

> Independent and qualitative studies not shown in this paper have demonstrated that in contrast to the reasonable performance of the technique for well-defined and short duration convective systems, poor results are common for stratiform cloud systems

Moreover, the study by Scofield and Kuligowski (2003) (titled 'Status and Outlook of Operational Satellite Precipitation Algorithms for Extreme-Precipitation Events') that introduces the Hydroestimator states:

> All of the estimates display relatively little bias for cold-top events, which is not surprising given that they were calibrated for such events and the assumptions behind satellite QPE algorithms generally work best for cold-top events.

**Regarding PERSIANN CCS:**

The available literature on the PERSIANN CCS algorithm indicates that the original cloud classes and $T_B$-precipitation curves are based on collocations from just a single summer month over the western CONUS (Hong et al., 2004):

> After completing the cloud-patch feature extraction, the system is calibrated using GOES infrared images and radar-rainfall maps for June 1999 over the region of 25°N 45°N and 100°N130°W (both datasets are mapped to 0.04° latitude × 0.04° longitude scale).

Although an a bias correction based on passive-microwave data has been added to the operational algorithm (Karbalaee et al., 2017), the underlying estimation method seems to have remained the same.

Moreover, Nguyen et al. (2018) states

> Recent developments include integrating deep learning approaches, adding water vapor channel information (Tao et al., 2017), using PMW data for bias adjustment of PERSIANN-CCS (Karbalaee et al., 2017), incorporating MODIS and CloudSat information (Nasrollahi et al., 2013), and using probability matching methods to improve warm rainfall detection in PERSIANN-CCS.

indicating that the precipitation from warm clouds remains an issue for the algorithm. In summary, both retrievals were developed with a focus on convective precipitation. A likely better explanation for the improved accuracy in estimating extreme precipitation is therefore that by restricting the analysis to the high percentiles of the precipitation distribution the contribution from convective precipitation events is increased, which leads to the observed improved performance from the two retrieval algorithms.

**Changes in manuscript:**

- We will reformulate the sentence to state that HYDRO and PERSIANN CCS were developed with a focus on convective precipitation and struggle with precipitation from colder clouds.

**1.2 Technical Comments**

**Reviewer comment 1:**

Line 39: For consistency, it might be better to cite Schmit et al. (2018) instead of Schmit et al. 2005) since the former is cited in lines 65 and 129.

**Author response:**

Since Schmit et al. (2005) is a peer-reviewed publication, we are under the impression that it is more suitable as reference for the GOES ABI. We will therefore replace the existing citation of Schmit et al. (2018) with Schmit et al (2005).

**Changes in manuscript:**

- We will replace the references to Schmit et al. (2018) with Schmit et al. (2005).

**Reviewer comment 2:**

Line 46, 55, 93, 574-577: Scofield and Kuligowski (2003a) and (2003b) are the same paper.

**Author response:**

We would like to thank the reviewer for pointing out this mistake, which we will of course correct in the revised version of the manuscript.

**Changes in manuscript:**

- We will remove the duplicated reference.

**Reviewer comment 3:**

Line 54: Please cite Nguyen et al. (2020) here in reference to PERSIANN-PDIR.

**Author response:**

We will add the citation in the revised version of the manuscript.

**Changes in manuscript:**

- We will add the citation in the revised version of the manuscript.

**Reviewer comment 4:**

Line 64: Is Hydronn an acronym (e.g., Hydro-Neural Network) or does the name have a different meaning?

**Author response:**

Hydronn is the name of the retrieval algorithm. It is named after the character Hydron from the He-man comic series. It also functions as a portmanteau of the words Hydro and NN (for neural network) but we prefer both spelling and pronounciation of Hydronn over HydroNN, which is also why we did not introduce it in this way in the manuscript.

**Reviewer comment 5:**

Line 80: Replace consists with consist (measurements is plural).

**Author response:**

We will correct this in the revised version of the manuscript.

**Reviewer comment 6:**

Lines 85, 86, 88: Northwest should not be capitalized unless it is a proper name.

**Author response:**

We will correct this in the revised version of the manuscript.

**Reviewer comment 7:**

Line 86 Many readers may not know that Amazonas is the proper name for a state in Brazil, so the Brazilian state of Amazonas would be clearer.

**Author response:**

We will correct this in the revised version of the manuscript.

**Reviewer comment 8:**

Line 88: Replace manifest with e.g., is associated with.

**Author response:**

We will correct this in the revised version of the manuscript.

**Reviewer comment 9:**

Line 118: Replace available first with available only.

**Author response:**

We will correct this in the revised version of the manuscript.

**Reviewer comment 10:**

Line 135: replace criterion with approach.

**Author response:**

We agree with the reviewer that 'criterion' is not a suitable expression here. However, we think that 'interpolation' is actually more specific than 'approach', so will will use 'interpolation' in the revised version of the manuscript.

**Reviewer comment 11:**

Lines 133, 386, and elsewhere: please ensure that all dates in this manuscript match the format used in EGUsphere.

**Author response:**

We would like to thank the reviwer for pointing out this issue. We will of course correct this in the revised version of the manuscript.

**Reviewer comment 12:**

Line 150: A better wording would be a long time series of geostationary sensors.

**Author response:**

We will change the formulation in the revised version of the manuscript.

**Reviewer comment 13:**

Line 350: Insert to before derive.

**Author response:**

We will correct this in the revised version of the manuscript.

**Reviewer comment 14:**

Line 354: Is retrieved meant rather than predicted?

**Author response:**

In the field of machine learning the term 'predict' is commonly used when a model is evaluated. We have therefore used the terms 'predict' and 'retrieve' interchangeably in the manuscript. We acknowledge that this may cause confusion for readers from meteorological backgrounds and will change the wording in the revised version of the manuscript.

**Reviewer comment 15:**

Line 354: Pixel should be plural.

**Author response:**

We will correct this in the revised version of the manuscript.

**Reviewer comment 16:**

Line 362: Is worse detection accuracy than at 5 mm/h meant here?

**Author response:**

Yes, this is what is meant here. We will reformulate the sentence to make this clear.

**Reviewer comment 17:**

Figure 12 caption: add at a rate of 5 mm/h to the end of the caption for clarity.

**Author response:**

We will adopt this change in the revised version of the manuscript.

**Reviewer comment 18:**

Line 387: Floodings should be singular or replaced with floods.

**Author response:**

We will correct this in the revised version of the manuscript.

**Reviewer comment 19:**

Lines 387, 404, 520: Is this citation and reference formatted correctly?

**Author response:**

According to the guidelines for referencing websites in AMT (`https://www.atmospheric-measurement-te` `net/submission.html#references`), the reference should be formatted correctly except for the wording used for the last access date.

**Changes in manuscript:**

- We will change the wording for the last access date of the reference in question.

**Reviewer comment 20:**

Line 388: Replace were with was.

**Author response:**

We will correct this in the revised version of the manuscript.

**Reviewer comment 21:**

Line 394: Please indicate the location of Duque de Caxias in Fig. 13.

**Author response:**

The location of Xerém, which is the neighborhood in Duque de Caxias in which the rain gauge is located, is already indicated in Fig. 13. However, the manuscript does not clearly state the relation between Duque de Caxias and Xerém. We will reformulate the introduction of Sec. 4.3 to make it clear where the flooding occurred.

**Reviewer comment 22:**

Line 430: Replace by with of.

**Author response:**

We will correct this in the revised version of the manuscript.

**Reviewer comment 23:**

Line 431: Runoff is a single word.

**Author response:**

We will correct this in the revised version of the manuscript.

**Reviewer comment 24:**

Line 433: Station should be plural.

**Author response:**

We will correct this in the revised version of the manuscript.

**Reviewer comment 25:**

Line 434: Replace similar accuracy as with accuracy similar to.

**Author response:**

We will correct this in the revised version of the manuscript.

**Reviewer comment 26:**

Line 461: A more precise wording might be correct for variations in the distribution of precipitation rates in the training data relative to comparable ground validation data.

**Author response:**

We will adopt this suggestion in the revised version of the manuscript.

**Reviewer comment 27:**

Line 465: Replace stronger with more strongly.

**Author response:**

We will correct this in the revised version of the manuscript.

**Reviewer comment 28:**

Line 473: Replace small with low.

**Author response:**

We will correct this in the revised version of the manuscript.

**Reviewer comment 29:**

Line 475: Constant in time, space, or both?

**Author response:**

We will correct this in the revised version of the manuscript.

**Reviewer comment 30:**

Line 485: Please define GPM CO in line 69 so the acronym is already defined.

**Author response:**

We will reformulate this sentence to correct the use of acronyms in the manuscript.

**Reviewer comment 31:**

Lines 485-486: the latitude range of the GPM DPR is actually 65 °S to 65 °N when the instrument swath is accounted for.

**Author response:**

We will correct this in the revised version of the manuscript.

**Reviewer comment 32:**

Line 489: This is the first time that the CNN is described as a probabilistic regression approach; this concept should be introduced earlier in the manuscript.

**Author response:**

We will revise the manuscript to introduce the concept of probabilistic regression already in the introduction.

**Reviewer comment 33:**

Line 494: Delete the comma after resolutions.

**Author response:**

We will correct this in the revised version of the manuscript.

**Reviewer comment 34:**

Line 587: The Python Language Foundation should be considered as starting with P since The is ignored when alphabetizing entries.

**Author response:**

We will correct this in the revised version of the manuscript.

**Bibliography**

Ashouri, H., Hsu, K.-L., Sorooshian, S., Braithwaite, D. K., Knapp, K. R., Cecil, L. D., Nelson, B. R., and Prat, O. P.: PERSIANN-CDR: Daily Precipitation Climate Data Record from Multisatellite Observations for Hydrological and Climate Studies, Bulletin of the American Meteorological Society, 96, 69 – 83, https://doi.org/10.1175/BAMS-D-13-00068.1, 2015.

Grupo de Previsão de Tempo CPTEC/INPE: ÍNTESE SINÓTICA DEZEMBRO DE 2020, `https://s1.cptec.inpe.br/admingpt/tempo/pdf/sintese_122020.pdf`, 2020.

Hong, Y., Hsu, K. L., Sorooshian, S., and Gao, X. G.: Precipitation estimation from remotely sensed imagery using an artificial neural network cloud classification system, J. Appl. Meteor., 43, 1834–1852, 2004.

Huffman, G. J., Bolvin, D. T., Braithwaite, D., Hsu, K.-L., Joyce, R. J., Kidd, C., Nelkin, E. J., Sorooshian, S., Stocker, E. F., Tan, J., Wolff, D. B., and Xie, P.: Integrated Multi-satellite Retrievals for the Global Precipitation Measurement (GPM) Mission (IMERG), pp. 343–353, Springer International Publishing, Cham, https://doi.org/10.1007/978-3-030-24568-9_19, 2020.

Karbalaee, N., Hsu, K., Sorooshian, S., and Braithwaite, D.: Bias adjustment of infrared-based rainfall estimation using Passive Microwave satellite rainfall data, Journal of Geophysical Research: Atmospheres, 122, 3859–3876, https://doi.org/https://doi.org/10.1002/2016JD026037, 2017.

Nguyen, P., Ombadi, M., Sorooshian, S., Hsu, K., AghaKouchak, A., Braithwaite, D., Ashouri, H., and Thorstensen, A. R.: The PERSIANN family of global satellite precipitation data: a review and evaluation of products, Hydrology and Earth System Sciences, 22, 5801–5816, https://doi.org/10.5194/hess-22-5801-2018, 2018.

Scofield, R. A. and Kuligowski, R. J.: Status and outlook of operational satellite precipitation algorithms for extreme-precipitation events, Weather and Forecasting, 18, 1037–1051, 2003.

Vicente, G. A., Scofield, R. A., and Menzel, W. P.: The Operational GOES Infrared Rainfall Estimation Technique, Bulletin of the American Meteorological Society, 79, 1883 – 1898, https://doi.org/10.1175/1520-0477(1998)079<1883:TOGIRE>2.0.CO;2, 1998.

---

## Author Response (AR1)

**An improved near real-time precipitation retrieval for Brazil**

**Response to reviewer comments**

We want to thank both reviewers for taking the time to read our manuscript and providing feedback. The provided comments and suggestions helped us to improve the manuscript and we hope that we were able to address all of the reviewer's concerns. The two most fundamental changes that we have implemented are the following:

1. We have extended the evaluation of the retrievals for instantaneous precipitation rates as well as accumulations.

2. We have applied a different method to derive the correction factors.

Since the first change concerned the core subject of the study, we had to adapt large parts of the manuscript in addition to the direct changes that are listed in this response. In addition to this, we have corrected a number of mistakes that we encountered during the preparation of the revised manuscript. One of them was a bug in the training of our neural network models. This lead to slight improvements for all Hydronn retrievals. Finally, since the manuscript has grown in length, we have removed Table 4 from the original manuscript because it didn't add any new information to the manuscript. Below, the reviewer's comments are listed together with the author's response and changes in the manuscript. Line numbers are given with respect to the revised manuscript.

**1 Comments from reviewer 1**

The paper presents a convolutional neural network architecture for IR precipitation retrieval over Brazil. The training data are from IR and GPM combined retrievals. The framework is extended such that it can provide uncertainty of the retrievals. The estimates are compared with ground-based gauge data to validate the retrievals. The paper is well written and is of high-quality. I have the following comments.

**1.1 Major comments**

**Reviewer comment 1**

Validation only with a month of gauge data is not sufficient for clamming those improved results in the abstract. Seasonal to annual validation results are needed to make those claims.

**Author response:**

We agree with the reviewer that a more thorough evaluation of the retrieval accuracy over extended periods is desirable. However, we also want to point out that the evaluation presented in Sec. 3.1 in the original version of the manuscript covers the full year of 2020. Thus, the evaluation does already cover longer time scales than the month used to evaluate the precipitation accumulations. Nonetheless, it is true that an analysis of the accuracy across different time scales is missing from the manuscript.

One difficulty with extending the evaluation against gauge measurements is the storage capacity required to store input data and results. For example, input and output data of the Hydronn retrievals for one month require 2.5 TB of storage.

We therefore propose the following extension of our evaluation scheme, which will allow assessing the retrieval performance across seasonal time scales within the constraints of the compute resources that are currently available to us:

1. We will extend the evaluation against the GPM combined measurements to cover the full year of 2020. We will compare our retrievals to GPROF and HYDRO. We choose GPROF instead of IMERG for this comparison because the retrievals can be directly collocated in time with the reference data, which is not possible for the gridded IMERG data. The GPROF retrievals therefore constitute a stronger baseline for instantaneous precipitation estimates. We choose not to include PER-SIANN CCS because the data is only available at hourly resolution and comparison against the instantaneous reference measurements would make the product look overly bad.

2. We will extend the evaluation against the gauge measurements to also cover June 2020.

This extended evaluation scheme allows us to show the robustness of the accuracy of our retrievals for both instantaneous and accumulated measurements. Since the intended application of the Hydronn retrievals are near real-time retrievals, we consider the assessment of the retrieval accuracy across annual time scales outside the scope of this manuscript. We will also extend the discussion of the retrieval accuracy to reflect those points.

**Changes in manuscript**

- We extended the assessment of instantaneous precipitation estimates, which now compares Hydronn to GPROF GMI and HYDRO and assesses the seasonal variability of the accuracy.

> **Changes starting in line 355:**
>
> ### 1.1.1
>

[revised manuscript text omitted]

**Reviewer comment 2**

A single storm retrieval is missing. It is imperative to show the output of the algorithm in retrieval of a single or multiple storms and compare the results with the combined GPM retrievals as a reference. One retrieval snapshot speaks very clearly about the skill of the algorithm in reconstructing the training data and retrieve spatial structure of precipitation.

Table 1.1: Accuracy metrics for the retrieved mean precipitation compared to gauge measurements at different time scales. The best values in each column are marked using bold font. Definitions of all metrics are provided in appendix A1.

**June 2020**

| Retrieval | Bias [mm h$^{-1}$] | MAE [mm h$^{-1}$] Hourly | Daily | Monthly | M Hourly |
|---|---|---|---|---|---|
| HYDRO |  -0.055 |  0.106 |  0.079 |  0.06 |  0.611 |
| PERSIANN CCS |  -0.035 |  0.115 |  0.085 |  0.053 |  0.671 |
| IMERG |  -0.013 |  0.1 |  0.065 |  **0.034** |  0.393 |
| Hydronn$_{4,IR}$ |  **-0.002** |  0.108 |  0.07 | 0.036 | 0.404 |
| Hydronn$_{4,All}$ | -0.034 | 0.084 | 0.059 | 0.043 | 0.361 |
| Hydronn$_{2,All}$ | -0.031 | **0.084** | **0.058** | 0.04 | **0.345** |

**December 2020**

| Retrieval | Bias [mm h$^{-1}$] | MAE [mm h$^{-1}$] Hourly | Daily | Monthly | M Hourly |
|---|---|---|---|---|---|
| HYDRO | -0.037 | 0.32 | 0.215 | 0.106 | 3.1 |
| PERSIANN CCS | 0.096 |  0.398 |  0.285 |  0.151 |  3.594 |
| IMERG | 0.014 | 0.285 | 0.196 | 0.08 | 1.9 |
| Hydronn$_{4,IR}$ | **-0.002** | 0.283 | 0.189 | 0.088 | 2.011 |
| Hydronn$_{4,All}$ |  -0.006 |  0.235 |  0.159 |  0.076 |  1.797 |
| Hydronn$_{2,All}$ | 0.011 |  **0.226** |  **0.153** |  **0.074** |  **1.704** |

[Figure]

Figure 1.3:  Retrieved mean precipitation during June and December 2020. The first two columns show the results for June. Columns three and four show the results for December. Shading in the background of each panel shows the spatial distribution of the mean precipitation of the corresponding retrieval. Colored hexagons show the spatial distributions of the retrieval biases with respect to the gauge measurements.

[Figure]

Figure 1.4: Scatter plots of gauge measurements against retrieved daily accumulations against gauge measurements for the reference retrievals HYDRO, PERSIANN CCS, IMERG and the three Hydronn configurations. The first two columns show the results for June 2020. Columns three and four show the results for December 2020. Frequencies have been normalized columnwise to improve the visibility of high reference precipitation.

[Figure]

Figure 1.5: Measured and retrieved daily cycles of precipitation.  The first column displays the daily cycles retrieved by the three reference retrievals (solid lines) and the gauge measurements (dashed line) for reference.  The second column corresponding diurnal cycles for the three Hydronn configurations. The first row shows the results for June 2020 and the second results for December 2020.

**Author response:**

We will add retrieval results for an overpass of the GPM core observatory over a meso-scale convective system. We also add a comparison of the the retrieval results to GPM PMW retrievals and the HYDRO algorithm. To further illustrate the capabilities of our retrieval, we will also include a video of the retrieval results at 10-minute resolution over 24 hours as a supplement with the manuscript. The video can be found here: `https://doi.org/10.5281/zenodo.7117246`

**Changes in manuscript**

- We added a subsection that analyzes the retrieval performance for an overpass over a mesoscale convective system and compares it to HYDRO, GPROF and IMERG.

  > **Changes starting in line 331:**
  >
  >
  >
  > ### Case study
  >
  > As first step in the evaluation of the instantaneous precipitation estimates, we consider retrieved precipitation for an overpass of the GPM satellite over a meso-scale convective system (MCS) in the border region between Argentina, Paraguay and Brazil on 16 December 2020, 13:59:00 UTC. The retrieval results are displayed together with a natural color composite in Fig. ~~1.9. It is apparent that the assumptions of temporally dependent uncertainties yields better agreement with the gauge data than the assumption of temporally independent uncertainties. The resulting correction factors are thus closer to the $y = 1$ line for the dependence assumption. We found that it was necessary to truncate the correction factors corresponding to the independence assumption at $r = 10^3$ because larger values would amplify numerical noise leading to the rare occurrence of unrealistically high precipitation values, which would distort the retrieval results~~1.6. The GPROF GMI and IMERG retrievals exhibit good agreement with the GPM CMB results. This is expected, not only because GPROF and IMERG incorporate PMW observations, but also because GPM CMB is used to derive the retrieval database used by GPROF, and GPROF is in turn used by IMERG.
  >
  > The HYDRO retrieval, on the other hand, does not agree well with the GPM CMB results. The heavy precipitation retrieved by HYDRO is located in the western part of the MCS, whereas the GPM CMB shows the very heavy precipitation in the north-eastern parts of the system. The $\text{Hydronn}_{4,IR}$ retrieval captures the overall structure of the MCS better than HYDRO but fails to represent its smaller-scale structures. Both, the $\text{Hydronn}_{4,All}$ and $\text{Hydronn}_{2,All}$ retrievals improve upon this and yield results that are very similar to those of

GPROF GMI and IMERG.

Accuracy metrics for the the MCS overpass with respect to the CMB reference data are provided in table 1.2. $\mathrm{Hydronn}_{4,IR}$ and $\mathrm{Hydronn}_{2,All}$ both exhibit dry biases of the same magnitude as HYDRO and $\mathrm{Hydronn}_{4,All}$ even exceeds those. However, all Hydronn retrievals yield significantly more accurate results than HYDRO in terms of the other metrics. The $\mathrm{Hydronn}_{2,All}$ retrieval even surpasses IMERG in terms of MSE, MAE, and CSI and achieves results close to those of GPROF GMI.

**1.2 Results**

This ~~section presents the evaluation of the Hydronn retrievals, which is split into three parts. The first part analyzes the nominal performance of the three Hydronnconfigurations on the held-out test data. The second part compares the retrieved hourly accumulations to the gauge measurements and the reference precipitation algorithms. Finally, the third part presents a case study of a heavy precipitation event that occurred during the validation period~~evaluation indicates that, while the total amount of precipitation remains less accurate for Hydronn than for the PMW retrievals, the spatial structure of the retrieved precipitation is captured equally well. Moreover, it should be noted that the revisit time for the GPM constellation of PMW sensors at these latitudes is around 1 h (Hou et al., 2014). Hydronn, however, can provide precipitation retrievals every 10 min. While increasing the temporal coverage of the precipitation measurements is also what IMERG aims to achieve by merging PMW retrievals with observations and retrievals from geostationary sensors, this seems to degrade the accuracy of the instantaneous precipitation retrievals. To further illustrate the capabilities of the Hydronn retrievals a video of precipitation estimates for the MCS case is provided as a digital supplement to this manuscript (Pfreundschuh, 2022).

Table 1.2: Retrieval accuracy metrics for the MCS overpass shown in Fig. 1.6. Definitions of all metrics can be found in appendix A1.

| Retrieval | Bias | MAE $[\mathrm{mm\ h}^{-1}]$ | MSE $[(\mathrm{mm\ h}^{-1})^2]$ | Correlation | POD | FAR | C |
|---|---|---|---|---|---|---|---|
| Hydro | -0.598 | 2.495 | 46.291 | 0.228 | 0.707 | 0.169 | 0.6 |
| GPROF GMI | **−0.163** | **1.699** | **27.467** | 0.552 | **0.998** | 0.519 | 0.4 |
| IMERG | 0.28 | 2.204 | 40.664 | 0.429 | 0.973 | 0.213 | 0. |
| $\mathrm{Hydronn}_{4,IR}$ | 0.612 | 2.36 | 30.768 | 0.506 | 0.901 | 0.192 | 0. |
| $\mathrm{Hydronn}_{4,All}$ | 0.813 | 2.362 | 33.038 | 0.524 | 0.916 | **0.138** | 0.7 |
| $\mathrm{Hydronn}_{2,All}$ | 0.57 | 2.105 | 29.918 | **0.564** | 0.922 | 0.14 | **0.8** |

[Figure]

Figure 1.6: A mesoscale convective system over the border region between Argentina, Paraguay, and Brazil on 16 December 2020 observed by GPM and GOES 16. Panel (a) shows a natural color composite (generated using (Raspaud et al., 2021)). Panel (b) shows retrieved surface precipitation from CMB retrieved using combined radar and passive microwave observations. Panel (b) shows precipitation retrieved by GPROF GMI using only passive microwave observations. Panel (c) shows the surface precipitation from the IMERG Final product. Panel (d) shows surface precipitation retrieved by HYDRO from GOES ABI observation. Panels (e), (f), (g) show the corresponding results from the three Hydronn configurations.

**Reviewer comment 3**

Error metrics are only represented cumulatively. The expectation is that paper presents the quality of retrievals for an individual storm in terms of detection accuracy (e.g., probability of detection, miss) and then focuses on estimation quality metrics at different time scales from a storm scale to monthly and seasonal.

**Author response:**

As stated in response to reviewer comments 1 and 2, we will extend the evaluation of instantaneous precipitation estimates and include an assessment of the retrieval accuracy for a single storm case.

Estimation quality metrics are already reported for hourly, daily, and monthly time scales in Tab. 3. The extension of the evaluation scheme proposed in response to comment 1 will also allow to assess the retrieval accuracy across seasonal time scales.

**Changes in manuscript**

- We have added an evaluation of a single-storm retrieval. See response to reviewer comment 2.

- We have extended the evaluation of instantaneous precipitation estimates to include all available overpasses over the training domain in 2020 and compare the results to HYDRO and GPROF. See response to comment 1.

**Reviewer comment 4**

This needs to be clarified whether the training data were only over Brazil or not. If this is the case, then the provided improved statics are not of surprise. This issue needs to be stated in the abstract.

**Author response:**

We will add a statement to the abstract stating that the training data is restricted to South America and a figure showing the region used to extract the training data for the retrieval.

Moreover, we think the reviewer's suggestion that our reported improvements are 'not of surprise' brings up an interesting question. Namely, whether these improvements are due to the more representative training data or the more expressive statistical models used by Hydronn. To investigate this question, we will add a further evaluation of the retrieval accuracy over a separate region (R2 in Fig. 1.7) over the northern hemisphere.

**Change in manuscript:**

- We added the information that the training was restricted to South America to the abstract.

[Figure]

Figure 1.7: GOES-16 true-color composite from September 23, 2019 (generated using the `natural_color` composite in satpy (Raspaud et al., 2021)). The rectangle R1 marks the domain over South America, which was used for the extraction of training and testing collocations between the ABI on GOES 16 and GPM CMB. Dashed polygons show the boundaries of the training scenes extracted for this day together with the collocated GPM CMB results. The rectangle R2 marks the secondary domain which is used as an additional test domain to assess the impact of the spatially limited training domain.

**Changes starting in line 9:**

The retrieval is trained using more than three years of co-locations with combined radar and radiometer retrievals from the Global Precipitation Measurement (GPM)  core observatory over South America.

- We added a subsection that assess the retrieval accuracy outside of the training domain.

**Changes starting in line 375:**

**Accuracy over northern hemisphere**

The neural network models used by Hydronn were trained using only observations over Brazil (R1 in Fig. 1.7). The results from the previous section indicate that Hydronn achieves significantly higher accuracy than HYDRO and even approaches the accuracy of  GPROF GMI when all available ABI channels are used. This, of course, raises the question whether Hydronn still works outside the region used for its training.

 To investigates this, we have evaluated the retrievals using collocations from the 1st, 6th, 11th, 16th, 21st and 26th day of every month of 2020 over the northern hemisphere (marked as R2 in Fig. 1.7). The results for GPROF and the Hydronn retrievals are displayed in Fig. 1.8. While the accuracy of GPROF is higher than over Brazil, the accuracy of all Hydronn configurations decreases. However, the decrease remains relatively small compared to the improvements over HYDRO that were observed over Brazil. This suggests that the neural networks learned robust relationships between the ABI observations and surface precipitation that generalize to observations from outside their training domain.

[Figure]

Figure 1.8:  Retrieval accuracy with respect to GPM CMB for  overpasses on  the  1st, 6th, 11th 16th 21st and 26th day of each month of 2020 over the domain R2. Each panel shows the average of a metric over the full year as well as its seasonal variability.

**Reviewer comment 5**

The way the paper explains the Bayesian retrieval is confusing. First, what is the prior distribution? Just obtaining uncertainty of estimates does not mean that the approach is Bayesian, and we can call the distribution a posterior. We can quantify uncertainty in a frequentists sense. It seems that the approach counts the number of retrievals associated with Tbs within bins. Then the bin with maximum is labeled. The problem is then defined as classification problem and the output of the softmax function is considered as the posterior distribution of the retrievals. Even though, I found the approach creative, I am not convinced that it is a Bayesian approach.

**Author response:**

The connection between probabilistic neural network retrievals and Bayesian retrieval methods has been shown in Pfreundschuh et al. (2018). A reference to this article is included in l. 135 of the first version of the manuscript, which also states that the distribution of the training data in this case corresponds to the a priori distribution.
It is of course possible to interpret the probabilistic results in a frequentist sense, however, the Bayesian framework is, at least in our experience, more common for inverse problems in satellite remote sensing. It also has the advantage that it emphasizes the dependence of the retrieval results on the a priori assumptions, i.e., the training data of the neural network.

Since the relation between training data and a priori distribution of Bayesian retrievals is fundamental to our work, we will revise the manuscript to better convey the significance of the training data in the Bayesian retrieval framework.

**Changes in manuscript**

- We have added the following paragraph to the introduction that introduces the probabilistic regression approach upon which Hydronn is based.

  > **Changes starting in line 77:**
  >
  > Pfreundschuh et al. (2018) have shown that when a retrieval is cast as a probabilistic regression problem and solved using a neural network, the obtained results are equivalent to those obtained using traditional Bayesian retrieval methods, given that the a priori distribution matches the distribution of the data used to train the neural network. Neural-network-based probabilistic regression techniques thus provide a powerful and flexible way of combining recent advances in deep learning with the theoretically sound handling of retrieval uncertainties of Bayesian retrieval methods. Hydronn builds on this approach and uses a convolutional neural network (CNN) to predict a binned approximation of the probability density function (PDF) of the marginal posterior distribution of each output pixel.

- We have reformulated the beginning of the section that introduces the probabilistic retrieval approach used by Hydronn.

  > **Changes starting in line 246:**
  >
  > A defining characteristic of Hydronn is that precipitation is retrieved using a Bayesian framework. This means that, instead of predicting a single precipitation value, it provides an estimate of the full a posteriori distribution of the Bayesian retrieval problemHydronn builds on the findings from Pfreundschuh et al. (2018), which showed that probabilistic regression with neural network yields the same results as a traditional Bayesian retrieval using an a priori distribution that is the same as the training data of the neural network. Although Pfreundschuh et al. (2018) proposed to use used quantile regression neural networks (QRNNs) to perform Bayesian remote sensing retrievals with neural networks, a different approach is taken here. Following the work by Sønderby et al. (2020), the range of possible precipitation values is discretized and the probability neural network output is used to predict the probabilities of the observed precipitation falling into each bin is predictedany of the precipitation bins. By normalizing the predicted probabilities by the width of the corresponding bin, a binned approximation of the probability density function (PDF) of the Bayesian a posterior distribution can be obtained. We found this approach to be equivalent to QRNNs in retrieval accuracy. However, calculating the

distribution of the sum of two temporally independent predictions is easier on the binned PDF than on the predicted quantiles, which why the former approachwas chosen for the implementation of Hydronn.

**Reviewer comment 6**

It is claimed that spatially aware CNNs provide more accurate retrievals than pixel-level DNNs. The reason is not discussed, and no evidence is provided.

**Author response:**

The evidence for the higher accuracy of CNN retrievals stems from a preliminary study to which a reference is provided in the manuscript. Since it seem that this has not been made sufficiently clear, we will rewrite the section to more clearly state where these results can be found.

**Changes in manuscript**

- We have reformulated the paragraph that mentions the preliminary study.

    **Changes starting in line 203:**

    A preliminary study found CNNs to yield significantly more accurate results than CNNs have been shown to be able to learn semantic features directly from image data (Selvaraju et al., 2017), which sets them apart from conventional regression techniques. Since satellite imagery of clouds exhibits patterns that can be related to different precipitation regimes, we expect this information to help to constrain the precipitation retrieval. In fact, a preliminary study we have conducted found that CNNs yield more accurate precipitation retrievals than a fully-connected neural networks that use only a single pixel as input (Ingemarsson, 2021). The fully-convolutional networks are constructed using what we refer to as Xception blocks, which are based on the Xception architecture proposed by Chollet (2017). These blocks are combined in an asymmetric network operating on independent pixels. The results have been published as parts of a Master's thesis and are available online (Ingemarsson, 2021).

**Reviewer comment 7**

In equation 2, when the prior probability approaches to a small number, the likelihood ratio can be extremely large. The correction numbers in Fig. 4 are too large. Please explain why such a large difference might exist in the retrievals that need such a large correction factor. For correcting probability distribution we can use a simple CDF matching!

**Author response:**

Upon revisiting the likelihood ratios, we have come to the conclusion that the calculation presented in the first version of the manuscript was not correct. Instead of using the training data to calculate the correction factors, we will recalculate the probability ratios using a priori distributions derived from retrieval results. This will likely decrease the magnitude of probability ratios. However, large probability ratios are still possible whenever the a priori distribution of the retrieval approaches zero. Therefore, differences in the measurement characteristics between the GPM combined retrieval and the gauge measurements can still lead to large probability ratios.

This is certainly a drawback of our approach. However, the CDF matching approach proposed by the reviewer is typically used to correct scalar retrieval results. We are, therefore, not aware of a way to apply the method the probabilistic output provided by our retrievals.

**Changes in manuscript**

- We have devised a different method to derive the correction which leads to smaller correction factors.

  > **Changes starting in line 320:**
  >
  > The difficulty with this approach is that we only know the a priori distribution  corresponding to the instantaneous precipitation retrievals, i. e., the distribution of the training data, but not for the hourly accumulations retrieved using Hydronn. To infer them, we calculated hourly accumulations for randomly sampled hours over the full year of 2019 for each retrieval configuration. The resulting correction factors for the $\text{Hydronn}_{4,\text{All}}$ retrieval are displayed in Fig.  1.9.

**Reviewer comment 8**

The resolution of IR is higher than microwave data. In this sense, you have redundant samples. How were those samples treated in the training?

**Author response:**

We did not treat these samples in any particular way. Since all training samples are revisited multiple times during the training anyways, the induced redundancy is unlikely to be an issue for the retrieval.

[Figure]

Figure 1.9: A priori distributions of hourly accumulations and derived correction factors $r$ for the Hydronn$_{4,\mathrm{All}}$ retrieval. Panel (a) displays the a priori distributions of hourly precipitation accumulations derived assuming strong temporal dependence of measurements (blue) and complete independence (red). The gray, dashed line shows the PDF of the gauge measurements. Panel (b) displays the corresponding correction factors for the two assumptions calculated as the ratio between the respective PDFs and the PDF of the gauge measurements.

**Reviewer comment 9**

Explanation of the uncertainty quantification is too complex. Please consider simplifying the text and provide improved explanations.

**Author response:**

We will rewrite the section describing the approach to quantify uncertainties aiming to make it easier to understand.

**Changes in manuscript:**

- We have rewritten the section that describes the calculation of the distributions of hourly precipitation accumulations and added an example.

    **Changes starting in line 279:**

    #### 1.1.1 Calculation of hourly accumulations

    The precipitation estimates produced by Hydronn correspond to instantaneous precipitaiton rates. Since GOES 16 imagery is available every 10 minutes, a method is required to  aggregate the retrieved distributions of the instantaneous precipitation rates to hourly accumulations

,  in order to compare them to the gauge measurements. While this is not an issue when only the posterior mean is retrieved, it is unclear how  retrieval uncertainties should be  accumulated in time. The problem is illustrated in Fig. 2.2 using six, consecutive retrievals at a single output pixel. The green lines show the retrieved distributions for each input observation. Because Hydronn has no way of modeling the correlations between consecutive observations it is not clear how the instantaneous distributions can be aggregated to a posterior distribution for the hourly accumulations.

In lack of a formal way to resolve this, we have implemented two heuristics for calculating probabilistic estimates of hourly accumulations from instantaneous measurements.

The first heuristic is to  average the predicted posterior distributions. For the case of multiple identical observations, this preserves the retrieval uncertainties and thus corresponds to the assumption of strong dependence of the retrieval errors for consecutive observations. The second approach is to assume temporal independence of the retrieval uncertainty.

The blue and red curves in Fig. 2.2 show the resulting posterior distributions of the hourly accumulations for the assumptions of dependent errors and independent errors, respectively. Despite the differences in the two distributions they both have the same mean value. Under the assumption of temporal independence, the instantaneous retrieval errors have a tendency to compensate for each other, which reduces the retrieval uncertainty. Conversely, strongly dependent errors have a tendency to conserve the uncertainty of the instantaneous retrieval resulting in higher probabilities assigned to stronger precipitation.

**1.2 Minor comments**

**Reviewer comment 1**

Why both the second and third configurations are needed. They are just different in resolution. Line 160. Provide reasoning.

**Author response:**

We will elaborate on the motivation for the third configuration retrieval configuration.

[Figure]

Figure 1.10: Retrieved posterior distributions of instantaneous precipitation (green, solid lines) for an hour of ABI observations. The corresponding derived distributions for the hourly accumulations are show in red and blue for the assumptions of dependent and independent errors, respectively.

**Changes in manuscript**

- We have extended the motivation of the choice of the tested retrieval configurations.

  > **Changes starting in line 190:**
  >
  > The second retrieval configuration, denoted as $\text{Hydronn}_{4,\text{All}}$, uses all available GOES channels at a resolution of 4 km. , which is the resolution at which both HYDRO and PERSIANN CCS are operating. This model configuration is included to assess the benefit of including all ABI channels in the retrieval.
  >
  > The third configuration, $\text{Hydronn}_{2,\text{All}}$,  uses all observations at their native resolution and  retrieves precipitation at the resolution of the 2 km channels. This means that GOES Ch. 2 is ingested at a resolution of 500 m, Ch. 1, 3 4 at a resolution of 1 km and the remaining channels at 2 km. This is in contrast to the other Hydronn configurations and other precipitation retrievals which typically ingest all observations at the same resolution. This configuration aims to explore the extent to which high resolution observations can improve precipitation retrievals even if the reference precipitation measurement only have a resolution of 5 km. Comparison of the $\text{Hydronn}_{4,\text{All}}$ and $\text{Hydronn}_{2,\text{All}}$ configuration aims to address the question whether the increased computational complexity of $\text{Hydronn}_{2,\text{All}}$ can be justified by improvements in retrieval accuracy. The characteristics of the three configurations are summarized in Table 1.

**Reviewer comment 2**

Line 185. The range is too wide! The training GPM combined precipitation can range from 0.1 to 200 mm/hr. Why 1000 mm/hr?

**Author response:**

The range that we employ for the probability bins is certainly excessively wide. While it is true that this range could be reduced, this is very unlikely to affect the performance of the retrievals in any way. We thus don't consider this to be an issue.

**2 Comments from reviewer 2**

**2.1 Specific comments**

**Reviewer comment 1:**

What is the purpose for the 4-km experiments given that the ABI has a native resolution of 2 km?

**Author response:**

The principal motivation for the 4-km experiments is that the current operational algorithm, HYDRO, operates at this resolution. Running the operational retrieval at 2-km resolution quadruples the computational and storage requirements. The 4-km experiments thus serve as a baseline to assess the benefits of running the retrieval at a higher resolution.
Furthermore, the GPM combined retrieval, which is used to generate the training data, has a comparably low resolution of 5 km. Therefore, it is not evident that the retrieval can benefit from the increased resolution of the input data.

**Changes in manuscript:**

- We will reformulate the section that introduces the retrieval configurations and motivate them more clearly.

  > **Changes starting in line 190:**
  >
  > The second retrieval configuration, denoted as $\text{Hydronn}_{4,\text{All}}$, uses all available GOES channels at a resolution of 4 km, which is the resolution at which both HYDRO and PERSIANN CCS are operating. This model configuration is included to assess the benefit of including all ABI channels in the retrieval.
  >
  > The third configuration, $\text{Hydronn}_{2,\text{All}}$,  uses all observations at their native resolution and  retrieves precipitation at the resolution of the 2 km channels. This means that GOES Ch. 2 is ingested at a resolution of 500 m, Ch. 1, 3 4 at a resolution of 1 km and the remaining channels at 2 km. This is in contrast to the other Hydronn configurations and other precipitation

retrievals which typically ingest all observations at the same resolution. This configuration aims to explore the extent to which high resolution observations can improve precipitation retrievals even if the reference precipitation measurement only have a resolution of 5 km. Comparison of the Hydronn$_{4,\text{All}}$ and Hydronn$_{2,\text{All}}$ configuration aims to address the question whether the increased computational complexity of Hydronn$_{2,\text{All}}$ can be justified by improvements in retrieval accuracy. The characteristics of the three configurations are summarized in Table 1.

**Reviewer comment 2:**

Line 152: The availability of sunlight does not affect the other IR and WV bands, only the availability of VIS bands. Therefore, it does not justify the use of only a single IR channel. Please clarify the reasoning here.

**Author response:**

Hydronn$_{4,\text{IR}}$ uses only the IR window channel because the same channel is used by the HYDRO and PERSIANN CCS retrievals. Therefore, the Hydronn$_{4,\text{IR}}$ configuration can be used to assess the benefit of the neural-network-based retrieval over the traditional power-law-based retrieval. Moreover, because this channel has been continuously available on a long sequence of geostationary sensors, it is suitable for the generation of precipitation records and used, for example, in GPM IMERG (Huffman et al., 2020) and the PERSIANN CDR datasets (Ashouri et al., 2015).

**Changes in manuscript:**

- We will reformulate the section that introduces the Hydronn$_{4,\text{IR}}$ retrieval configuration.

  **Changes starting in line 182:**

  The Hydronn retrieval has been implemented in three different configurations in order to assess how the choice of input observations and  their resolution affects its performance. The most basic retrieval configuration is the Hydronn$_{4,\text{IR}}$ retrieval, which only uses brightness temperatures from the  10.3 $\mu$m channel as input.  The same longwave-IR window channel is also used by HYDRO  and the PERSIANN CCS retrieval. The availability of similar channels on a long  time series of geostationary sensors makes them suitable for the generation of climate data records.  Since this retrieval uses the same input as HYDRO and PERSIANN CCS it allows assessing the benefit afforded by the probabilistic, neural-network-based

retrieval technique used by Hydronn.

**Reviewer comment 3:**

Lines 155-156: How are the values of the visible and near-IR bands treated by the CNN to differentiate daytime from nighttime scenes? Or is this something the CNN does without any intervention?

**Author response:**

The CNN is trained with input from all GOES channels regardless of the time of the day. Therefore, it learns to handle both day- and night-time observations, and no intervention is required to handle them.

**Changes in manuscript:**

- We will add a sentence to the description of the training scheme to mention this feature of the CNN retrieval.

  **Changes starting in line 240:**

  All available collocations from the training period are used for the training and no distinction is made between day- and nighttime observations. The Adam optimizer (Kingma and Ba, 2014) with an initial learning rate of $0.0005, \beta_1 = 0.9, \beta_2 = 0.99$ and a cosine-annealing learning rate schedule (Loshchilov and Hutter, 2016) is used for training. Warm restarts are performed every 20 epochs and repeated until the retrieval accuracy on a held-out part of the training data converges. Training of a single retrieval model takes about 3 days on an NVIDIA V40 GPU.

**Reviewer comment 4:**

Sections 3.3 and 3.4: The description of the CNN needs much more detail to be understood by readers who are not experts on CNNs.

**Author response:**

We will extend the description of the CNN model upon which Hydronn is based.

**Changes in manuscript:**

- Section 3.3, which describes the neural network model used by Hydronn, will be extended.

  **Changes starting in line 202:**

**2.1.1 Neural network model**

All Hydronn retrievals are based on a  similar convolutional neural network (CNN) architecture, which is illustrated in Fig. 2.1.  CNNs have been shown to be able to learn semantic features directly from image data (Selvaraju et al., 2017), which sets them apart from conventional regression techniques. Since satellite imagery of clouds exhibits patterns that can be related to different precipitation regimes, we expect this information to help to constrain the precipitation retrieval. In fact, a preliminary study we have conducted found that CNNs yield more accurate precipitation retrievals than a fully-connected neural  network operating on independent pixels. The results have been published as parts of a Master's thesis and are available online (Ingemarsson, 2021).

[revised manuscript text omitted]

**Reviewer comment 5:**

Lines 181-183: Please briefly define terms such as cross-entropy loss, logits, and softmax activation that would probably be unfamiliar to most readers.

**Author response:**

We will add definitions of these terms to the manuscript.

**Changes in manuscript:**

- The softmax activation function is defined in Eq. (1.1).

- A definition of cross-entropy loss is added to the manuscript.

> **Changes starting in line 258:**
>
> DRNNs can be implemented by treating the retrieval as a classification problem over a discretized range of precipitation values and using the cross-entropy loss to train the network.  The cross-entropy loss is defined as
>
> $$L(\vec{\hat{y}}, y) = -\log(\hat{y}_{\mathrm{bin}(y)}) \tag{2.2}$$
>
> where $\vec{\hat{y}}$ is the vector of probabilities predicted by the network  and $\mathrm{bin}(y)$ is the index of the probability bin corresponding to the true precipitation rate $y$.

**Reviewer comment 6:**

Line 165: Why is downsampling done for the 2-km retrievals rather than the 4-km retrievals? Shouldńt it be the other way around?

**Author response:**

Hydronn$_{2,\mathrm{All}}$ ingests GOES observations at resolutions of $500\,\mathrm{m}$, $1\,\mathrm{km}$ and $2\,\mathrm{km}$, which means that the network has to handle three input streams of different sizes. The additional downsampling layers in the Hydronn$_{2,\mathrm{All}}$ retrieval are required to reduce the size of the $500\,\mathrm{m}$ and $1\,\mathrm{km}$ inputs so that they can be combined with the observations at $2\,\mathrm{km}$ resolution. Since the network applies learnable transformations before and after the downsampling layers, it can learn to make use of the information at $500\,\mathrm{m}$ and $1\,\mathrm{km}$ resolution. This would not be the case if the observations were down-sampled prior to feeding them into the network.

The inputs for the 4 km retrievals are down-sampled prior to feeding them into the network. Therefore no additional downsampling layers are required to reduce the size of the input to that of the output.

**Changes in manuscript:**

- We have extend the description of the neural-network architecture to better explain the role of the downsampling layers. See changes in response to comment 4.

**Reviewer comment 7:**

Line 167: Please provide references to support this assertion about the number of internal features relative to other architectures.

**Author response:**

We reconsidered the sentence in question an decided to remove it from the manuscript. The principal reason for this is that by correcting a bug in our training code we were able to increase the number of features used in the NN architectures. In retrospect, we also consider the sentence imprecise and not really helpful for the reader.

**Changes in manuscript:**

- We have removed the sentence in question from the manuscript. See changes in response to comment 4.

**Reviewer comment 8:**

Line 178: What in particular makes it easier to compute this sum on the binned PDF than on the quantiles? Please explain this more thoroughly.

**Author response:**

The principal reason for this is that we are not aware of any other way to calculate the distribution of the sum of two independent random variables.
Therefore, if a distribution is given in terms of a sequence of quantiles, it would be necessary to (1) use the quantiles to calculate the PDF of the distribution, (2) calculate the binned PDF of the sum of the variables, and finally (3) compute the desired quantiles of the resulting distribution.
If, on the other hand, the retrieval results is already a PDF in binned form, the sum can be calculated directly.

**Changes in manuscript:**

- We will extend the explanation of the calculation of the sum of the retrieval results to make the advantage of the binned PDF format clearer.

**Changes starting in line 253:**

We chose this approach, which we  will refer to as density regression neural network (DRNN),  because we didn't find an efficient way to calculate the sum of two independent random variables from the quantiles predicted by a QRNN. For two PDFs given over discrete bins the sum can be calculated by weighing all possible sums of bin centers by the product of the corresponding probabilities, and accumulating the resulting probabilities into the bins of the result PDF.

**Reviewer comment 9:**

Line 186, 442: Please explain what the degeneracy of (low) quantiles means.

**Author response:**

Non-raining pixels, which are assigned a precipitation rate (PR) of exactly $0\,\mathrm{mm\,h^{-1}}$, cause a discontinuity in the CDF of the PR distribution. Since this makes it impossible to invert the CDF, not all quantiles are well defined. For example, it is impossible to determine the 25th percentile of the CDF of a pixel that is assigned a raining probability of $50\,\%$. This is because the CDF is 0 for all PR $< 0$ and larger than 0.5 for all PR $\geq 0$.

**Changes in manuscript:**

- We will improve reformulate the sentence in question.

  **Changes starting in line 266:**

  The reference precipitation of pixels without rain was set to a log-uniform random value between $10^{-3}$ and $10^{-2}$ mm h$^{-1}$.  Replacing zero values with small random values has the advantage of  making the corresponding cumulative distribution function (CDF) continuous, which ensures that all quantiles of the  distribution are always well defined. This allows us to verify the  probabilistic predictions from the network using calibration curves.

**Reviewer comment 10:**

Line 189: What is the rationale for creating outputs for 128 bins if only 14 quantiles are used?

**Author response:**

The retrieval results of Hydronn are represented as a binned approximation of the posterior PDF. The number of bins must be such that the full range of possible output values is covered and that the bins are sufficiently fine to ensure that they can accurately represent the posterior PDF in the region where most of its mass is located. Since it is impossible to know a priori where the mass will be located, Hydronn employs a relatively fine grid across the full range of possible output values.

A posteriori, the PDF can be represented more compactly using quantiles. To ensure that these quantiles, but also the mean, mode and samples of the posterior distribution, can be calculated with high accuracy, Hydronn such a high number of output bins.

**Changes in manuscript:**

- We will extend Sec. 3.4 to more clearly motivate the choice for number of output bins.

  **Changes starting in line 271:**

  Those are the posterior mean as well as a sample and 14 quantiles of the posterior distribution. Note that the quantiles are always located around the region of the posterior distribution that contains most of its mass and thus provide a much compacter way of storing the probabilistic retrieval results than the full 128 probabilities.

**Reviewer comment 11:**

Line 193: What does inference mean in this context?

**Author response:**

In the field of machine learning, 'inference' refers to the application of a statistical model to unseen data. It is used to distinguish the actual usage of a machine-learning model from its training process. In this case it means during the retrieval processing.

**Changes in manuscript:**

- We will rephrase the sentence in question.

  **Changes starting in line 276:**

  Compared to training a separate classifier to perform this task, this approach has the advantage that the precipitation threshold can be chosen dependent on the application context after the network has been trained.

**Reviewer comment 12:**

Line 196: What does posterior mean in this context?

**Author response:**

The posterior distributions here are just the results of the retrieval for each observation.

**Changes in manuscript:**

- We will rephrase the sentence in question.

  > **Changes starting in line 280:**
  >
  > The precipitation estimates produced by Hydronn correspond to instantaneous precipitation rates. Since GOES 16 imagery is available every 10 minutes, a method is required to  aggregate the retrieved distributions of the instantaneous precipitation rates to hourly accumulations   in order to compare them to the gauge measurements.

**Reviewer comment 13:**

Lines 203-204: Why specifically will assuming that the retrieval uncertainty is temporally independent cause the uncertainty to decay for consecutive identical observations?

**Author response:**

Our assertion that the retrieval uncertainty of independent measurements decays is based on the following reasoning:
Given a sequence of random variables $X_1, \ldots X_n$ with finite mean and finite and bounded variance, let $\sigma_{\max} = \max\{\mathrm{Var}(X_1), \ldots \mathrm{Var}(X_n)\}$ denote the maximum variance of any of the distributions. It is then possible to derive the following upper bound for the variance of the mean of the random variables:

$$\mathrm{Var}(\frac{1}{n}\sum_i X_i) = \frac{1}{n^2}\sum_i \mathrm{Var}(X_i) \tag{2.3}$$

$$\leq \frac{1}{n^2}n\sigma_{\max} \tag{2.4}$$

$$\leq \frac{\sigma_{\max}}{n} \tag{2.5}$$

Note that (1.3) holds because of the independence of the random variables. This means that the variance of the mean of the random variables will always be lower than that of the distribution with the highest variance and decay as more observations are included in the mean.

**Changes in manuscript:**

- We added an example to illustrate the effect.

  > **Changes starting in line 292:**
  >
  > The blue and red curves in Fig. 2.2 show the resulting posterior distributions of the hourly accumulations for the assumptions of dependent errors and independent errors, respectively. Despite the differences in the two distributions they both have the same mean value. Under the assumption of temporal independence, the instantaneous retrieval errors have a tendency to compensate for each other, which reduces the retrieval uncertainty. Conversely, strongly dependent errors have a tendency to conserve the uncertainty of the instantaneous retrievals resulting in higher probabilities assigned to stronger precipitation.

[Figure]

Figure 2.2: Retrieved posterior distributions of instantaneous precipitation (green, solid lines) for an hour of ABI observations. The corresponding derived distributions for the hourly accumulations are show in red and blue for the assumptions of dependent and independent errors, respectively.

**Reviewer comment 14:**

Lines 279-280: This is true, but it would be very scientifically interesting to see the relative degree of improvement during the day and night e.g., to quantify the value of the visible and near-IR data.

**Author response:**

We agree with the reviewer that this would be an interesting question to investigate. However, we fear that a simple comparison of retrievals during day and night time would be misleading due to the confounding effect of the daily cycle of precipitation. A fairer comparison may be to add an additional retrieval configuration to the study. However, considering the associated computational cost and that the objective of our study was

maximizing the accuracy of the precipitation retrievals, we consider this to be outside the scope of our study.

**Reviewer comment 15:**

Line 342: Why does assuming dependent retrieval errors lead to the uncertainties being overestimated?

**Author response:**

That the assumption of dependent retrieval errors causes uncertainties to be estimated can be seen from the fact that the calibration curves in Fig. 10 lie above the diagonal. This means that the retrieved confidence intervals are too wide, which causes the true precipitation value to lie within them more often than is expected based on the interval. The observation that the assumption of dependent retrieval errors leads to an overestimation of the uncertainties is primarily an experimental result. Likely, the reason for this is that the true errors are not completely dependent.

**Changes in manuscript:**

- We will extend the discussion of Fig. 10 to reflect the above reasoning.

> **Changes starting in line 474:**
>
> For the assumption of dependent retrieval errors, the calibration curve tends to lie above the diagonal, which signifies that the true precipitation values fall into the predicted interval more often than expected. The retrieved confidence intervals thus overestimate the retrieval uncertainty. The opposite effect is observed for the assumption of independent errors. Applying the a priori correction  improves the calibration for both assumptions.

**Reviewer comment 16:**

Lines 361-362 and 473: How precisely does varying the probability threshold have a calibrating effect on the retrieval results?

**Author response:**

Upon reconsidering the statements in question, we have come to the conclusion that our results don't provide any evidence of a calibrating effect. We will therefore remove the statements from the manuscript.

**Changes in manuscript:**

- We will remove the statements from the manuscript.

**Reviewer comment 17:**

Lines 378-379: What probability threshold was tuned, and why was a FAR close to IMERG the criterion for doing so?

**Author response:**

Due to its probabilistic nature, Hydronn is able to predict the probability that the precipitation at a given pixel exceeds a certain threshold. This probability can be used to detect strong precipitation by choosing a probability threshold above which a pixel is assumed to contain heavy precipitation. The probability threshold can be used to tune either POD or FAR to an arbitrary value. In practice, the threshold should be chosen according to the application at hand. Since purpose of the evaluation was the comparison to IMERG, we tuned the FAR to that of IMERG since makes the results easier to compare.

**Changes in manuscript:**

- Because this section did not contribute any novel information to the analysis that wasn't already contained in the PR curves, we decided to remove it from the revised manuscript.

**Reviewer comment 18:**

Table 4: Why precisely does correcting when assuming independent errors actually degrade the POD, FAR, and CSI relative to the uncorrected version?

**Author response:**

We think that the degradation of POD, FAR, and CSI for the assumption of independent retrieval uncertainties is due to an error in the calculation of the correction factors. The calculation of the a priori distribution of hourly accumulations assumed fully independent samples of the a priori distribution of instantaneous precipitation estimates. Since this neglects the dependence that is introduced by the temporal coherence of the satellite observations, it likely caused the correction factors to be incorrect.

While we changed the method to calculate the correction factors in the revised manuscript, the corrections still didn't lead to consistent improvements for the detection of extreme precipitation. Our interpretation is that the current approach remains too crude to improve the detection of these rare events.

**Changes in manuscript:**

- We have reformulated the discussion of the proposed correction to reflect these findings.

  > **Changes starting in line 602:**
  >
  > We have proposed a method to correct for variations in the distribution of precipitation  in the training data ~~. The corrections have improved the agreement between the distribution of retrieved precipitation rates as well as the calibration of the uncertainty intervals (Fig. 9, Fig. 10). Although for the assumed independent uncertainties the calibration was improved, the distribution of precipitation rates did exhibit slight deviations from the distribution of the gauge measurements. We suspect that the reason for the correction working worse in the latter case is that the a corresponding a priori assumption deviates stronger from the distribution of the gauge measurements (Fig. 9). This led to much higher correction factors, which were truncated to avoid numerical issues.~~
  >
  >  relative to comparable ground validation data. The most distinct effect of the a priori correction was observed when the predicted confidence intervals were evaluated against gauge data (Fig. 15, Fig. A1). This allowed us to show that the Hydronn retrievals can provide well-calibrated uncertainty estimates for their predictions when the differences between the a priori distributions of the training data and the gauge measurements are taken into account.
  >
  > However, the correction  had only minor effects on the observed distribution of precipitation and did not improve the calibration of the detection of heavy precipitation events. We suspect the reason for this to be that the correction mostly affects
  >
  >  the probabilities of light precipitation, which because of their frequency have a strong effect on the calibration of the confidence intervals. However, the statistics used to derive the  correction may not be precise enough to correct

for the differences of the much rarer heavy precipitation events. Whether more specialized corrections that take into account seasonal variability can help with the detection of extreme precipitation remains to be investigated.

**Reviewer comment 19:**

Figure 13: The use of grayscale for the rain rates and colors for the errors makes the plots very hard to read. Would it be possible to instead plot the satellite rain rates in color and plot the corresponding gauge values using the same color scheme? Similar values would have very little contrast whereas large errors would produce sharp contrasts.

**Author response:**

We agree with the reviewer that Fig. 13 contains too much information for a single plot. We will revise the figure.

**Changes in manuscript:**

- We have revised Fig. 13 to only show the gauge measurements and the corresponding retrieved accumulations separately. The new Fig. 13 is shown in Fig. 2.3.

[Figure]

Figure 2.3: Retrieved precipitation accumulations for an extreme precipitation event in the city of Duque de Caxias in the state of Rio de Janeiro. Panel (a) shows gauge-measured precipitation accumulations using colored hexagons. Locations of the gauges are marked using red points. The red star marks the location of the Xerém neighborhood of Duque de Caxias in which the gauge station closest to the reported flooding is located. The remaining panels show the retrieved precipitation accumulations for the tested retrieval algorithms.

[Figure]

Figure 2.4: The 99th percentile of the distribution hourly precipitation during December 2020 of each gauge station. Points show the locations of the gauges used for the evaluation during December 2020. The coloring shows the 99th percentile of the distribution of hourly precipitation

**Reviewer comment 20:**

Line 432 and Fig. 15: Please define more precisely what the 99th percentile of the distribution means. If each point in Fig. 15 is the 99th percentile of all of the rainfall values for a particular gauge location during the month of Dec. 2020, why are there so many values < 5 mm/h? Is it the dry season in some of these locations?

**Author response:**

Figure 15 does, in fact, show the 99th percentile of the rainfall values for each gauge location for December 2020. Although December is generally the beginning of the rain season in many parts of the country, the 99th percentile of the hourly precipitation remains below 5 mm for several stations.

As can be seen in Fig. 2.4, most of them are located in the semi-arid east of the country which makes these results plausible. In addition to this, the precipitation in December 2020 was below average in large parts of the country (Source in Portuguese: Grupo de Previsão de Tempo CPTEC/INPE, 2020).

Nonetheless, some of the stations in the western parts of the country exhibit very low values for the 99th percentile of the distribution of precipitation. This indicates that some of these measurements may be faulty.

**Changes in manuscript:**

- We have updated the caption of Fig. 15, which now looks as shown in Fig. 2.5.

[Figure]

Figure 2.5: Scatter plots of the 99th percentile of the monthly distribution of hourly precipitation accumulations of each gauge station for June (blue) and December (red) plotted against the 99th percentile of the corresponding retrieved distribution of precipitation accumulations. The results of the Hydronn retrievals use samples from the posterior distribution of hourly accumulations obtained assuming dependent retrieval errors instead of the posterior mean.

**Reviewer comment 21:**

Lines 435-436: Are there any specific assertions in the published literature that HYDRO and PERSIANN-CCS were both developed to correctly represent heavy precipitation at the presumed expense of skill for lighter precipitation?

**Author response:**

We would like to thank the reviewer for commenting on this statement, whose formulation we consider problematic in hindsight. A more suitable statement would be that HYDRO and PERSIANN CCS were developed with a focus on convective precipitation at the expense of retrieval skill for stratiform scenarios. This is also acknowledged in the published literature.

**Regarding HYDRO:**

The HYDRO retrieval is based on the Hydroestimator, which is in-turn based on the Autoestimator. The study by Vicente et al. (1998) presents the original form of the Autoestimator. Regarding the data used to derive the regression curve that relates IR brightness temperatures and precipitation rates the manuscript states:

The original set of observations, collected during the months of March to

June 1995, was composed of 120 pairs of IR cloud-top temperatures and radar- derived rainfall estimates with 4 km by 4 km pixel resolution. Only convective rain systems were considered.

Although the algorithm includes corrections to adopt the relation to other meteorological regimes, the methodology exhibits a bias for convective precipitation, which is also acknowledged in the conclusions of the paper:

> Independent and qualitative studies not shown in this paper have demonstrated that in contrast to the reasonable performance of the technique for well-defined and short duration convective systems, poor results are common for stratiform cloud systems

Moreover, the study by Scofield and Kuligowski (2003) (titled 'Status and Outlook of Operational Satellite Precipitation Algorithms for Extreme-Precipitation Events') that introduces the Hydroestimator states:

> All of the estimates display relatively little bias for cold-top events, which is not surprising given that they were calibrated for such events and the assumptions behind satellite QPE algorithms generally work best for cold-top events.

**Regarding PERSIANN CCS:**

The available literature on the PERSIANN CCS algorithm indicates that the original cloud classes and $T_B$-precipitation curves are based on collocations from just a single summer month over the western CONUS (Hong et al., 2004):

> After completing the cloud-patch feature extraction, the system is calibrated using GOES infrared images and radar-rainfall maps for June 1999 over the region of 25 °N  45 °N and 100 °N130 °W (both datasets are mapped to 0.04° latitude × 0.04° longitude scale).

Although an a bias correction based on passive-microwave data has been added to the operational algorithm (Karbalaee et al., 2017), the underlying estimation method seems to have remained the same.
Moreover, Nguyen et al. (2018) states

> Recent developments include integrating deep learning approaches, adding water vapor channel information (Tao et al., 2017), using PMW data for bias adjustment of PERSIANN-CCS (Karbalaee et al., 2017), incorporating MODIS and CloudSat information (Nasrollahi et al., 2013), and using probability matching methods to improve warm rainfall detection in PERSIANN-CCS.

indicating that the precipitation from warm clouds remains an issue for the algorithm. In summary, both retrievals were developed with a focus on convective precipitation. A likely better explanation for the improved accuracy in estimating extreme precipitation is therefore that by restricting the analysis to the high percentiles of the precipitation distribution the contribution from convective precipitation events is increased, which leads to the observed improved performance from the two retrieval algorithms.

**Changes in manuscript:**

- We will reformulate the sentence to state that HYDRO and PERSIANN CCS were developed with a focus on convective precipitation.

  > **Changes starting in line 573:**
  >
  > To illustrate this, Fig. 2.5 shows scatter plots of the 99th percentile of the distribution of gauge-measured and retrieved precipitation during December 2020 for all gauge station . Also here the Hydronn retrievals yield the best estimates. For this evaluationmonthly distribution of hourly accumulations at each gauge station and the 99th percentile of the corresponding retrievals for June and December 2020. HYDRO and PERSIANN CCS yield similar accuracy as accuracy similar to IMERG in this analysis, despite IMERG having higher accuracy for all other metrics considered in this study. This is likely because both Both HYDRO and PERSIANN CCS were both developed to correctly represent heavy precipitation , which harms their accuracy in terms of other statistics. developed with a focus on convective precipitation. The regression relations underlying both retrievals were developed from summer precipitation in the US and enforce monotonically decreasing relationship (Hong et al., 2004; Vicente et al., 1998)between brightness temperatures and precipitation rates. This may explain why they succeed in representing heavy, convective precipitation events but fail to represent more general conditions. By explicitly resolving the probabilistic nature of the precipitation retrieval, HYDRONN can provide both climatologically accurate accumulations (see Tab.Table 3) and correct improved representation of heavy precipitation.

**2.2 Technical Comments**

**Reviewer comment 1:**

Line 39: For consistency, it might be better to cite Schmit et al. (2018) instead of Schmit et al. 2005) since the former is cited in lines 65 and 129.

**Author response:**

Since Schmit et al. (2005) is a peer-reviewed publication, we are under the impression that it is more suitable as reference for the GOES ABI. We will therefore replace the

existing citation of Schmit et al. (2018) with Schmit et al (2005).

**Changes in manuscript:**

- We will replace the references to Schmit et al. (2018) with Schmit et al. (2005).

  > **Changes starting in line 73:**
  >
  > This study presents Hydronn, a novel real-time precipitation retrieval that uses VIS/IR observations from the GOES 16 Advanced Baseline Imager (ABI, Schmit et al., 2005) to retrieve precipitation over Brazil.

**Reviewer comment 2:**

Line 46, 55, 93, 574-577: Scofield and Kuligowski (2003a) and (2003b) are the same paper.

**Author response:**

We would like to thank the reviewer for pointing out this mistake, which we will of course correct in the revised version of the manuscript.

**Changes in manuscript:**

- We will remove the duplicated reference. Note that because of the automatic the change may be rendered differently from the manuscript.

  > **Changes starting in line 50:**
  >
  > The operational use of geostationary VIS/IR observations for precipitation retrievals dates back more than 40 years (Scofield and Oliver, 1977) and a large number of different algorithms have been developed over the years  (Arkin and Meisner, 1987; Adler and Negri, 1988; Vicente et al., 1998; Sorooshian et al., 2000; Ku .

**Reviewer comment 3:**

Line 54: Please cite Nguyen et al. (2020) here in reference to PERSIANN-PDIR.

**Author response:**

We will add the citation in the revised version of the manuscript.

**Changes in manuscript:**

- We will add the citation in the revised version of the manuscript.

  **Changes starting in line 60:**

  PERSIANN CCS is superseded by the PERSIANN PDIR (Nguyen et al., 2020) algorithm, which , in addition to refining the mathematical formulation of the regression scheme of PERSIANN CCS, adds a regional correction scheme.

**Reviewer comment 4:**

Line 64: Is Hydronn an acronym (e.g., Hydro-Neural Network) or does the name have a different meaning?

**Author response:**

Hydronn is the name of the retrieval algorithm. It is named after the character Hydron from the He-man comic series. It also functions as a portmanteau of the words Hydro and NN (for neural network) but we prefer both spelling and pronounciation of Hydronn over HydroNN, which is also why we did not introduce it in this way in the manuscript.

**Reviewer comment 5:**

Line 80: Replace consists with consist (measurements is plural).

**Author response:**

We will correct this in the revised version of the manuscript.

**Changes in manuscript:**

  **Changes starting in line 101:**

  The rain gauge measurements that are used in this study were compiled by the National Institute of Meteorology of Brazil and  consist of hourly gauge  measurements covering the time range May 2000 until May 2020.

**Reviewer comment 6:**

Lines 85, 86, 88: Northwest should not be capitalized unless it is a proper name.

**Author response:**

We will correct this in the revised version of the manuscript.

**Changes in manuscript:**

> **Changes starting in line 107:**
>
>  gauge density is fairly high on the south-eastern coast of Brazil  but decreases markedly towards the  northwest.

> **Changes starting in line 112:**
>
> December 2020 saw high precipitation amounts on the south-western coast of the country extending towards the  northwest, which are associated the South Atlantic Convergence Zone (SACZ, Satyamurty et al., 1998). Very low precipitation rates are observed in the northeast of the country, which is influenced by large scale subsistence patterns (de Siqueira and Vila, 2019).

**Reviewer comment 7:**

Line 86 Many readers may not know that Amazonas is the proper name for a state in Brazil, so the Brazilian state of Amazonas would be clearer.

**Author response:**

The sentence has been removed from the revised manuscript.

**Change in manuscript:**

See changes response to reviewer comment 6.

**Reviewer comment 8:**

Line 88: Replace manifest with e.g., is associated with.

**Author response:**

We will correct this in the revised version of the manuscript.

**Change in manuscript:**

See changes in response to reviewer comment 6.

**Reviewer comment 9:**

Line 118: Replace available first with available only.

**Author response:**

We will correct this in the revised version of the manuscript.

**Changes in manuscript:**

> **Changes starting in line 148:**
>
> IMERG-Final is adjusted using global gauge measurements but available  only after 3.5 months.

**Reviewer comment 10:**

Line 135: replace criterion with approach.

**Author response:**

We agree with the reviewer that 'criterion' is not a suitable expression here. However, we think that 'interpolation' is actually more specific than 'approach', so will will use 'interpolation' in the revised version of the manuscript.

**Changes in manuscript:**

> **Changes starting in line 165:**
>
> The surface precipitation from GPM CMB was mapped to the 2 km resolution of the ABI's IR channels using  nearest-neighbor interpolation.

**Reviewer comment 11:**

Lines 133, 386, and elsewhere: please ensure that all dates in this manuscript match the format used in EGUsphere.

**Author response:**

We would like to thank the reviwer for pointing out this issue. We will of course correct this in the revised version of the manuscript.

**Changes in manuscript:**

> **Changes starting in line 166:**
>
> interpolation. Collocations were extracted for the time range  1 January 2018 until 1 January 2020 and 1 January 2021 until 1 September 2021.

**Changes starting in line 510:**

As final part of this evaluation, a case of heavy precipitation in the city of Duque de Caxias in the State of Rio de Janeiro is considered, which occurred between the  22 and 24 December 2020 and lead to  flooding (Fohla De S. Paulo, 2020).

**Reviewer comment 12:**

Line 150: A better wording would be a long time series of geostationary sensors.

**Author response:**

We will change the formulation in the revised version of the manuscript.

**Changes in manuscript:**

**Changes starting in line 186:**

The availability of similar channels on a long  time series of geostationary sensors makes them suitable for the generation of climate data records.

**Reviewer comment 13:**

Line 350: Insert to before derive.

**Author response:**

We will correct this in the revised version of the manuscript.

**Changes in manuscript:**

**Changes starting in line 483:**

The retrieved quantiles can also be used  to estimate the probability that an observed pixel  exceeds certain precipitation thresholds.

**Reviewer comment 14:**

Line 354: Is retrieved meant rather than predicted?

**Author response:**

In the field of machine learning the term 'predict' is commonly used when a model is evaluated. We have therefore used the terms 'predict' and 'retrieve' interchangeably in the manuscript.

**Reviewer comment 15:**

Line 354: Pixel should be plural.

**Author response:**

We will correct this in the revised version of the manuscript.

> **Changes starting in line 488:**
>
> For the non-probabilistic retrievals the curves were generated using the predicted precipitation and classifying all  pixels above a varying threshold as exceeding the sought-after precipitation rate.

**Reviewer comment 16:**

Line 362: Is worse detection accuracy than at 5 mm/h meant here?

**Author response:**

Yes, this is what is meant here. We will reformulate the sentence to make this clear.

**Changes in manuscript:**

> **Changes starting in line 494:**
>
> For events exceeding 20 mm h$^{-1}$, all retrievals yield worse detection accuracy than at 5 mm h$^{-1}$.

**Reviewer comment 17:**

Figure 12 caption: add at a rate of 5 mm/h to the end of the caption for clarity.

**Author response:**

We will adopt this change in the revised version of the manuscript.

**Changes in manuscript:**

The updated Fig. 12 is shown in Fig. 2.6

**Reviewer comment 18:**

Line 387: Floodings should be singular or replaced with floods.

**Author response:**

We will correct this in the revised version of the manuscript.

[Figure]

Figure 2.6: Calibration of the probabilistic precipitation event detection for precipitation exceeding 5 mm h$^{-1}$

**Changes in manuscript:**

> **Changes starting in line 511:**
>
> As final part of this evaluation, a case of heavy precipitation in the city of Duque de Caxias in the State of Rio de Janeiro is considered, which occurred between the  22 and 24 December 2020 and lead to  flooding (Fohla De S. Paulo, 2020).

**Reviewer comment 19:**

Lines 387, 404, 520: Is this citation and reference formatted correctly?

**Author response:**

According to the guidelines for referencing websites in AMT (`https://www.atmospheric-measurement-tec net/submission.html#references`), the reference should be formatted correctly except for the wording used for the last access date.

**Changes in manuscript:**

- We will change the wording for the last access date of the reference in question.

**Reviewer comment 20:**

Line 388: Replace were with was.

**Author response:**

We will correct this in the revised version of the manuscript.

**Changes in manuscript:**

> **Changes starting in line 512:**
>
> About 250 mm of accumulated precipitation  was measured by the rain gauge in the neighborhood of Xerém over the period of two days.

**Reviewer comment 21:**

Line 394: Please indicate the location of Duque de Caxias in Fig. 13.

**Author response:**

The location of Xerém, which is the neighborhood in Duque de Caxias in which the rain gauge is located, is already indicated in Fig. 13. However, the manuscript does not clearly state the relation between Duque de Caxias and Xerém. We will reformulate the introduction of Sec. 4.3 to make it clear where the flooding occurred.

**Changes in manuscript:**

The updated Fig. 13 is shown in Fig. 2.3.

**Reviewer comment 22:**

Line 430: Replace by with of.

**Author response:**

We will correct this in the revised version of the manuscript.

**Changes in manuscript:**

> **Changes starting in line 570:**
>
> The deviations of the distribution of the posterior mean from the gauge measurements  should thus be understood as a consequence  of the statistical properties of this estimator instead of a retrieval deficiency.

**Reviewer comment 23:**

Line 431: Runoff is a single word.

**Author response:**

We will correct this in the revised version of the manuscript.

**Changes in manuscript:**

> **Changes starting in line 572:**
>
> The random samples may be useful for applications that are sensitive to heavy precipitation rates, such as  runoff modeling or climatological studies.

**Reviewer comment 24:**

Line 433: Station should be plural.

**Author response:**

The sentence was reformulated in the revised version of the manuscript.

**Changes in manuscript**

> **Changes starting in line 573:**
>
> Fig. 2.5 shows scatter plots of the 99th percentile of the distribution of  hourly accumulations at each gauge station and the 99th percentile of the corresponding retrievals for June and December 2020.

**Reviewer comment 25:**

Line 434: Replace similar accuracy as with accuracy similar to.

**Author response:**

We will correct this in the revised version of the manuscript.

**Changes in manuscript**

> **Changes starting in line 575:**
>
> HYDRO and PERSIANN CCS yield  accuracy similar to IMERG despite IMERG having higher accuracy for all other metrics considered in this study.

**Reviewer comment 26:**

Line 461: A more precise wording might be correct for variations in the distribution of precipitation rates in the training data relative to comparable ground validation data.

**Author response:**

We would like to thank the reviewer for this suggestion, which is, in fact, a better description of the proposed correction scheme. We will adopt this suggestion in the revised version of the manuscript.

**Changes in manuscript:**

We have rewritten the discussion of the utility of the a priori correction given that the do not improve the detection of heavy precipitation. See changes in response to specific comment 18.

**Reviewer comment 27:**

Line 465: Replace stronger with more strongly.

**Author response:**

The sentence in question has been removed from the revised manuscript.

**Changes in manuscript:**

See changes in response to reviewer comment 18.

**Reviewer comment 28:**

Line 473: Replace small with low.

**Author response:**

The sentence in question has been removed from the revised manuscript.

**Changes in manuscript:**

See changes in response to reviewer comment 18.

**Reviewer comment 29:**

Line 475: Constant in time, space, or both?

**Author response:**

The sentence has been removed from the revised version of the manuscript.

**Changes in manuscript:**

See changes in response to reviewer comment 18.

**Reviewer comment 30:**

Line 485: Please define GPM CO in line 69 so the acronym is already defined.

**Author response:**

We have removed the acronym from the manuscript because of its infrequent use.

**Reviewer comment 31:**

Lines 485-486: the latitude range of the GPM DPR is actually 65 °S to 65 °N when the instrument swath is accounted for.

**Author response:**

We will correct this in the revised version of the manuscript.

**Reviewer comment 32:**

Line 489: This is the first time that the CNN is described as a probabilistic regression approach; this concept should be introduced earlier in the manuscript.

**Author response:**

We will revise the manuscript to introduce the concept of probabilistic regression already in the introduction.

**Changes in manuscript**

- We have added the following paragraph to the introduction that introduces the probabilistic regression approach upon which Hydronn is based.

  > **Changes starting in line 77:**
  >
  > Pfreundschuh et al. (2018) have shown that when a retrieval is cast as a probabilistic regression problem and solved using a neural network, the obtained results are equivalent to those obtained using traditional Bayesian retrieval methods, given that the a priori distribution matches the distribution of the data used to train the neural network. Neural-network-based probabilistic regression techniques thus provide a powerful and flexible way of combining recent advances in deep learning with the theoretically sound handling of retrieval uncertainties of Bayesian retrieval methods. Hydronn builds on this approach and uses a convolutional neural network (CNN) to predict a binned approximation of the probability density function (PDF) of the marginal posterior distribution of each output pixel.

**Reviewer comment 33:**

Line 494: Delete the comma after resolutions.

**Author response:**

We will correct this in the revised version of the manuscript.

**Reviewer comment 34:**

Line 587: The Python Language Foundation should be considered as starting with P since The is ignored when alphabetizing entries.

**Author response:**

We will correct this in the revised version of the manuscript.